# Escape from X inactivation is directly modulated by Xist noncoding RNA

Antonia Hauth[1,2,14], Jasper Panten[3,4,5,14], Emma Kneuss [1,12,14], Christel Picard [1,13], Nicolas Servant [6], Isabell Rall[1], Yuvia A. Pérez-Rico [1], Lena Clerquin[1], Nila Servaas [1], Laura Villacorta[1], Ferris Jung[1], Christy Luong[7], Howard Y. Chang [7,8], Judith B. Zaugg [1,9], Oliver Stegle [1,4], Duncan T. Odom [3,5,10], Edith Heard [1,11,12,15] ✉ & Agnese Loda [1,15] ✉

In placental XX females, one X chromosome is silenced during a narrow developmental time window by X-chromosome inactivation, which is mediated by Xist noncoding RNA. Although most X-linked genes are silenced during X-chromosome inactivation, some genes can escape. Here, by increasing its endogenous level, we show that Xist RNA can silence escapees well beyond early embryogenesis both in vitro, in differentiated cells, as well as in vivo, in mouse pre- and post-implantation embryos. We further demonstrate that Xist RNA plays a role in eliminating topologically associating domain-like structures spanning clusters of escapees, and this is dependent on SPEN. The function of Xist in silencing escapees and eliminating topological domains is initially fully reversible, but sustained Xist upregulation leads to irreversible silencing and CpG island DNA methylation of escapees. Thus, gene activity and three-dimensional topology of the inactive X chromosome are directly controlled by Xist, well beyond an early developmental time window.

The consequences of sex-specific gene expression, including increased levels of X-linked gene products owing to genes that escape X-chromosome inactivation (XCI), can contribute to sex-specific regulatory programmes and sexually dimorphic phenotypes, including diseases[1–7]. The mechanisms underlying escape from XCI remain obscure[8,9]. Xist RNA is essential to trigger XCI during early development[10,11], and the prevailing view has been that it becomes dispensable later on for XCI maintenance in differentiated cells, where multiple layers of epigenetic factors can lock in the silent state of the inactive

X chromosome (Xi)[12–16]. In the mouse, Xist is indispensable for the initiation of two waves of XCI[9], the first being imprinted (iXCI), leading to silencing of the paternally inherited X chromosome (Xp) by E3.5[17–20]. The second wave of XCI is random (rXCI) and occurs in the embryo proper at around implantation (-E5.5)[18]. XCI can be faithfully recapitulated in vitro upon differentiation of mouse embryonic stem (mES) cells or by ectopic Xist induction in undifferentiated cells[21–24]. At the onset of XCI, Xist RNA spreads in *cis* on one of the two X chromosomes leading to its almost complete transcriptional repression by recruiting

[1]European Molecular Biology Laboratory, Heidelberg, Germany. [2]Faculty of Biosciences, Collaboration for Joint PhD Degree between EMBL and Heidelberg University, Heidelberg, Germany. [3]Division of Regulatory Genomics and Cancer Evolution, German Cancer Research Centre, Heidelberg, Germany. [4]Division of Computational Genomics and Systems Genetics, German Cancer Research Centre, Heidelberg, Germany. [5]Faculty of Biosciences, Heidelberg University, Heidelberg, Germany. [6]Bioinformatics and Computational Systems Biology of Cancer, INSERM U900, Paris, France. [7]Center for Personal Dynamic Regulomes, Stanford University, Stanford, CA, USA. [8]Howard Hughes Medical Institute, Stanford University, Stanford, CA, USA. [9]Molecular Medicine Partnership Unit, European Molecular Biology Laboratory–University of Heidelberg, Heidelberg, Germany. [10]Cancer Research UK Cambridge Institute, University of Cambridge, Cambridge, UK. [11]Collège de France, Paris, France. [12]Present address: The Francis Crick Institute, London, UK. [13]Present address: Institute of Molecular Genetics of Montpellier, Montpellier, France. [14]These authors contributed equally: Antonia Hauth, Jasper Panten, Emma Kneuss. [15]These authors jointly supervised this work: Edith Heard, Agnese Loda. ✉e-mail: edith.heard@crick.ac.uk; agnese.loda@embl.de

SPEN[25–30]. Xist also recruits (via hnRNPK) the Polycomb Repressive Complex 1 (PRC1), which in turn recruits PRC2[31–33]. The polycomb complexes are thought to mediate the early maintenance of SPEN-induced gene silencing[31]. Later appearance of DNA methylation at CpG islands on the Xi is thought to maintain XCI[34]. Using tetracycline-responsive Xist transgenes in mES cells previously defined a critical early time window around 2–3 days of differentiation that marked a shift from Xist-dependent gene silencing to Xist-independent and irreversible XCI[12]. The capacity of Xist RNA to initiate chromosome-wide gene silencing de novo also appeared to be lost after this time window[12].

How and why a subset of X-linked 'escapees' can evade or override Xist-mediated silencing, to be bi-allelically expressed from both the active X chromosome (Xa) and the Xi is not known[35,36]. A few constitutive escapees (~3–7% in mice and ~4–11% in humans[36–39]) can evade silencing from the very onset of XCI and in most cell types and individuals and are conserved across different species[9,36]. By contrast, a larger subset of X-linked genes (at least 20% in humans and mice) show variable 'facultative' escape, becoming re-expressed following silencing in some tissues and often variably between individuals[17,18,21,40–42]. Given the profound impact that escapees may have on sexual dimorphism[43,44], understanding how XCI escape is modulated could have important implications for female biology and women's health.

Here, we set out to test whether Xist RNA plays any role in regulating escapees. Recent evidence has revealed that both reactivation of silenced genes and de-dampening of escapee expression levels from the Xi can in fact occur upon XIST/Xist loss and that this may impact tissue homeostasis and even contribute to disease[45–50]. In these studies, genes that show reactivation upon Xist loss often correspond to genes that have been shown to escape XCI in other contexts, tissues or cell types[16,45–48,50,51]. Thus, contrary to previous thinking that Xist exerts its silencing role only during early embryogenesis, it may also influence escapee expression in later life. Nevertheless, how exactly transcriptional regulation of escapees is affected by Xist is unclear and has important implications for development and disease.

## Results

### Increased Xist RNA levels can silence Xi escapee genes in NPCs

To explore whether Xist RNA levels directly modulate escapee expression in post-XCI differentiated cells, we first assessed whether messenger RNA levels of escapees correlated with Xist RNA levels in 21 clonal, female neural progenitor cell (NPC) F1 hybrid lines (129/Sv × Cast/EiJ)[52]. Allelic RNA sequencing (RNA-seq) analysis confirmed substantial variability in numbers of escapees per NPC clone[53,54], ranging from 48 to 124 escapees out of 379 informative X-linked genes (Extended Data Fig. 1a–d). Examining Xist RNA levels and escapee expression in different clones revealed a significant negative correlation, suggesting that Xist RNA may indeed play a role in modulating X-linked escapee expression on the Xi (Fig. 1a). We also checked whether in human somatic tissues, the expression levels of XIST correlate inversely with that of escapees using brain transcriptomic data from the GTEx project[55]. Even if rXCI in these samples prevents us from assessing allele-specific Xi transcriptional activity, we found that in brain areas in which we detected higher levels of XIST, escapees are expressed at lower levels, supporting our observation in mouse NPCs (Extended Data Fig. 1e).

To test the direct role of Xist RNA on XCI escape, we established a system that allows the induction of higher Xist RNA levels on the Xi in NPCs. We generated clonal NPC lines by in vitro differentiation of female TX1072 mES cells[56], which is a polymorphic F1 hybrid line ((Cast/EiJ) × (C57BL/6)) carrying a doxycycline (Dox)-inducible ptet promoter upstream of the Xist endogenous locus on the C57BL/6 X chromosome[56,57]. When grown in the absence of Dox, TX1072 mES cells undergo rXCI during differentiation and the resulting NPCs are a heterogeneous pool with either the C57BL/6 or Cast/EiJ X chromosome inactivated (Fig. 1b). We selected one clonal line (E6) in which the B6 X chromosome was inactivated and used two previously described clonal NPC lines (CL30 and CL31) both carrying an Xi of B6 origin[25] (Fig. 1b and Extended Data Fig. 1f). Thus, in these three lines, Xist expression levels from the B6 Xi can be increased by adding Dox to the culture media (Fig. 1b).

In clone E6, 133 genes escape XCI, whereas clones CL30 and CL31 show less escape, with 61 and 48 escapees, respectively (Extended Data Fig. 1f,g). We compared the escapees identified in our NPC clones with those identified in 19 previous studies in which the XCI status of X-linked genes had been assessed across many different cell types, both for rXCI and iXCI[13,18,19,21,40,41,50,53,58–68] (Extended Data Fig. 1h and Supplementary Table 1). On the basis of this comparison, we defined three categories of escapees: (1) 'constitutive' escapees

**Fig. 1 | Increased levels of Xist RNA silences Xi escapees in NPCs. a**, Scatterplot showing a correlation between average escape and Xist expression using RNA-seq data from 21 NPC clones (129/Sv × Cast/Eij genetic background). Mean escape is calculated as the average allelic ratio (Xi/(Xi+Xa)) across 379 informative genes. Normalized Xist expression is calculated as library-size scaled counts per million (CPM), divided by the value for the lowest clone. R specifies Pearson's correlation coefficient, and the P value is given by a correlation test. The error band depicts 95% confidence intervals. **b**, The experimental outline showing that TX ES cells (Cast/Eij × C57BL/6 genetic background) carrying a tetracycline-responsive promoter (ptet) upstream of the Xist gene on the B6 X chromosome were differentiated into NPCs without Dox. Single clones carrying the inactivated B6 allele were picked and expanded, and Xist RNA levels were increased by adding Dox to the culture media. **c**, FISH for Xist RNA (green) in NPC clone E6 in the untreated condition (control) and after 3 days of Dox treatment (Dox (3 d)). DNA is stained with DAPI (blue). **d**, RNA-seq data showing the fold change in Xist expression (normalized CPM) compared with untreated cells across the time course of Dox treatment. Data relative to the mean of measurements in clone E6 are shown. Individual biological replicates are shown for each timepoint (control, Dox (3 d) and Dox (7 d)): n = 3, Dox (14 d) and Dox (21 d): n = 2 biological replicates. **e**, Schematic of the mouse X chromosome and heat map showing X-linked transcript allelic ratios in untreated clone E6 and after 3, 7, 14 and 21 days of Dox treatment. Allelic ratio indicates the fraction of reads from the Xi compared with reads from the Xi and Xa (ratio, 1: Xi monoallelic expression; ratio, 0: Xa monoallelic expression; ratio, 0.5: biallelic expression; ratio, >0.1: escape). Gene groups are defined as contiguous groups of escapees within 100 kb of each other (Methods). **f**, Heat map showing the allelic ratio of 133 escapees identified in clone E6 and shown in **e**. Escapees are assigned to three different categories as shown in Extended Data Fig. 1h and described in the Methods. The escape category for each gene is indicated below the heat map together with the zoom-in of the gene groups shown in **e**. **g**, Box plot showing the changes in allelic ratios for the different escape categories across the time course of Dox treatment. Data of clone E6 are shown. All the differences between control and Dox-treated measurements for escapee sets are significant at Padj < 0.01 (Wilcoxon rank sum test, Benjamini–Hochberg adjusted). Box plots show the median, 25th and 75th percentiles as well as 1.5× the interquartile range. **h**, Schematic of the exponential decay models used to study gene silencing kinetics. Data can be described by a full silencing model (blue, allelic ratio approaches 0) or a residual escape model (green, allelic ratio approaches value >0.1). The steepness of the curve corresponds to the gene's silencing half-life. **i**, Beeswarm plot showing the distributions of silencing half-life fit to all escapees using the offset model and stratified by escape category. P values are calculated using Wilcoxon's rank sum test (not adjusted for multiple testing). The large dots and error bars depict the median and 25th and 75th percentiles. **j**, Silencing half-lives per gene group as shown in **f**. Large dots show the mean and whiskers show the standard deviation across genes in the group. **k**, Beeswarm plot showing the distributions of residual escape parameters fit to all escapees using the offset model and stratified by escape category. P values are computed using Wilcoxon's rank sum test (not adjusted for multiple testing). The large dots and error bars depict the median and 25th and 75th percentiles. Averaged data from individual biological replicates per timepoint (control, Dox (3 d) and Dox (7 d)): n = 3, Dox (14 d) and Dox (21 d): n = 2 biological replicates (**e–g** and **i–k**). Data from 133 genes (constitutive n = 12; facultative n = 84; NPC-specific n = 37 genes) (**g, h, j** and **k**).

(escape in >50% of the studies); (2) 'facultative' escapees (variable escape, in different contexts); and (3) 'NPC-specific' escapees (escape only seen in NPCs so far, behave like facultative escapees). Out of the 133 escapees identified in clone E6, 12 are constitutive, 84 are facultative and 37 are NPC-specific escapees (Supplementary Table 2). Within the category of NPC-specific genes, only seven genes were found to escape XCI in all NPC clones (Extended Data Fig. 1i and Supplementary Table 2).

This comparison also revealed that in contexts in which facultative and NPC-specific escapees are inactivated, in the large majority of the cases, they do so either more slowly or later compared with other X-linked genes (Supplementary Table 3). Characterizing escape variability across clones, in light of previous data (Supplementary Table 1), is key to determining if certain X-linked genes are sensitive to Xist RNA level changes across cell types, tissues or developmental contexts. We went on to test whether Xist overexpression directly leads

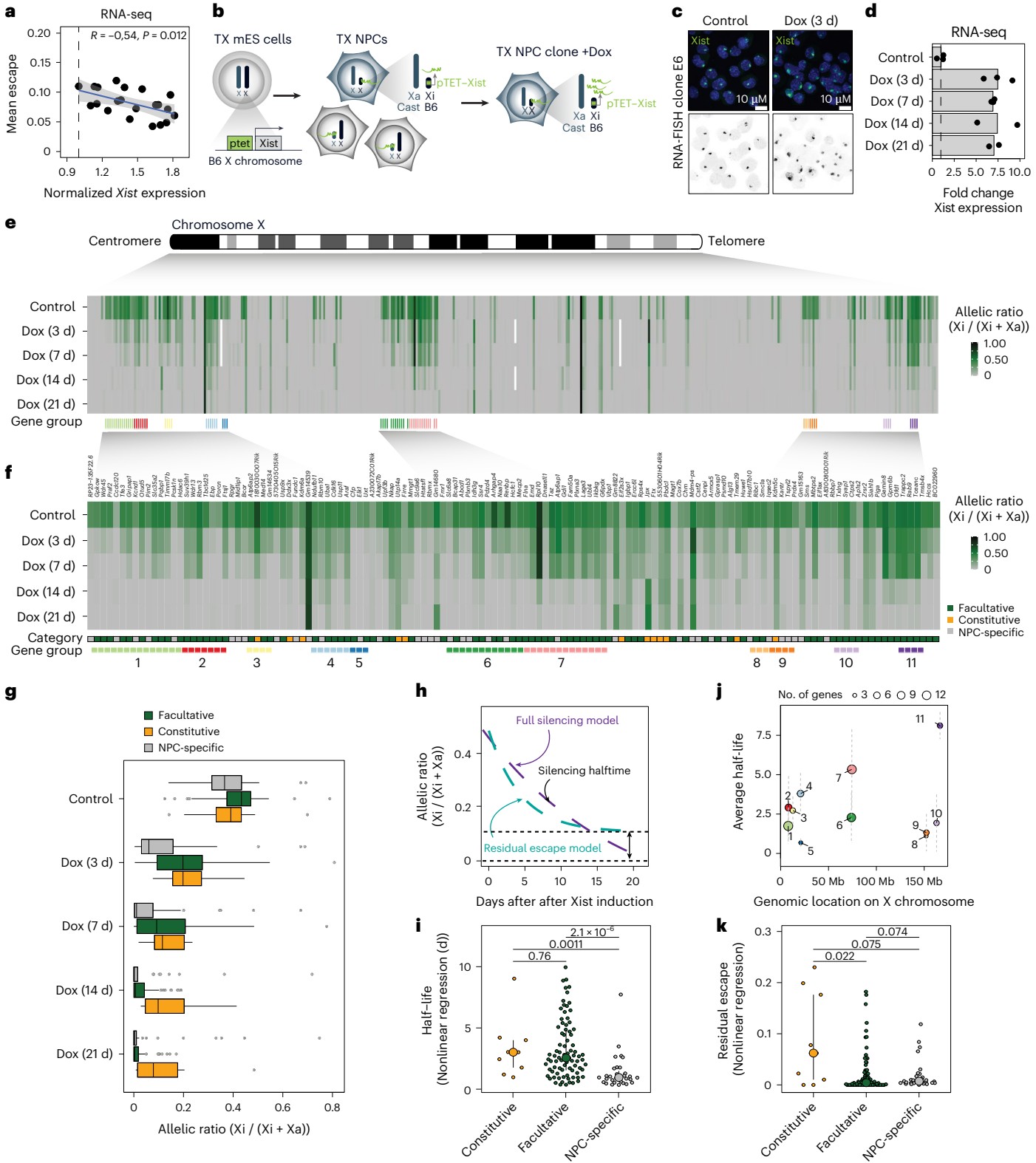

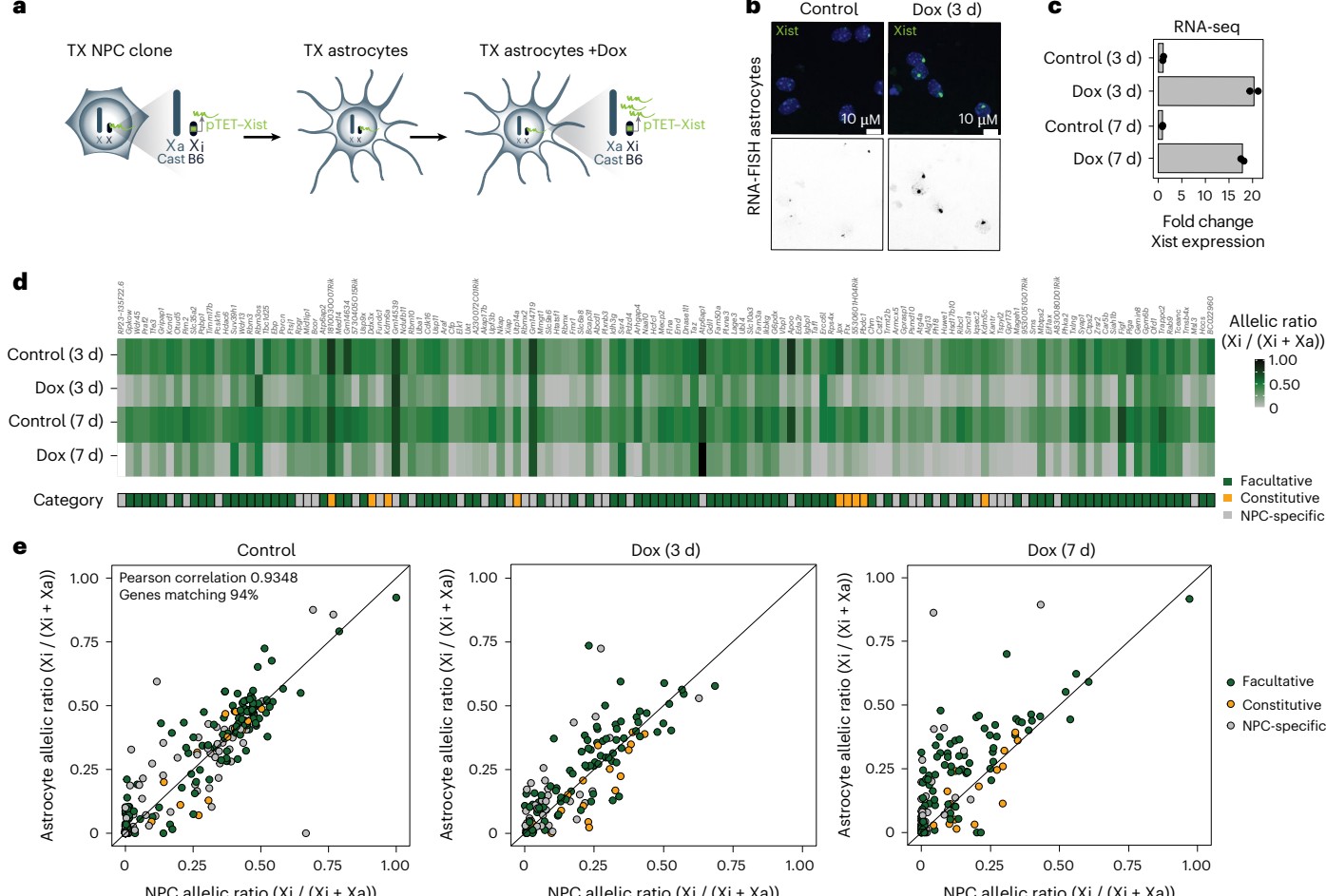

**Fig. 2 | Increased levels of Xist RNA silences Xi escapees in astrocytes.**
**a**, Experimental outline showing that TX NPCs (E6 clone) carrying a ptet promoter upstream of the *Xist* gene on the B6 X chromosome were differentiated into astrocytes without Dox. Xist RNA levels were increased by adding Dox to the astrocyte culture media. **b**, FISH for Xist RNA (green) in the untreated-astrocyte condition (control) and after 3 days of Dox treatment. DNA is stained with DAPI (blue). **c**, RNA-seq data showing the fold change in *Xist* expression (normalized CPM) compared with untreated cells across the time course of Dox treatment (17.5–21.1-fold enrichment). Two individual biological replicates are shown per condition. **d**, Heat map showing the mean allelic ratios between two replicates of escapees after 3 and 7 days of Dox treatment. Escapees are assigned to three different categories as shown in Extended Data Fig. 1g and described in the Methods. The escape category for each gene is indicated below the heat map. **e**, Scatterplot comparing the allelic ratios of individual escapees in astrocytes and NPCs (E6 clone) for the different Dox treatment conditions. The number of genes matching and Pearson correlation between the two cell types are indicated in the 'control' scatterplot.

to silencing of these escapees in different NPC lines (Fig. 1 and Extended Data Fig. 2).

*Xist* RNA levels increased by 6.9–7.5 fold following Dox induction for 3, 7, 14 and 21 days (Fig. 1c,d and Extended Data Fig. 2a–e). When escapee expression levels were assessed after 3 days of *Xist* induction, 78 out of 133 escapees showed silencing, with a significant reduction in allelic ratio of at least 50%, and a total of 89 genes were still expressed from the Xi (Extended Data Fig. 2i). By day 14 and 21 of Dox treatment, 25 and 21 escapees showed residual escape, respectively (albeit being expressed at a reduced level) (Fig. 1e,f and Extended Data Fig. 2i). Overall, during the time course, the expression levels of all the escapee categories were found to be consistently reduced, demonstrating the capacity of Xist RNA to initiate gene silencing in differentiated cells, beyond the early developmental time window previously defined for Xist action (Fig. 1e,f and Extended Data Fig. 2i). Similar results were obtained upon Xist upregulation up to 21 days in CL30 and CL31 clones (Extended Data Fig. 2d). These data demonstrate progressive Xist-mediated silencing over time in all tested NPC clones, even if a direct comparison of silencing dynamics between clone E6 and clones CL30 and CL31 is limited by the lower number

of escapees in CL30 (61) and CL31 (48) compared with clone E6 (133) (Fig. 1g, Extended Data Fig. 1i and Extended Data Fig. 2f,h). The reduced number of escapees in clones CL30 and CL31 is probably because they were originally differentiated in the presence of Dox from TX1072 ES cells (to skew XCI[25]), unlike clone E6. Thus, clones CL30 and CL31 had a prolonged exposure to elevated Xist levels during their derivation, consistent with our observation that the timing and level of Xist expression affect escape.

To characterize the differential silencing dynamics of escapees (Fig. 1f,g and Extended Data Fig. 2f), we quantified the speed at which escape was lost upon Xist overexpression (silencing half-life) and asked whether after prolonged Dox treatment certain genes retained some expression (residual escape) (Fig. 1h and Extended Data Fig. 3a,b). We found that for 31 genes, full silencing was not established even after persistent *Xist* overexpression, while the majority of genes become robustly silenced (Fig. 1h and Extended Data Fig. 2i). Escapees varied widely in silencing speeds, with constitutive escapees being silenced more slowly than NPC-specific genes and facultative escapees showing the widest range of silencing kinetics (Fig. 1i). Residual escape was more frequent among constitutive escapees than among facultative

and NPC-specific escapees (Fig. 1k and Extended Data Fig. 3a,b). Allele-specific expression analysis of X-linked genes showed that Xist overexpression led to significant downregulation of most escapees on the Xi, while Xa expression remained unchanged (Extended Data Fig. 4). This observation suggests the lack of compensatory mechanisms that control the overall dosage of escapees, at least at the mRNA level (Extended Data Fig. 4).

We also found changes in the expression levels of autosomal genes after 7 days of Dox treatment, particularly in clones CL30 and CL31 (that is, 60 and 35 upregulated genes and 49 and 43 downregulated genes, respectively) (Extended Data Fig. 3f–h). The fact that autosomal genes are not only downregulated but also upregulated, and that the affected genes are not clustered at specific autosomal loci (Extended Data Fig. 3i), suggests that this is probably a secondary, indirect effect of Xist upregulation, presumably due to the change in dosage of various X-linked escapee gene products, many of which can affect chromatin and transcription[69,70]. However, we cannot exclude the possibility of the direct action of Xist RNA in *trans* also accounting for some of the autosomal effects. Finally, we asked whether the expression levels of escapees or their position along the Xi explain the differences found in silencing upon Xist upregulation (Fig. 1j and Extended Data Fig. 3c–e). We found no correlation between escapee expression levels from either both the Xi and Xa or only the Xi and their efficiency in silencing upon Xist overexpression (Extended Data Fig. 3c,d). However, escapees pairs within 100 kb of each other showed more similar half-lives than randomly chosen pairs, implying that neighbouring escapees share similar responses to increased Xist levels (Fig. 1j and Extended Data Fig. 3e). This contrasts with the effects seen on autosomes where genes that show up- or downregulation are not in proximity to each other (Extended Data Fig. 3i). Along the Xi, we identified 11 gene groups that include 3 to 14 genes that share similar silencing speeds (Fig. 1e,f,j).

Thus, increased Xist RNA levels directly reduce the expression levels of X-linked escapees in post-XCI cells, leading to almost full loss of escape after 21 days of Dox induction.

## Xist RNA can initiate XCI of escapees in terminally differentiated astrocytes

To test whether *Xist* overexpression can also induce Xi-escapee silencing in non-dividing, post-mitotic cells, we differentiated NPC clone E6 into astrocytes[71]. Astrocyte verification involved staining for the astrocyte marker GFAP and the proliferation marker Ki-67 (Fig. 2a and Extended Data Fig. 5a). The quantification of Ki-67-positive cells in differentiated astrocytes confirmed that 99,3% of the cell population is non-dividing, whereas in E6 NPCs, 75% of cells are dividing before differentiation (Extended Data Fig. 5a). Transcriptomic analysis of astrocytes confirmed that the XCI status of X-linked genes remained remarkably stable upon differentiation from NPCs (Fig. 2e, Extended Data Fig. 5b,c and Supplementary Table 2). In fact, the overwhelming majority of escapees identified in NPCs are still transcriptionally active on the Xi in astrocytes (Fig. 2d,e and Extended Data Fig. 5b–d). Xist upregulation in astrocytes following 3 and 7 days of Dox was manifested both by RNA fluorescence in situ hybridization (RNA-FISH) and RNA-seq (Fig. 2b,c). The allelic ratios of the majority of escapees changed significantly following 3 and 7 days of *Xist* upregulation, demonstrating that Xist RNA can efficiently initiate gene silencing in post-mitotic cells (Fig. 2d and Extended Data Fig. 5e,f). To exclude that escapee silencing might be explained by a small number of dividing cells, we simulated mixtures of untreated astrocytes with defined fractions of Dox-treated NPCs (Extended Data Fig. 5g). Notably, while 3 days of Dox led to similar escapee silencing effects in NPCs and astrocytes, with 57% and 60% of genes showing a significant reduction in allelic ratio of at least 50%, respectively, 7 days of increased Xist levels did not lead to further silencing in astrocytes, unlike in NPCs (Fig. 2e, Extended Data Fig. 2i and Extended Data Fig. 5h). This may reflect different kinetics of gene silencing between dividing and non-dividing cells. Non-dividing cells may need longer exposure to Xist RNA levels to enable efficient escapee silencing. Alternatively, cell division may be a requirement to establish efficient gene silencing over time.

We also tested whether autosomal genes were affected upon *Xist* upregulation in astrocytes and found 466 up- and 412 downregulated genes after 7 days of Dox (Extended Data Fig. 3j). Similar to NPCs, these genes do not colocalize, again pointing to a probable indirect effect (Extended Data Fig. 3k). We found limited overlap between genes that are affected in NPCs and astrocytes, suggesting that the autosomal targets of Xist upon Xist upregulation are potentially different across different cell types (Extended Data Fig. 3l,m).

Altogether, these results demonstrate that the capacity of Xist to silence escapee genes is not restricted to multipotent, rapidly dividing stem cells but can also occur in non-dividing terminally differentiated cells.

## Xist RNA-mediated silencing of escapees in NPCs is SPEN dependent

Next, we investigated whether Xist-mediated silencing in post-XCI cells is SPEN-dependent, as during early development[25]. In clones CL30 and CL31,

**Fig. 3 | Xist-mediated silencing in NPCs is SPEN dependent. a**, Experimental outline showing that Xist RNA levels were increased in NPC clones carrying a SPEN–AID degron[2]. SPEN was depleted by adding auxin (indole-3-acetic acid) to the culture media for 2 days before inducing *Xist* upregulation with Dox for 3 or 7 days in the presence of auxin. **b**, Box plots showing the changes in allelic ratios of escapees across the time course of Dox and auxin treatment. Both CL30 and CL31: control and Dox (7 d): $n = 3$, others: $n = 2$ biological replicates. The data across 67 (CL30) and 50 (CL31) genes show a significant increase in allelic ratios between control and auxin treatment ($P$ value < 0.05) and a significant decrease in allelic ratios upon Dox (3 d) (CL30: adjusted $P$ value ($P_{adj}$) = $5.7 \times 10^{-28}$, CL31: $P_{adj} = 4.6 \times 10^{-17}$) and 7 d (CL30: $P_{adj} = 3.3 \times 10^{-37}$, CL31: $P_{adj} = 1.7 \times 10^{-21}$) compared with control but no significant differences between control and Dox-treated samples in the absence of SPEN (that is, +Dox, +Aux) after 3 d (CL30: $P_{adj} = 1$, CL31: $P_{adj} = 1$) and 7 d (CL30: $P_{adj} = 1$, CL31: $P_{adj} = 1$). The data also show significant changes in allelic ratios upon Dox treatment in the presence of SPEN compared with its absence (that is, +Dox, −/+Aux) after 3 d (CL30: $P_{adj} = 2.3 \times 10^{-21}$, CL31: $P_{adj} = 5 \times 10^{-13}$) and 7 d (CL30: $P_{adj} = 5.2 \times 10^{-25}$, CL31: $P_{adj} = 6 \times 10^{-15}$). $P$ values are calculated using Wilcoxon rank sum tests and adjusted using the Benjamini–Hochberg procedure. All box plots show the median, 25th and 75th percentiles as well as 1.5× the interquartile range. **c**, RNA-seq data showing the fold change in *Xist* expression compared with untreated cells following Dox treatment for 3 and 7 days and in combination with auxin treatment for 5 and 9 days, respectively.

**d**, Heat map showing allelic ratios of escapees (67 (CL30) and 50 (CL31)) upon *Xist* induction for 3 days (Dox (3 d)) and 7 days (Dox (7 d)) and in combination with auxin treatment (Dox (3 d), Aux (5 d) and Dox (7 d), Aux (9 d)). Data from NPC clones CL30 and CL31 are shown. The escape category for each gene is indicated below the heat map. The four genes *Gm14539*, *Gm8822*, *G6pdx* and *Mdm4-ps* show either an unchanged or increased (rather than decreased) allelic ratio upon Dox treatment. *Gm14539*, *Gm8822* and *Mdm4-ps* are pseudogenes with homologues on other chromosomes, and the annotation of SNPs at these loci is likely to be subjected to misannotations. The *G6pdx* locus in the CL30 and CL31 clones is tagged with a GFP/Tomato Hygro/Blasticidin resistance cassette, which may influence the transcriptional status of this particular gene on both alleles. The allelic changes observed for *G6pdx* are therefore difficult to interpret in these clones. **e**–**f**, Dot plots representing changes in allelic ratios upon Dox and auxin treatments for the 3-day (**e**) and 7-day timepoints (**f**). The average allelic ratios of 42 genes escaping in both CL30 and CL31 clones are shown. The diagonal dashed line represents no change compared with an untreated cell line. The error band depicts 95% confidence intervals. All box plots show the median, 25th and 75th percentiles and 1.5× the interquartile range. Averaged data from individual biological replicates per timepoint (control and Dox (7 d): $n = 3$, Aux (2 d), Dox (3 d), Dox (3 d), Aux (5 d) and Dox (7 d), Aux (9 d): $n = 2$ biological replicates) (**b** and **d**–**f**). Data from 67 (CL30) and 50 (CL31) genes (**b**, **d** and **e**).

both *Spen* alleles have an auxin-inducible degron (AID) domain, allowing its acute degradation upon the addition of auxin[25]. SPEN was depleted for 2 days before inducing *Xist* upregulation for 3 and 7 days (Fig. 3a). Depletion of SPEN in the absence of *Xist* upregulation led to significant upregulation of escapees, consistent with previous data[25] (Fig. 3b). We also confirmed that SPEN is dispensable for XCI maintenance, as we observed no reactivation of Xi silenced genes in its absence (Supplementary Fig. 1 and Supplementary Table 4). Upon combined Dox and Auxin treatment, *Xist* was efficiently upregulated (6.0–8.7-fold enrichment) (Fig. 3c). While the induction of increased Xist RNA levels for 3 and 7 days in the presence of SPEN leads to statistically significant silencing of escapees in both NPC clones (that is, −Dox versus +Dox comparison), when this is done in the absence of SPEN (that is, −Dox versus +Dox, +Aux comparison), no significant changes in the allelic ratios of

escapees were found (Fig. 3b,d–f and Supplementary Table 5). These data clearly demonstrate that the silencing of escapees by increased Xist RNA levels cannot occur in the absence of SPEN (Fig. 3d,e). Indeed, the overwhelming majority of escapees across the three categories were unaffected by *Xist* overexpression in the context of SPEN depletion (Fig. 3d,e). A few exceptions include *Ctps2*, *Ap1s2*, *Srpk3*, *Zdhhc9* and *G530011O06Rik*, which are dampened upon *Xist* upregulation even in the absence of SPEN in at least one clone or timepoint (Fig. 3d and Supplementary Table 5). This is consistent with previous observations showing that SPEN is dispensable for the silencing of a small subset of genes during XCI[25]. In summary, our data unexpectedly reveal that SPEN is essential for Xist-mediated silencing of nearly all escapees in differentiated cells, similarly to early embryonic development when XCI is initially established.

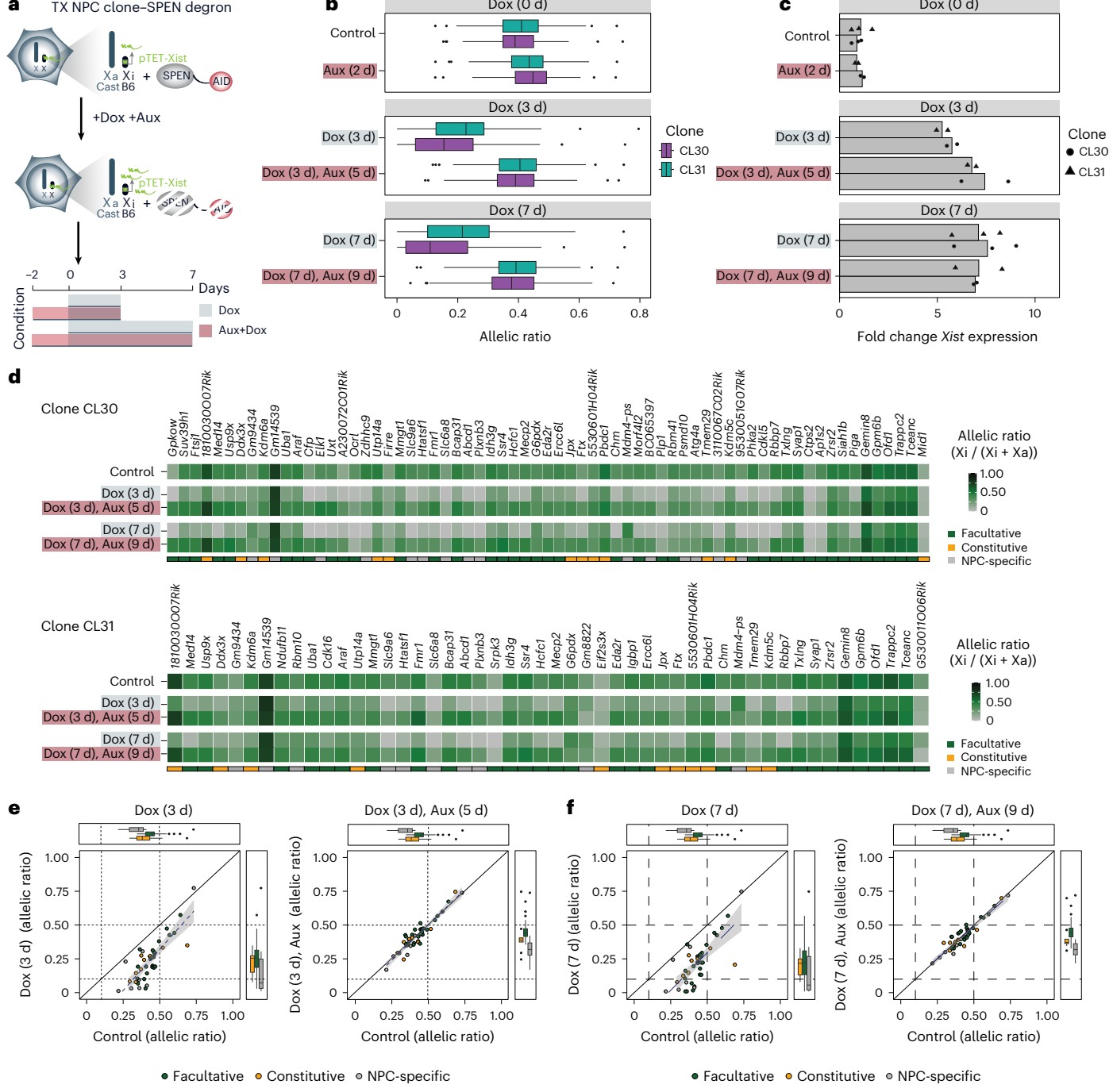

## Xist-mediated silencing of escapees leads to loss of TAD-like domains on the Xi

Facultative escapee clusters reside in TAD-like structures on the Xi, which is otherwise devoid of TADs[54,72]. Whether this is a cause or a consequence of escapee expression is unknown. We investigated whether escapee TAD-like structures are affected by increased *Xist* expression, with or without SPEN. We performed allele-specific Capture Hi-C analysis of >1-Mb genomic regions encompassing TAD-like domains previously observed on the Xi in NPCs[54] (Fig. 4a,b). One of these (the 'Mecp2–Hcfc1 cluster') spans ~800 kb including 24 facultative and 6 NPC-specific escapees (Fig. 1e,f). This is the most gene-dense region of the X chromosome, and escapees within this cluster show the highest variability in terms of facultative escape patterns between different NPC clones (Extended Data Fig. 1f). A second 'Kdm5c cluster' spans ~500 kb and includes one constitutive escapee, *Kdm5c*, as well as three facultative and four NPC-specific escapees (Fig. 1e,f).

As previously reported, TAD-like structures are oon the Xi, where topological organisation resembles that on the Xa (Fig. 4a,b). Following 7 days of *Xist* induction, the *Mecp2–Hcfc1* locus TAD-like structures on the Xi become attenuated, while the Xa remains unchanged (Fig. 4a and Extended Data Fig. 6a). Following 21 days of *Xist* upregulation, TAD-like structures are completely lost across the entire *Mecp2–Hcfc1* region, and all escapees are fully silenced (Fig. 4a). Again, no changes in three-dimensional (3D) structure were observed on the Xa (Extended Data Fig. 6b). Similar results were obtained at the *Kdm5c* cluster, where just 7 days of *Xist* upregulation resulted in efficient silencing of all three facultative and four NPC-specific escapees in the cluster as well as loss of TAD-like structures (Fig. 4b and Extended Data Fig. 6c). Only the constitutive escapee *Kdm5c* resisted complete inactivation, and a long-range looping interaction upstream of *Kdm5c* was maintained, probably representing long-range contacts with other expressed loci[13,54] (Fig. 4b). Thus, TAD-like structures at escapee clusters on the Xi may be the result of ongoing transcription and their silencing, or increased Xist levels, leads to their loss.

To assess whether loss of TAD-like structures occurred owing to increased Xist RNA levels, as has been previously postulated[13,30] or to escapee silencing, Capture Hi-C was performed in NPCs carrying the SPEN–AID degron (Fig. 5a–c). No differences in 3D topology on the Xa were observed upon Dox (*Xist* upregulation) and auxin (SPEN depletion) treatment (Extended Data Fig. 6d). *Xist* upregulation for 21 days in the context of acute SPEN depletion resulted in no loss in local 3D topology and no change in escapee transcription within the *Mecp2–Hcfc1* cluster, even though Xist RNA levels were tenfold higher in Dox-treated cells (Fig. 5c). Thus, increased Xist levels can only lead to loss of TAD-like structures at the *Mecp2–Hcfc1* cluster if SPEN is present. This suggests that Xist RNA alters Xi TAD-like structures only if gene silencing is induced. We also looked at undifferentiated TX1072 ES cells, where induction of *Xist* does not induce complete gene silencing[68,73], and showed that Xist induction in the presence of SPEN does not lead to loss of TAD-like structures in regions where genes are not fully inactivated (Extended Data Fig. 7a,b), further demonstrating that loss of 3D structure on the Xi occurs owing to loss of gene expression.

## Silencing of most facultative escapees becomes Xist independent after prolonged *Xist* upregulation

Next, we tested whether the Xist-mediated silencing of escapees in NPCs is irreversible or strictly Xist dependent. In clone E6, Xist was induced for 7, 14 or 21 days, followed by 7 days of Dox washout and RNA-seq (Fig. 6a–d). After washout, Xist returned to baseline levels (Fig. 6b,c), but escapee silencing became increasingly irreversible with longer induction, showing distinct reactivation dynamics across escape categories (Fig. 6d–f and Supplementary Table 6).

After 7 days of *Xist* upregulation followed by 7 days of Dox washout, 95 escapees out of 133 become reactivated, 16 escapees are partially irreversible and 22 escapees are irreversibly silenced (Fig. 6f). All 12 constitutive escapees are fully reversible at this timepoint (Fig. 6f). After 14 days of *Xist* upregulation, 57 escapees are irreversibly silenced (Fig. 6f). The number of irreversibly silenced escapees remains mostly unchanged across all categories after 21 days of *Xist* induction. Notably, all constitutive escapees except *Kdm6a* and *Ddx3x* are still reactivated upon washout following 21 days of *Xist* upregulation. Thus, although all constitutive escapees are sensitive to Xist RNA levels, the majority of them remain resistant to complete XCI (Fig. 6d–f). By contrast, for irreversibly silenced genes, Xist RNA becomes dispensable after 2 weeks of upregulation (Fig. 6f). Longer washout of Dox after 14 days of Xist induction led to only a limited increase in the number of reversible genes compared with shorter washout times (7 days) (Extended Data Fig. 8a,b). We confirmed these results in clones CL30 and CL31 (Extended Data Fig. 8c–f).

We also assessed whether differences in reactivation reflect differences in silencing dynamics of escapees (Fig. 6g). Escapees that undergo fast silencing upon increased Xist levels are less prone to reactivate, whereas slowly silenced genes tend to retain the capacity to become reactivated, when *Xist* returns to basal expression levels (Fig. 6g).

Given the silencing reversibility of both *Kdm5c* and *Hcfc1–Mecp2* clusters (Fig. 6d), we tested whether their 3D TAD-like organization on the Xi also reappeared. After 7 days of *Xist* upregulation followed by Dox washout, the 3D organization of these loci is re-established across the genes that become re-expressed within the *Kdm5c* cluster (Fig. 4c and Extended Data Fig. 6c). Similar observations were made at the *Hcfc1–Mecp2* cluster, even if gene silencing and TAD loss were less pronounced after 7 days of Dox treatment (Extended Data Fig. 6e). However, following 21 days of Dox treatment and washout, TAD-like structures were not re-established in regions of the cluster that encompass irreversibly silenced genes (for example, most genes of group 6), but 3D structures were restored at the reversible genes of group 7 (Extended Data Fig. 6e). These 3D structures are less pronounced than those we observed after 7 days of Dox treatment and washout, reflecting the different number of escapees that are actively transcribed and their expression levels between the two conditions (Extended Data Fig. 6e). Thus, the emergence of TAD-like structures at escapee clusters seems to be a direct consequence of transcription.

In summary, we show that most escapees become irreversibly silenced after 14 days of increased Xist levels; constitutive escapees remain reversible even after prolonged (21 d) Xist upregulation, and 3D structures at escapee clusters are directly linked to gene transcription.

**Fig. 4 | Increased levels of Xist RNA leads to loss of TAD-like domains on the Xi. a–b**, Capture Hi-C interactions and insulation score at the *Mecp2–Hcfc1* (**a**) and *Kdm5c* (**b**) cluster in clone E6 before and upon Dox treatment. Capture Hi-C interactions are shown for the Xa and the Xi in the untreated condition (control; top) and upon Dox treatment for 7 days (middle) and 21 days (bottom) for the *Mecp2–Hcfc1* cluster and 7 days for the *Kdm5c* cluster. Capture Hi-C data are shown at 10 kb resolution. Heat maps show the allelic ratios for 29 (**a**) and 25 (**b**) X-linked genes included in the captured regions. Escapees belonging to groups 6 (green) and 7 (pink) within the *Mecp2–Hcfc1* cluster are highlighted. Differential map shows changes in genome topology between Dox-treated samples and control samples. **c**, Capture Hi-C interactions and insulation score at the *Kdm5c* cluster for the Xi in clone E6 after 7 days of Dox treatment and subsequent washout (4 days). Arrows indicate areas of increased interaction frequencies upon washout. Heat map shows the allelic ratios for 25 X-linked genes included in the captured regions. Differential maps show changes in genome topology between Dox-treated and washout samples.

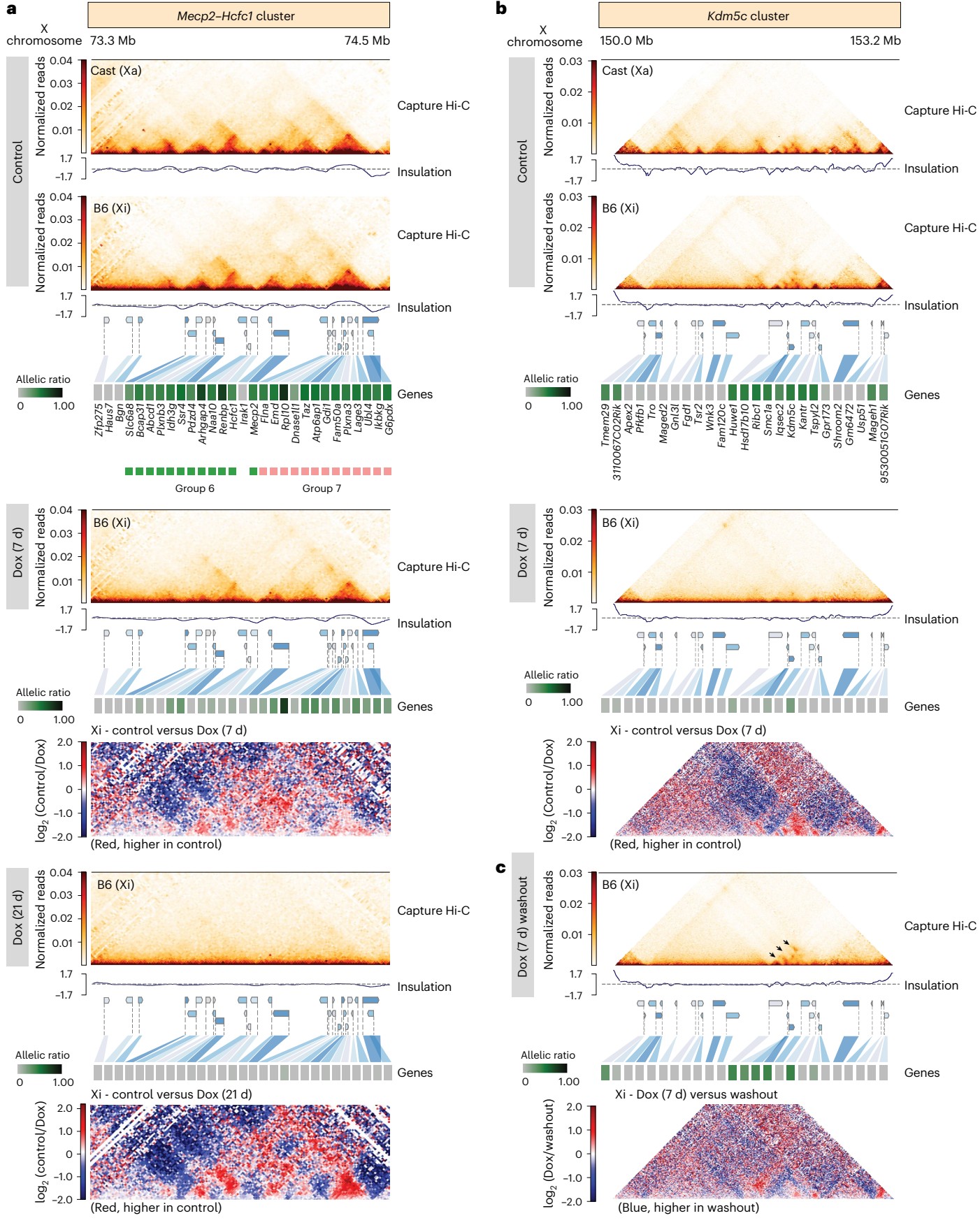

### Irreversible silencing of escapees is associated with DNA methylation at promoters and H3K27me3 enrichment

To define the epigenetic mechanisms underlying irreversible gene silencing following prolonged Xist upregulation, we assessed XCI maintenance marks: polycomb-deposited chromatin marks[32,33] and CpG island DNA methylation[34]. We performed CUT&RUN for H3K27me3 and H2AK119Ub following 7 and 21 days of Dox treatment and after Dox washout. Increased Xist RNA levels trigger enrichment of both chromatin marks X chromosome wide, whereas Dox washout leads to their overall decrease to basal levels (Fig. 7a and Extended Data Fig. 9a,b). After 7 and 21 days of *Xist* upregulation, H3K27me3 and H2AK119Ub are enriched on the Xi, both at repressed genes and escapees and including gene promoters, gene bodies and intergenic regions. Enrichment of both H3K27me3 and H2AK119Ub is largely reversible upon Dox washout (Fig. 7b,c and Extended Data Fig. 9c,d). Thus, variations in Xist RNA levels result in the reversible recruitment of the polycomb machinery to an already inactivated Xi in post-XCI cells.

We also assessed the reversible Xist-dependent enrichment of H3K27me3 and H2AK119Ub at autosomal loci (Extended Data Fig. 9e) to ascertain whether the autosomal genes that are misregulated upon *Xist* upregulation colocalize with regions that become Polycomb marked and found no correlation (Extended Data Fig. 9f).

We examined polycomb changes following Dox induction and washout at reversible and irreversible escapees on the Xi. After 7 and 21 days of Dox induction, H2AK119Ub is enriched at both reversible and irreversible escapees and it returns to levels close to basal enrichment upon washout at both timepoints (Fig. 7d–f and Extended Data Fig. 9g). By contrast, H3K27me3 is decreased at reversible escapees after 7 and 21 days of Dox induction and washout but is retained at irreversible genes albeit at lower levels (Fig. 7d–f). Although the difference in reduction between reversible and irreversible escapees remains after 21 days of Dox induction and washout, in this latter case, the H3K27me3 levels do not return to basal levels, also in the case of reversible escapees (Fig. 7d–f and Extended Data Fig. 9h). This difference in dynamics between PRC1- and PRC2-deposited chromatin marks probably reflects the mechanisms by which these complexes are recruited to the Xi by Xist RNA: directly via hnRNPK in the case of PRC1 and indirectly following H2AK119Ub enrichment in the case of PRC2[32,33].

We next asked whether irreversible escapee silencing is linked to DNA methylation, a hallmark of Xi genes[9]. Promoters and CpG islands of silent Xi genes are hypermethylated, while gene-body methylation marks active escapees[37,74]. Following 7 and 21 days of Xist upregulation and Dox washout, progressive CpG methylation gain was observed across the Xi (Fig. 7g), alongside reduced methylation at escapee gene bodies, indicating their transition towards Xi-like silencing (Extended Data Fig. 9i). Unlike polycomb marks, CpG methylation persisted after Dox washout, with irreversible genes showing higher promoter CpG methylation than reversible ones after 21 days (Fig. 7d,h).

In summary, prolonged Xist upregulation renders escapee silencing irreversible, with stable promoter hypermethylation and loss of gene-body methylation. Irreversibly silenced genes retain H3K27me3,

whereas H2AK119Ub follows Xist levels. Reversibly silenced escapees gain little promoter methylation and are fully re-expressed when Xist levels drop.

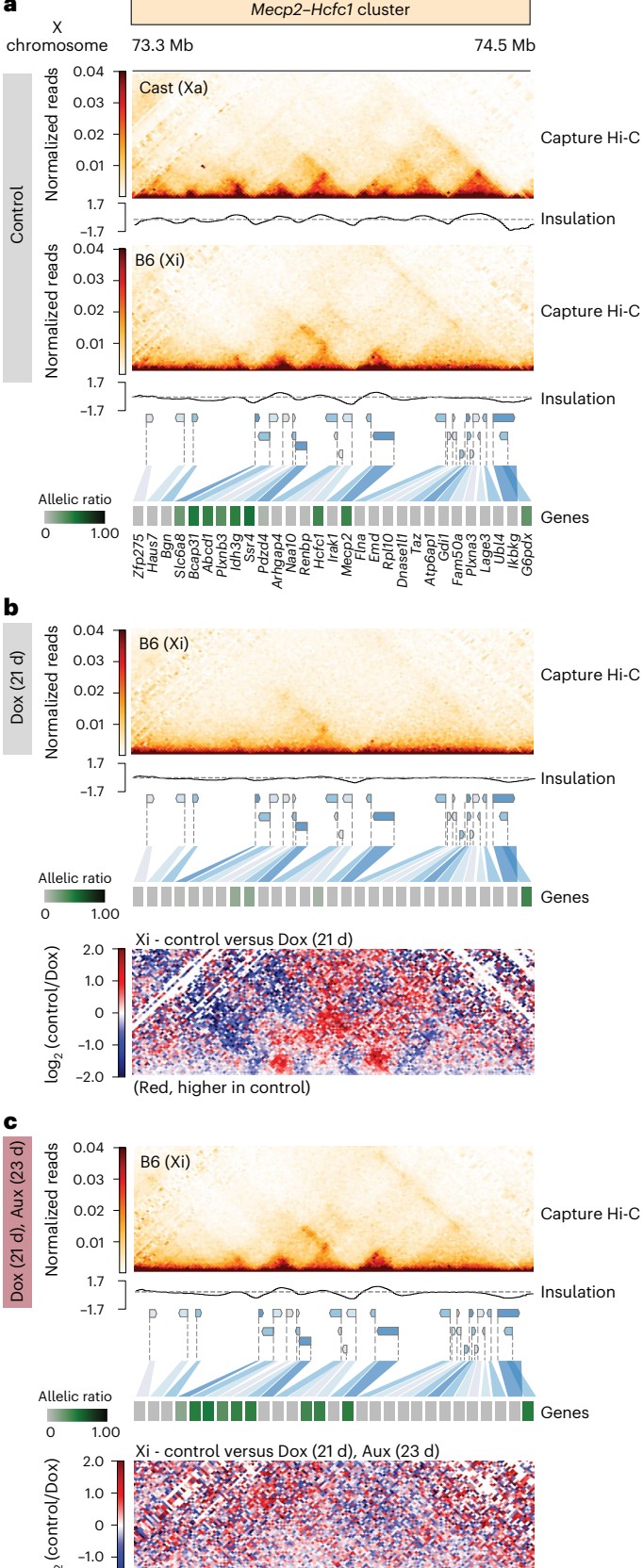

**Fig. 5 | Loss of TAD-like domains at escapee clusters occurs owing to gene silencing by Xist RNA. a–c**, Capture Hi-C interactions and insulation score at the *Mecp2–Hcfc1* cluster for the Xi in clone CL30.7 are shown. Capture Hi-C interactions are shown for the Xa and the Xi in the untreated condition (**a**), upon Dox treatment for 21 days (**b**) and after 23 days of auxin treatment and 21 days of Dox treatment (**c**). Capture Hi-C data are shown at 10 kb resolution. Heat maps show allelic ratios for 29 X-linked genes included in the captured region. Differential maps show changes in genome topology between 21-day Dox-treated samples and control samples (**b**) and 23-day auxin and 21-day Dox-treated samples compared with control samples (**c**).

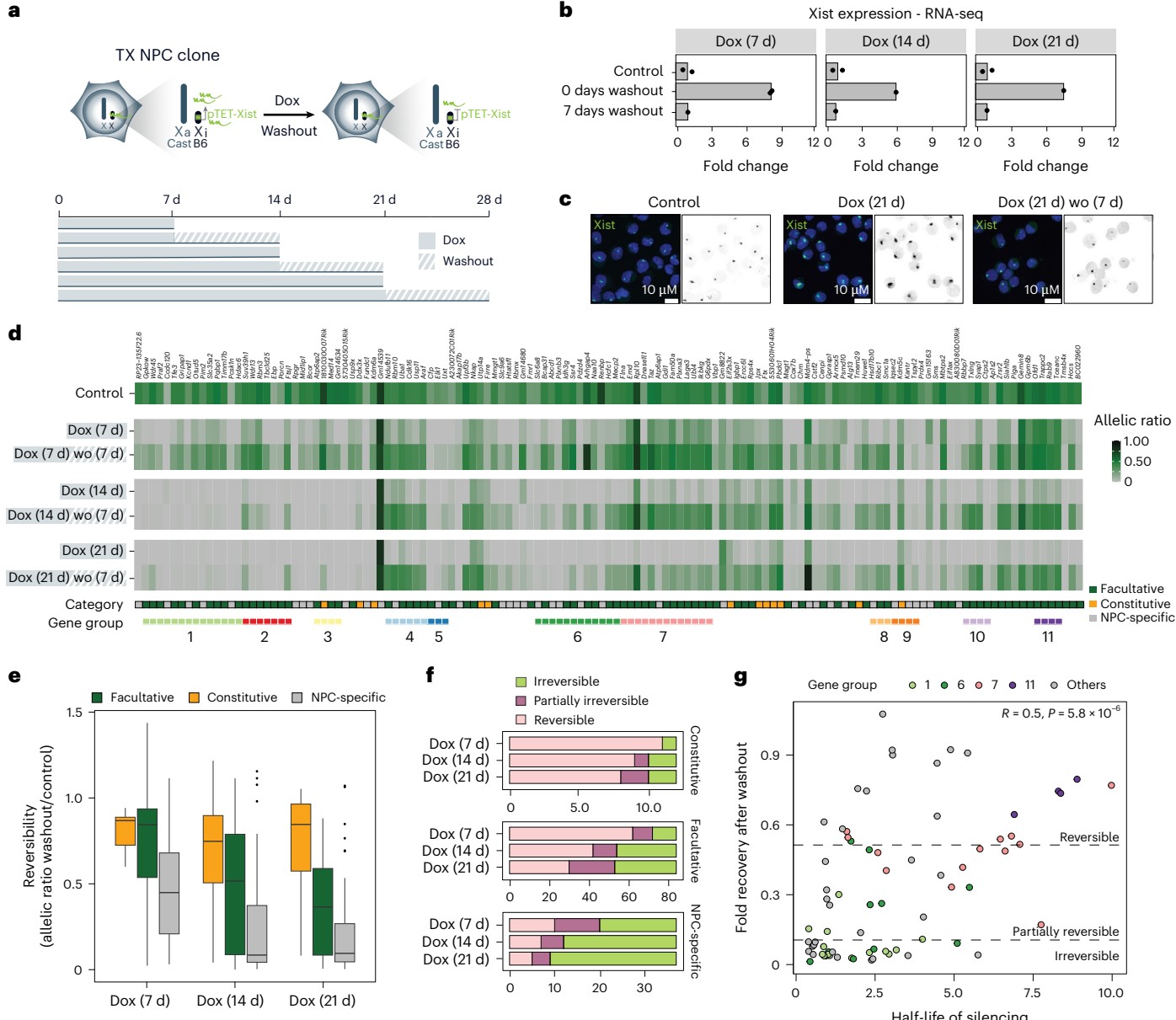

**Fig. 6 | Silencing of most facultative escapees becomes Xist independent after prolonged *Xist* upregulation. a**, Experimental outline showing *Xist* upregulation was induced by Dox treatment for 7, 14, and 21 days following 7 days of Dox washout. **b**, RNA-seq data showing fold changes in *Xist* expression (normalized CPM) compared with untreated control cells after 7, 14 and 21 days of Dox treatment, followed by 7 days of Dox washout. Data relative to the mean of measurements in clone E6 are shown. **c**, FISH for Xist RNA (green) in NPC clone E6 in the untreated condition (control), after 21 days of Dox treatment (Dox (21 d)) and following 7 days of Dox washout (Dox (21 d)–washout (7 d)). DNA is stained with DAPI (blue). **d**, Heat map showing allelic ratios of 133 escapees identified in clone E6 upon Dox treatment for 7, 14, and 21 days and following 7 days of Dox washout after these timepoints. Escapees are assigned to three different categories as shown in Extended Data Fig. 1h and described in the Methods. Gene groups are defined as contiguous groups of escapees within 100 kb of each other as in Fig. 1f. **e**, Box plots quantifying the reversibility of escape per gene (constitutive, 12; facultative, 84 and NPC-specific, 37 genes) for each timepoint by computing the ratio of the allelic ratio after washout divided by the allelic ratio in untreated samples. Differences between Dox (7 d) and Dox (14 d) and between

Dox (7 d) and Dox (21 d) are significant for the facultative and NPC categories (Dox (14 d): facultative $P = 1.46 \times 10^{-6}$, NPC-specific $P = 0{,}00608$; Dox (21 d): facultative $P = 2.08 \times 10^{-12}$, NPC-specific $P = 0{,}00421$). $P$ values are calculated using a paired Wilcoxon's rank sum test and adjusted using the Benjamini–Hochberg procedure. All box plots show the median, 25th and 75th percentile as well as 1.5× the interquartile range. **f**, Stack histogrammes showing the classification of genes into irreversible, partially irreversible and fully reversible escapees across the silencing time course. Genes are considered reversible when they reach at least 50% of untreated escape as well as an allelic ratio >0.1 after washout. Similarly, they are considered partially irreversible when they reach 10–50% of untreated escape and an allelic ratio >0.1, and they are considered irreversible otherwise. **g**, Scatterplot comparing silencing half-lives (Fig. 1) to fold recovery after washout for the gene groups (74 genes in total) indicated in **d**. Genes in a subset of local gene groups are highlighted to show their coordinated behaviour. Reversible, partially irreversible and irreversible fold recovery thresholds are indicated. $R$ indicates Pearson's correlation and the $P$ value is given by a correlation test.

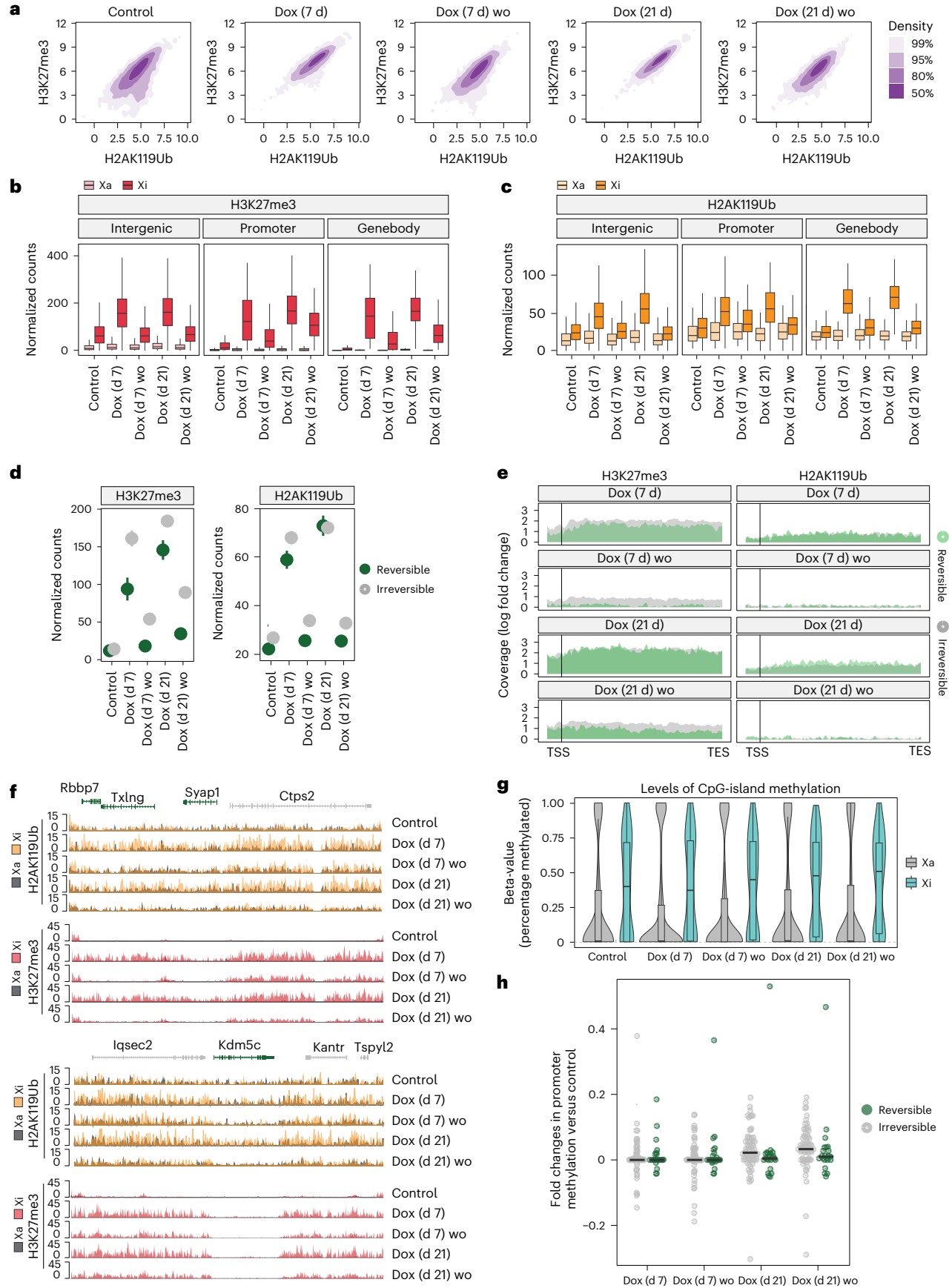

**Fig. 7 | Irreversible silencing of escapees is associated with DNA methylation at promoters and H3K27me3 enrichment. a**, Density plots showing the correlation between H3K27me3 and H2AK119Ub enrichments across 10 kb windows with more than ten average read counts spanning the Xi during the Dox treatment time course (control, 7 days (Dox (7 d)), 21 days (Dox (21 d))) and washout (for 7 days after each treatment (Dox (7 d) wo, Dox (21 d) wo)). The axes represent normalized log counts. **b**, Box plot comparing H3K27me3 enrichment (normalized counts) across intergenic regions, as well as promoters and gene bodies of escapees during Dox treatment and washout. H3K27me3 levels are shown separately for the Xa (light red) and Xi (dark red) ($n = 6,514$ intergenic, 107 promoter, 126 genebody genomic intervals in all conditions). **c**, The same as **b** but for for H2AK119Ub. H2AK119Ub levels are shown separately for the Xa (yellow) and Xi (orange) ($n = 6,514$ intergenic, 107 promoter, 126 genebody genomic intervals). **d**, Dot plot comparing the gene-length normalized H3K27me3 (left) and H2AK119Ub (right) levels over gene bodies of reversibly (green) and irreversibly (grey) silenced escapees during Dox treatment and washout. Normalized counts are shown ($n = 86$ irreversible, 30 reversible genes). The dot depicts the mean, and the error bars depict standard errors. **e**, Aggregate plots of H3K27me3 (left) and H2AK119Ub (right) coverage over the promoter region (1 kb upstream to 500 bp downstream of the transcription start site (TSS)) and gene bodies (scaled to gene length, until the transcription end site (TES)) of reversibly (green) and irreversibly (grey) silenced escapees during Dox treatment and washout. The $\log_2$ fold change in coverage is calculated to

the untreated condition (control). **f**, Genome tracks showing H2AK119Ub and H3K27me3 accumulation and loss on the Xa (grey) and Xi (H3K27me3 (pink) and H2AK119Ub (yellow)) during Dox treatment and washout. Two example regions containing both reversibly (green) and irreversibly (grey) silenced escapees are shown. **g**, Violin plots showing allele-specific DNA methylation levels of CpG islands during Dox treatment and washout. For each CpG island, the average fraction of methylated cytosines is calculated across CpG sites from haplotype-informative reads. The Xa (grey) and Xi (light blue) chromosomes are shown separately. Differences observed for Xi between control and treated conditions are significant for Dox (21 d) ($P = 0.0155$) and Dox (21 d) wo ($P = 0.00312$). $P$ values are calculated over $n = 42$ CpG islands using a paired Wilcoxon's rank sum test. **h**, Beeswarm plot showing the fold change in promoter methylation compared with the untreated condition (control) during Dox treatment and washout for reversible (green) and irreversible (grey) escapees. Promoters are defined as regions of accessible chromatin as measured by ATAC-seq that overlap (−500 bp, 1,000 bp) intervals around annotated TSSs. The median is indicated with a black bar. Differences between reversible and irreversible genes are significant for Dox (21 d), $n = 63$ irreversible, 16 reversible promoters ($P = 0.0194$) and Dox (21 d) wo ($P = 0.0456$). $P$ values are calculated using a paired Wilcoxon's rank sum test. All box plots show the median, 25th and 75th percentiles as well as $1.5\times$ the interquartile range. All figures show average data across two biological replicates.

## Xist RNA levels modulate X-linked escapee gene expression in vivo

Finally, we assessed whether increased Xist RNA levels similarly impact escapees during mouse embryonic development. The role of Xist RNA and the kinetics of gene silencing and escape have been shown to be very similar between iXCI and rXCI[22]. Thus, iXCI was used to extend our findings from in vitro differentiated cells to an in vivo setting.

The same mice from which TX1072 mES cells[56] were derived were used, with a Dox-inducible endogenous *Xist* allele (TX) and a tetracycline-responsive transactivator (rtTA) integrated at the ubiquitously expressed *Rosa26* locus[57].

We first examined how increased Xist RNA impacts escape from iXCI in late pre-implantation embryos. TX/Y males (Xptet/Y; R26rtTA/WT, where WT is wild type) were crossed with JF1 females to obtain F1 hybrids (Fig. 8a). RNA-seq was performed after Dox-induced Xist overexpression at E3.5 and E4.5, when XCI and escape are already established on the paternal Xp[19]. F1 embryos were collected at E2.5 and E3.5, cultured 24 h with Dox and analysed using single-embryo RNA-seq

(Fig. 8a). Female embryos carrying rtTA showed Xist induction, while rtTA-negative siblings served as controls (Fig. 8a). RNA-FISH confirmed larger, more intense Xist clouds in rtTA+ blastomeres (Fig. 8b and Extended Data Fig. 10a), although whole-embryo RNA-seq did not reveal major Xist upregulation at E3.5 or E4.5 (Extended Data Fig. 10b), consistent with blastomere variability[75,76]. Nonetheless, rtTA+ embryos showed reduced X-linked allelic ratios compared with controls at E4.5, indicating enhanced Xi silencing (Extended Data Fig. 10c,d).

To address the impact of *Xist* upregulation on escapee expression from the Xi, we focused on E4.5 embryos. We found 251 inactivated Xp genes out of 381 informative genes and 131 escapees (Extended Data Fig. 10e,f). All constitutive escapees identified in embryos and 74 of the facultative genes were also detected in NPCs. Of the 31 NPC-specific escapees described earlier, only 14 escaped XCI in embryos as well, confirming their facultative nature, and we therefore classified them in the facultative gene category. We identified a further 24 E4.5 embryo-specific escapees (Extended Data Fig. 10f and Supplementary Table 7). Following

**Fig. 8 | Xist levels modulate X-linked dosage in vivo. a**, Experimental outline showing male TX B6 mice ($X^{ptet}Y$; $R26^{rtTA/WT}$) were crossed with WT JF1 females. F1 embryos were collected at E2.5 and E3.5 and cultured for 24 h while adding Dox to the culture media. RNA-seq was performed at E3.5 and E4.5. **b**, RNA-FISH for Xist RNA (green) in E4.5 XX embryos obtained by crossing male TX B6 mice ($X^{ptet}Y$; $R26^{rtTA/rtTA}$) with WT JF1 females. DNA is stained with DAPI (blue). **c**, Box plots showing mean allelic ratios for −rtTA and +rtTA embryos for the different escapee categories. The constitutive, facultative (includes genes annotated as facultative in Extended Data Fig. 1h and NPC-specific escapees) and E4.5-specific categories contain 10, 97 and 24 escapee genes, respectively. Biological replicates: for E3.5, −rtTA $n = 2$ embryos, +rtTA $n = 4$ embryos; for E4.5, −rtTA $n = 17$ embryos, +rtTA $n = 20$ embryos. The adjusted $P$ values for the comparison between −rtTA and +rtTA embryos are $P_{adj} = 3.9 \times 10^{-2}$ for the constitutive category, $P_{adj} = 6.60 \times 10^{-16}$ for facultative and $P_{adj} = 8.2 \times 10^{-5}$ for E4.5-specific escapees (two-sided Wilcoxon rank sum test, adjusted using the Benjamini–Hochberg procedure). **d**, Experimental outline shows TX/Y males carrying the rtTA transactivator ($X^{ptet}/Y$; $R26^{rtTA/rtTA}$ or $X^{ptet}/Y$;$R26^{rtTA/WT}$) were crossed with WT JF1 females. Xist RNA overexpression was induced by adding Dox to the drinking water of pregnant females for 5 days, from E3.5 to E8.5. RNA-seq was performed using RNA extracted from ExE tissue. **e**, RNA-seq data showing fold changes in *Xist* expression (normalized CPM) for rtTA− +Dox (control) and rtTA+ +Dox (Dox). Fold changes are calculated to the mean Xist levels of rtTA+ −Dox samples. Each dot represents an individual embryo. Biological replicates: −rtTA, +Dox

$n = 2$ embryos; +rtTA, +Dox $n = 6$ embryos; +rtTA, −Dox $n = 8$ embryos. **f**, Plot showing average escape for each ExE sample, categorized by rtTA genotypes and Dox treatment. Biological replicates: −rtTA, +Dox $n = 2$ embryos; +rtTA, +Dox $n = 6$ embryos; +rtTA, −Dox $n = 8$ embryos. PCA was performed on the whole transcriptome excluding X chromosomes. **g**, Heat map showing the mean allelic ratios of 74 escapees in E8.5 ExE tissues across rtTA+ −Dox, rtTA+ +Dox and rtTA− +Dox conditions. Escapees are categorized as in Extended Data Fig. 1h, with NPC-specific genes included in the facultative category. Genes are additionally called 'E8.5-specific' if they show an allelic ratio >0.1 and <0.8 in more than 50% of rtTA+ −Dox ExE samples and do not escape in NPCs. The constitutive, facultative (includes genes annotated as facultative in Extended Data Fig. 1h and NPC-specific escapees) and E8.5-specific categories contain 10, 48 and 16 escapee genes, respectively. **h**, Box plots showing mean allelic ratios rtTA+ −Dox, rtTA+ +Dox and rtTA− +Dox ExE samples for the different escapee categories. The adjusted $P$ values for the comparison between rtTA+ −Dox and rtTA+ +Dox ExE samples are $P_{adj} = 7.8 \times 10^{-2}$ for the constitutive category, $P_{adj} = 5.6 \times 10^{-12}$ for facultative and $P_{adj} = 8.5 \times 10^{-4}$ for E8.5-specific escapees (two-sided Wilcoxon rank sum test, adjusted using the Benjamini–Hochberg procedure). All box plots show the median, 25th and 75th percentile and $1.5\times$ the interquartile range. **i**, Scatterplot showing the same data as in **h** but comparing allelic ratios of individual escapee genes between rtTA+ −Dox and rtTA+ +Dox. Biological replicates: −rtTA, +Dox $n = 2$; +rtTA, +Dox $n = 6$; +rtTA, −Dox $n = 8$ (**h** and **i**).

Dox-induced *Xist* upregulation, most escapees become downregulated on the Xi at E4.5 (Extended Data Fig. 10f–h). Similar to NPCs, constitutive escapees were less sensitive to *Xist* upregulation (Fig. 8c and Extended Data Fig. 10g). Interestingly, E4.5-specific escapees show variable responses to *Xist* upregulation, with most genes showing silencing but a subset being unaffected (Fig. 8c and Extended Data Fig. 10g). Closer examination of the latter revealed that two of these, *Rho5x* and *Fthl17f* (also known as *Gm5635*), are imprinted and only expressed from the Xp[77,78] at this stage of development[19].

To assess the effects of Xist upregulation for longer times and later in vivo, we induced Xist RNA from E3.5 to E8.5 (Fig. 8d). Male TX/Y mice with rtTA ($X^{ptet}$/Y; R26$^{rtTA/rtTA}$ or $X^{ptet}$/Y;R26$^{rtTA/WT}$) were crossed with WT JF1 females, and Dox was added to the drinking water of pregnant females for 5 days starting at E3.5. RNA-seq was performed at E8.5 focusing on extraembryonic (ExE) tissues where the Xp is always the inactive X. Increased Xist RNA levels were observed

in rtTA+ E8.5 ExE tissue (up to 17.9 fold) but not in rtTA− controls (Fig. 8e, Extended Data Fig. 10i and Supplementary Table 7). Xist upregulation caused a consistent reduction in X-linked allelic ratios in all rtTA+ tissues (Fig. 8f and Extended Data Fig. 10j,k). At E8.5, we identified 74 escapees, including 9 ExE-specific ones (Fig. 8g). Following Xist upregulation, all categories of escapees were significantly downregulated, with only 11 genes still escaping XCI, including *Ogt* (previously shown to escape iXCI[18,19]), as well as *Taf1* and *Rpsx4* (shown to escape in mid-gestation placenta[59]) (Fig. 8h,i). Thus, increased Xist levels from E3.5 to E8.5 silence most facultative and tissue-specific escapees, while most constitutive escapees still resist complete inactivation (Fig. 8h,i).

In summary, our in vivo experiments, focused on cell and tissue contexts with iXCI, clearly demonstrate that Xist RNA is a direct regulator of escape in multiple contexts well after the initiation of X-chromosome-wide inactivation.

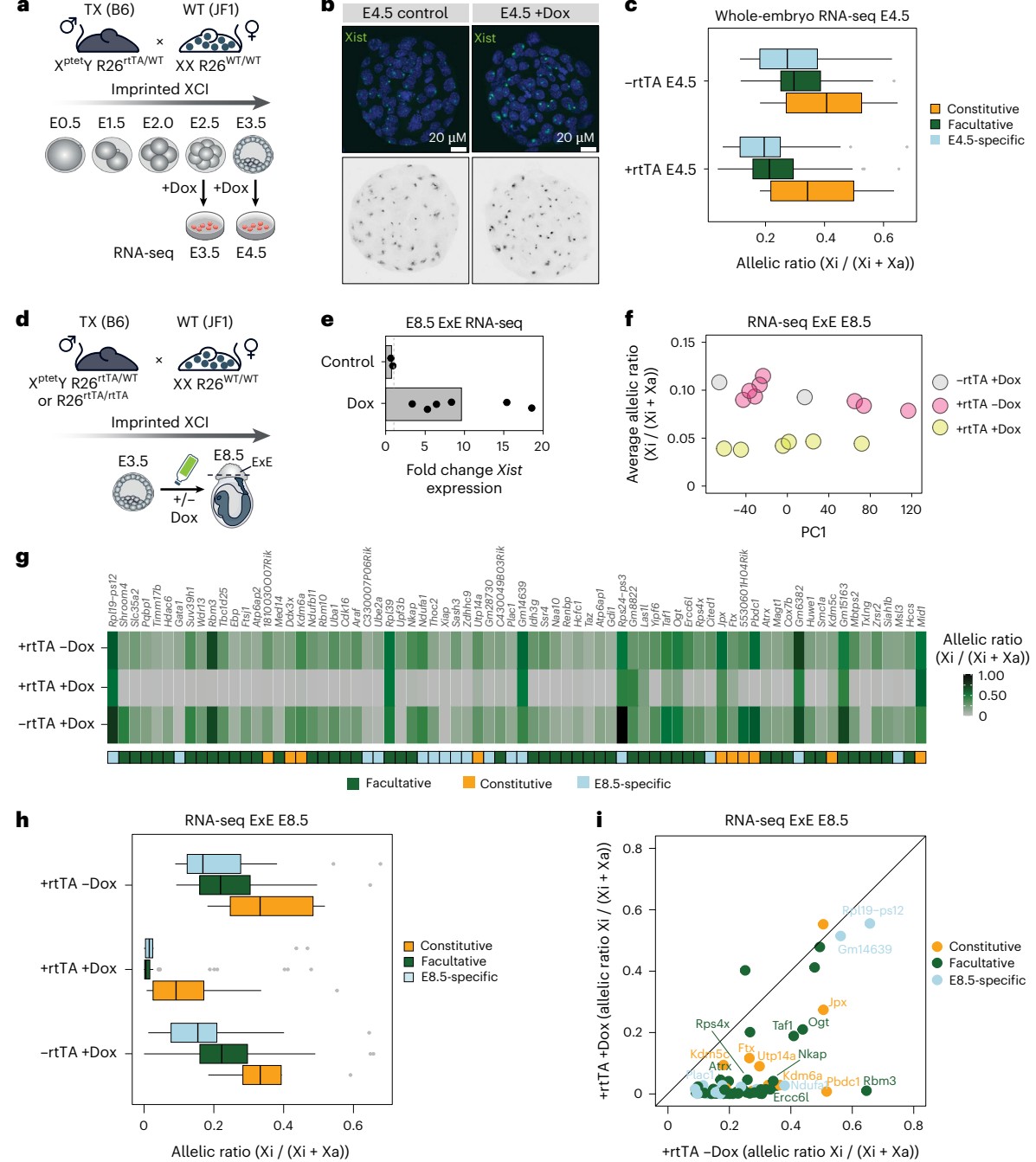

## Discussion

Here, we demonstrate that Xist RNA can directly modulate X-linked escapee expression in dividing NPCs and non-dividing astrocytes as well as in pre- and post-implantation embryos. The silencing action that Xist RNA exerts on escapees is largely SPEN dependent and occurs well outside of the early differentiation time window that XCI was thought to be restricted to. In light of these findings, natural fluctuation in Xist/XIST levels driven by genetic variation at the locus or its regulators can strongly affect X-linked escapee gene expression across and within individuals. Even small increases in X-linked escapee expression, as shown in humans and mice, may contribute to cancer development[45,49].

In human brain transcriptomic data, we found that higher XIST levels correlate with lower escapee expression. This is consistent with a recent study showing that XIST RNA levels are heterogeneous across single cells in the human brain, and lower levels of Xi-associated XIST correlate with larger sex differences in gene expression, an indicator of Xi transcriptional activity[79]. A similar approach showed that XIST RNA levels vary across breast cancer subtypes, with those expressing the highest XIST levels being characterized by lower levels of breast cancer-specific X-linked escape[45].

We also found evidence for misregulation of autosomal gene expression upon *Xist* overexpression, which might indicate spreading of Xist RNA on autosomal loci[80]. However, autosomal genes were both up- and downregulated without locus-specific clustering, suggesting indirect effects of Xist upregulation via altered escapee dosage, many of which are transcription and chromatin regulators.

A recent study showed that XIST can re-silence genes that had been reactivated on the Xi in B cells, if XIST was first switched off and then back on[46]. Furthermore, ectopic induction of a human XIST complementary DNA (cDNA) from an autosome in cancer cells led to repression of a reporter gene[81], and the overexpression of XIST from chromosome 21 in neural stem cells was found to silence autosomal genes[82]. However, the integration of XIST transgenes onto autosomes does not explore the role of endogenous XIST in modulating X-linked gene dosage and Xi transcriptional activity.

Our work directly addresses the fundamental question of the role of Xist in regulating escapee expression by increasing Xist levels on an already inactive X chromosome in post-XCI cellular and developmental contexts. Escapees that have either lost their repressed state (facultative) or that never acquired it (constitutive) during differentiation are still fully susceptible to Xist–SPEN-mediated transcriptional repression in differentiated cells. Irreversible escapee silencing is accompanied by higher promoter DNA methylation. H2AK119Ub is reversibly recruited to the Xi, whereas H3K27me3 is retained at irreversibly silenced escapees, suggesting that PRC2 may participate in maintaining silencing independently of Xist. However, given the known kinetic differences between these two chromatin marks[83], we cannot exclude that a longer Dox washout would have resulted in loss of H3K27me3.

Our study also shed light on the interplay between the 3D topology of Xi escapee regions and their transcriptional activity. We demonstrate that TAD-like domains encompassing Xi escapee clusters[54] are eliminated by gene repression and reappear upon re-establishment of transcription. In NPCs, this reversibility of 3D structure at the '*Mecp2–Hcfc1*' cluster is not simply a consequence of higher levels of Xist RNA but strictly depends on SPEN-mediated gene silencing. The lack of structural changes upon *Xist* upregulation in ES cells suggests that the recruitment of SPEN alone is not sufficient to change the 3D structure of the Xi upstream of gene silencing, at least in this cellular context. In this study, we focused on two clusters, but we expect these effects on 3D topology to occur at all clusters of escapees on the Xi. Furthermore, although increased levels of Xist RNA and ensuing escapee silencing can overrule TAD-like structures on the Xi in post-XCI cells, such structures might still facilitate the onset of facultative escape during development.

The sensitivity and reversibility of facultative and constitutive escapees to Xist RNA silencing is strikingly different. Constitutive escapees are less susceptible and maintain the capacity to resist full XCI. These genes are involved in fundamental functions, including chromatin regulation, protein translation and ubiquitination[84,85], and are believed to be highly dosage-sensitive, as they have retained a Y-chromosome homologue during sex chromosome evolution[84]. Thus, they presumably need to be protected from complete Xist-mediated silencing not only during development, when XCI is first initiated, but also in adult tissues, as any increase in Xist RNA levels could potentially lead to their inactivation with deleterious effects.

By contrast, facultative escape is highly variable and unlikely to have the same evolutionary pressures. Whether facultative escape reflects inefficient XCI maintenance or is actually purposeful is still not fully understood. As prolonged high levels of Xist RNA lead to irreversible inactivation of these genes, we speculate that the capacity of facultative genes to become reactivated and to escape XCI in some contexts may be determined by the levels of Xist RNA to which they were exposed at the onset of XCI. During embryonic development and ES cell differentiation, XCI is not perfectly synchronous[18] and different cells will express different levels of Xist RNA[23], potentially setting a threshold for facultative escapees to remain susceptible to varying Xist RNA levels in adult somatic cells. Accordingly, the modulation of their expression levels following changes in Xist RNA expression may underlie the plasticity of these X-linked genes to allow for fine-tuning gene dosage regulation in different conditions. Our study uncovers a role for endogenous Xist RNA levels in tuning escapee dosage and lays the foundation for exploring its impact on disease, identifying biomarkers and developing X-linked gene dosage therapies.

## Online content

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

## Methods

### Cell culture

mES cells TX1072 (*Mus musculus castaneus* (Cast/EiJ) × *Mus musculus domesticus* (C57BL/6)) have been previously derived in the laboratory[56]. Cells were cultured on a gelatine-coated (0.1% gelatine in 1× PBS) cell culture dish in 2i-containing ES cell media (DMEM, 15% fetal bovine serum, 0.1 mM β-mercaptoethanol, 1,000 U mL$^{-1}$ leukaemia inhibitory factor, CHIR99021 (3 µM) and PD0325901 (1 µM)).

### NPC differentiation

mES cells TX1072 were differentiated and sub-cloned as previously described[53]. In brief, $1 \times 10^6$ ES cells were seeded in a gelatin-coated 10-cm petri dish in N2B27 media (DMEM/F12:Neurobasal (1:1), L-glutamine, 0.1 mM 2-mercaptoethanol). At day 7 of differentiation, $3 \times 10^6$ cells were plated in N2B27 media supplemented with epidermal growth factor (EGF) and fibroblast growth factor (FGF) (10 ng ml$^{-1}$ each) in bacterial Petri dishes to prevent cells from attaching to the plates. After 3 days of cell growth in suspension as cellular aggregates (or spheres), the aggregates were harvested and plated onto gelatine-coated 10-cm Petri dishes in N2B27 media supplemented with EGF and FGF. Monolayer NPCs grew out of the attached sphere. To generate the E6 NPC clone used in this study, 5,000–10,000 single cells were plated in 10-cm Petri dishes and single clones were manually picked after 15–20 days. Single NPC clones were expanded and characterized by RNA-FISH to assess karyotype stability. In case of unstable karyotype resulting in gain or loss of chromosomes, NPC clones were either discarded or further sub-cloned. Clone E6 was regularly tested for karyotype stability by RNA-FISH at the end point of each experiment, before sequencing. The NPC clones CL30 and CL31 carrying endogenous SPEN alleles tagged with AID–Halo have been previously generated in the laboratory[25]. Each line was sub-cloned to obtain karyotypically stable clones named CL30.7 and CL31.16. NPCs were cultured on a gelatine-coated (0.1% gelatine in 1× PBS) cell culture dish in NPC media (N2B27 media supplemented with with FGF2 (10 ng ml$^{-1}$) and EGF (10 ng ml$^{-1}$), both from PeproTech).

### Astrocyte differentiation

NPC clone E6 was used for astrocyte differentiation. Tissue culture plates or coverslips (for RNA-FISH and immunostaining) were coated with Poly-D-Lysine overnight and washed three times in water before laminin coating for at least 2 h. NPCs were seeded in astrocyte differentiation media (N2B27 medium supplemented with 20 ng ml$^{-1}$ bone morphogenetic protein (BMP4, R&D Systems)). Cells were differentiated for 3 days before starting Dox treatment in terminally differentiated astrocytes.

### Cell treatments

*Xist* expression in TX1072 was induced by addition of Dox (1 µg ml$^{-1}$) to the NPC media or astrocyte media. Culture media supplemented with Dox were renewed every 24 h. For Dox washout, Dox-containing media were removed and cells were refreshed with Dox-free culture media. Auxin-mediated depletion of SPEN was achieved by supplementing the culture media with Auxin (Sigma) at the concentration of 500 µM, as previously described[25]. Auxin-containing media were renewed every 24 h.

### RNA-FISH

RNA-FISH on NPCs and pre implantation embryos was performed as previously described[86,87]. NPCs were dissociated using Accutase (Invitrogen) and attached to Poly-L-Lysine (Sigma)-coated coverslips for 10 min. Cells were fixed with 3% paraformaldehyde in PBS for 10 min at room temperature and permeabilized with ice-cold permeabilization buffer (1× PBS, 0.5% Triton X-100, 2 mM vanadyl–ribonucleoside complex (New England Biolabs)) for 4 min on ice. Coverslips were stored in 70% ethanol at −20 °C. Cells were dehydrated in increasing ethanol concentrations (80%, 95% and 100%) and air dried quickly. Probes were prepared from plasmid p510. Probes were fluorescently labelled by nick translation (Abbott). We used dUTP labelled with ATTO-488 green (Jena Bioscience) or Cy5 (Merck). Labelled probes were co-precipitated with mouse Cot-1 DNA (ThermoFisher) in the presence of ethanol and salt, resuspended in formamide, denatured at 75 °C for 8 min and competed at 37 °C for 40 min. Probes were hybridized in FISH hybridization buffer (50% formamide, 20% dextran sulfate, 2X SSC, 1 µg µl$^{-1}$ BSA (New England Biolabs), 10 mM vanadyl–ribonucleoside complex) at 37 °C overnight. The next day, coverslips were washed three times for 5 min with 50% formamide in 2X SSC at 42 °C and three times for 5 min with 2X SSC at room temperature. 4,6-Diamidino-2-phenylindole (DAPI; 0.2 mg ml$^{-1}$) was added to the second wash, and coverslips were mounted with Vectashield (Vectorlabs). For E3.5 and E4.5 embryos, we followed a similar protocol with these modifications: coverslips were incubated in Denhardt's solution (3X SSC, 0.2 mg ml$^{-1}$ BSA, 0.2 mg ml$^{-1}$ Ficoll-400 and 0.2 mg ml$^{-1}$ polyvinylpyrrolidone (PVP40)) in water) for 3 h at 65 °C, followed by incubation in 3:1 methanol–glacial acetic acid solution for 20 min at room temperature and in 0.25% (vol/vol) glacial acetic acid 0.1 M triethanolamine solution for 10 min at room temperature. The zona pellucida was removed by treatment with acidified Tyrode's solution. E3.5 embryos were permeabilized for 13 min and E4.5 for 20 min, respectively. Images were acquired with an OLYMPUS iXplore Spin-SR Spinning Disk microscope with a 60× or 100× objective. Images were analysed using ImageJ software (Fiji).

### Astrocyte immunostaining

Astrocytes were fixed in 3% paraformaldehyde for 10 min at room temparature and washed once with PBS. Cells were permeabilized with permeabilization solution (0.25% Triton, 2 mM vanadyl–ribonucleoside complex in PBS) for 10 min at room temperature. After permeabilization, cells were blocked with blocking solution (2.5% BSA, 1 U µl$^{-1}$ RNasin (Promega), 0.1% Tween-20 in PBS) for 1 h at room temperature and incubated overnight with primary antibodies in blocking solution (1:400 GFAP (#173002, Synaptic System), 1:200 Ki67 (#556003, BD biosciences)). After three 10-min washes with PBS + 0,1% Tween (PBST), cells were incubated with secondary antibodies in blocking solution for 1 h at room temperature (1:100 Alexa Fluor, Thermo Fisher Scientific). After three 10-min washes with PBST (DAPI was added to the second wash at 1:1000), samples were mounted using ProLong Diamond antifade mountant and imaged with an OLYMPUS iXplore Spin-SR Spinning Disk microscope 100×.

### NPC RNA extraction and RNA-seq

RNA was extracted from >1 M NPCs using the RNeasy kit and on-column DNAse digestion (QIAGEN). RNA integrity was measured using the Bioanalyzer Nano Kit. Only high-quality RNA was used for subsequent library preparation using the NEBNext Poly(A) mRNA Magnetic Isolation Module (E7490L) and NEBNext Ultra II Directional RNA Library Prep Kit for Illumina (E7760L, New England Biolabs (NEB)) implemented on the liquid handling robot Beckman i7. Obtained libraries that passed the quality control (QC) step were pooled in equimolar amounts, and the pools were loaded on the Illumina sequencer NextSeq 500 or NextSeq 2000 and sequenced bi-directionally, generating ~500 million paired-end reads that were each 75 bases long. For the 129/Sv × Cast-EiJ NPCs, RNA was quantified with Qubit RNA Broad Range assay (Q10211, Invitrogen). RNA was sent to Novogene for RNA integrity and purity quality check followed by eukaryotic strand-specific mRNA (with PolyA enrichment) library preparation. Libraries were then pooled and sequenced on an Illumina Novaseq 6000 platform for 2 × 150 bp paired-end reads for a total of 80 million reads per sample.

### Blastocyst collection and whole-embryo RNA-seq

All animal experimental designs and procedures were performed in agreement with the rules and regulations of the Institutional

Animal Care and Use Committee (IACUC) under protocol numbers 019-03-21EH and 24-007_HD_EH. Embryos were derived from mating between 8–40-week-old C57BL/6 TX males $X^{ptet}$/Y; R26$^{rtTA/WT}$ (whole embryo RNA-seq) or $X^{ptet}$/Y; R26$^{rtTA/rtTA}$ (RNA-FISH) and superovulated 5–7-week-old WT JF1 females. Superovulation of JF1 females was induced by injecting 50 µl anti-inhibin serum (AIS) followed by injecting 2.5 U hCG 45–47 h after the AIS injection. Embryos were harvested at E2.5 and E3.5 and cultured in vitro in G-1 PLUS media (Vitrolife) in the presence of 10 µg ml$^{-1}$ Dox for 24 h. For RNA-FISH analysis, approximately half of the collected embryos at each timepoint were grown in culture medium without Dox. The sex of the embryos was determined either by Xist RNA-FISH or by PCR after RNA-seq library preparation. Single embryos were picked and washed three times with transfer buffer (1× PBS, 0.4% BSA) and transferred into 0.2-ml PCR tubes containing 2 µl lysis buffer (0.7% Triton X-100, 2 U µl$^{-1}$ RNasin (Promega), 1 µl oligo-dT$_{30}$VN primer (10 µM 5′-aagcagtggtatcaacgcagagtact30vn-3′) and 1 µl 10 mM dNTP mix (ThermoFisher). Illumina libraries were prepared by using a modified smart-seq2 protocol[83] using SuperScript IV Reverse Transcriptase (RT) and tagmentation procedure as previously described (Henning 2018). The RT reaction mix was as follows: 2 µl SSRT IV 5x buffer; 0.5 µl 100 mM dithiothreitol; 2 µl 5 M betaine; 0.1 µl 1 M MgCl$_2$; 0.25 µl 40 U µl$^{-1}$ RNAse inhibitor; 0.25 µl SSRT IV; 0.1 µl 100 uM template-switching oligonucleotide, 1.15 µl RNase-free H$_2$O. The RT thermal conditions were 52 °C for 15 min and 80 °C for 10 min. cDNA was generated using 16 PCR cycles. The cDNA cleanup (0.6× solid-phase reversible immobilization ratio) was carried out omitting the ethanol wash steps, and the elution volume was 13 µl H$_2$O. For tagmentation, the sample input was normalized to 0.2 ng µl$^{-1}$. Obtained libraries that passed the QC step were pooled in equimolar amounts. After library preparation, the sex and genotype of each embryo were assessed by PCR for rtTA (rtTA_F, acgccttagccattgagatg, rtTA_R, tctttagcgacttgatgctc); Xist (Xist_F, ggttctctctccagaagctaggaa, Xist_R, tggtagatggcattgtgtattatatg) and Eif2s3y (Eif2s3y_F, aattgccaggttattttcattttc, Eif2s3y_R, agttcagtggtgcacagcaa). Libraries were sequenced at 50 bp paired-end reads on a NextSeq 2000 platform. The sequence of the rtTA transactivator integrated at the Rosa26 locus in TX mice is shown in Supplementary Fig. 2.

### *Xist* induction in vivo, ExE RNA extraction and RNA-seq

All experimental designs and procedures were performed in agreement with the rules and regulations of IACUC under protocol numbers 019-03-21EH and 24-007_HD_EH. We mated 8–40-week-old C57BL/6 TX males $X^{ptet}$/Y; R26$^{rtTA/WT}$ or $X^{ptet}$/Y; R26$^{rtTA/rtTA}$ with 8–10-week-old WT JF1 females. Xist was induced by adding Dox (1 g l$^{-1}$ Dox and 100 g l$^{-1}$ sucrose[57]) to the drinking water of pregnant females 3.5 days after coitum. The water bottles were changed every 48 h and protected from light. Non-induced embryos were obtained from pregnant females provided with water containing only sucrose. After 5 days of Dox treatment (E8.5 after coitum), embryos were dissected from the uteri of the pregnant females. ExE tissues were dissected in PBS and collected in 150 µl 1x RNA Protection Reagent (#T2011-1, NEB), snap-frozen and stored at −70 °C. RNA extraction was performed using the Monarch Total RNA Miniprep Kit (T2010S, NEB) following the protocol for mammalian whole-blood RNA extraction with a few modifications[88]. RNA was eluted in 30 µl water. Sexing of the ExE samples was performed using quantitative PCR (qPCR). cDNA was generated from 20 ng RNA using SuperScript IV RT (Invitrogen) and random hexamers. qPCR experiments were performed with Fast SYBR Green Master Mix (Applied Biosystems) according to manufacturer's instructions and analysed on a QuantStudio Real-Time PCR Light Cycler (Thermo Fisher Scientific). The following primers were used: Xist (Xist_F, ggttctctctccagaagctaggaa, Xist_R, tggtagatggcattgtgtattatatg), Eif2s3y (Eif2s3y_F, aattgccaggttattttcattttc, Eif2s3y_R, agttcagtggtgcacagcaa) and actin (Actin_F aaccctaaggccaaccgtgaaaag, Actin_R catggctgggggtgttgaaggtctc). For RNA-seq library preparation, 1 ng RNA (in 2.4 µl) was mixed

with 1 µl 10 mM dNTP mix and 1 µl oligo-dT$_{30}$VN primer. Subsequent library preparation steps were carried out as previously described for whole-embryo RNA-seq. Libraries were sequenced at 50 bp paired-end reads on a NextSeq 2000 platform.

### Capture Hi-C

Capture Hi-C was performed as previously described[89]. Two arrays of biotinylated RNA probes were designed to tile 3 Mb targets on the mouse X chromosome (*Hcfc1–Mecp2* cluster; ChrX: 72,590,000–75,430,000; *Kdm5c* cluster; ChrX: 150,210,000–153,045,000; *Kdm6a* cluster; ChrX: 15,725,000–18,725,000). The probe arrays were synthesized by Agilent according to SureSelect DNA target-enrichment technology.

### Enzymatic methylation sequencing

Genomic DNA (gDNA) from >1 M NPC was extracted using a column-based DNeasy Blood & Tissue Kit (QIAGEN). DNA integrity was tested on a 0.8% agarose gel, and high-quality gDNA was used to prepare libraries according to the NEBNext Enzymatic Methyl-seq Kit following the section for large insert libraries with minor modifications. A total of 55–100 ng gDNA was used per library, including a spike-in of pUC and lambda DNA as a control for methylation efficiency. Samples were fragmented using the Covaris S2 System to achieve an average fragment size of 350–400 bp and barcoded using five PCR cycles for 100 ng input and six PCR cycles for 55 ng input. The obtained libraries were pooled in equimolar amounts and sequenced at 100 bp paired-end reads on a NextSeq 2000 platform.

### ATAC-seq

Assay for transposase-accessible chromatin using sequencing (ATAC-seq) was performed following the Omni-ATAC protocol[90]. A total of 25,000 cells were collected, lysed for 3 min on ice in lysis buffer (10 mM Tris–HCl pH.5, 5 M NaCl, 1 M MgCl$_2$, 0.1% NP-40, 0.1% Tween-20, 0.01% digitonin), washed in wash buffer (10 mM Tris–HCl pH.5, 5 M NaCl, 1 M MgCl$_2$, 0.1% Tween-20) and spun down at 500 RCF at 4 °C for 10 min. Pellets were resuspended in 50 µl transposition reaction (2X TD buffer, 1× PBS, 0.1% Tween-20, 0.1% digitonin, 5 µl Illumina Tn5 transposase) and incubated for 30 min at 37 °C with 1,000 rpm agitation. DNA was isolated with a Zymo DCC5 kit and eluted in 21 µl elution buffer. DNA samples were initially amplified by five cycles of PCR, followed by a variable number of additional amplification cycles estimated by qPCR for each sample. PCR products were purified using the Zymo DCC5 kit and eluted in 20 µl water. A two-size selection of fragments was performed using 0.5× and 1.3× volume of AMPure XP beads (Beckman Coulter). Libraries were quantified and analysed using Qubit and Tapestation assays, before preparing equimolar dilutions. Paired-end sequencing was performed on a NextSeq 500 (Illumina).

### CUT&RUN

CUT&RUN was performed as previously described[91]. In brief, 0.5 million cells were collected and permeabilized in 1 ml nuclear extraction buffer (20 mM HEPES pH 7.9, 10 mM KCl, 0.5 mM spermidine, 0.1% Triton X-100, 20% glycerol, c0mplete EDTA free). If needed, cells were frozen at −80 °C using slow-freezing pots to preserve integrity and then thawed on ice before starting the protocol. Cells were spun down at 3,500 rpm for 5 min and resuspended in 600 µl nuclear extraction buffer. Nuclei were then gently mixed with 300 µl activated bead slurry, prepared from 10 µl Bio-Mag Plus Concanavalin A-coated beads (86057, Polysciences) per 0.5 million cells and incubated at room temperature for 5–10 min on a rotating wheel. Blocking was performed on ice for 5 min in blocking buffer (wash buffer 20 mM HEPES pH 7.5, 150 mM NaCl, 0.5 mM spermidine, 0.1% BSA, c0mplete EDTA free, supplemented with 2 mM EDTA). After blocking, nuclei were washed with 1 ml wash buffer, resuspended in 500 µl wash buffer containing target antibodies diluted 1:100 (H3K27me3 (9733, Cell Signaling), H2AK119Ub (D27C4, Cell Signaling)) and incubated overnight at 4 °C on a rotating

wheel. The next day, nuclei were washed three times with wash buffer, A-MNase fusion protein (pA-MNase, generated by the EMBL PepCore facility) was added at 700 ng ml⁻¹ in 500 μl wash buffer and they were incubated at 4 °C for 1 h on a rotating wheel. After three washes, nuclei were resuspended in 150 μl wash buffer and equilibrated to 0 °C in a metal block on ice for 5 min. Chromatin digestion was performed for 30 min on ice by adding 3 μl 100 mM CaCl₂ to the sample. Digestion was stopped by adding 150 μl 2X STOP buffer (200 mM NaCl, 20 mM EDTA, 4 mM EGTA, 0.1% NP-40, 40 μg ml⁻¹ glycogen). Next, the NaCl concentration in the sample was raised to 300 mM by adding 20 μl 5 M NaCl to the sample, and RNA digestion was performed using 1.5 μl RNAse A (Thermo Scientific, 10 mg ml⁻¹) for 20 min at 37 °C. Following beads removal, the supernatant was treated with ProteinaseK (Thermo Scientific, 300 μg ml⁻¹) in 0.1% SDS for 30 min at 56 °C. Total DNA was extracted using phenol-chloroform, precipitated with 100% EtOH at −20 °C overnight and size selection was performed using Ampure XP beads (double-sided size selection: first round: bead slurry added at 0.5× the sample volume; second round: bead slurry added at 1.3× the sample volume). The DNA was eluted in 25 μl low-EDTA buffer (10 mM Tris, 1 mM EDTA (pH8)), quantified using Qubit and analysed on Tapestation (Agilent). Barcoded CUT&RUN libraries were prepared from 25 ng DNA using the NEBNext Ultra II DNA Library Prep Kit for Illumina according to the manufacturer's protocol and sequenced on a NextSeq 2000 with 75 bp paired-end read settings.

### Bioinformatics

**Allele-specific pre-processing of RNA-seq data.** All steps for the pre-processing of RNA-seq data can be reproduced using a nextflow pipeline available at https://github.com/yuviaapr/allele-specific_RNA-seq. Reads were trimmed using trim_galore (v0.6.6). To construct reference genomes for allele-specific mapping, genomes in which known heterozygous variants (https://ftp.ebi.ac.uk/pub/databases/mousegenomes/REL-2112-v8-SNPs_Indels/mgp_REL2021_snps.vcf.gz) were masked by the ambiguous base N were constructed using SNPsplit (SNPsplit_genome_preparation script, v0.5.0) and converted to STAR references (STAR v2.5.3a). For the E6, CL30/CL30.7 and CL31/CL31.16, the mm10 genome (GRCm38) was used. For the embryo and 129/Sv × CAST/EiJ cell lines, the mm11 genome was used (GRCm39). Reads were aligned to the N-masked genomes using the options –sjdbOverhang 99 –outFilterMultimapNmax 1 –outFilterMismatchNmax 999 –outFilterMismatchNoverLmax 0.06 –alignIntronMax 500000 –alignMatesGapMax 500000 –alignEndsType EndToEnd. The rtTA transgene was included in the reference genome used for the mapping of the embryo data (Supplementary Fig. 2). Reads mapping to the mitochondrial genome were removed and split into parental genotypes using SNPsplit and known heterozygous variants. Gene-level read counts for each haplotype and without allelic resolution were derived using featureCounts (v2.0.1).

**Computation of allelic ratios in RNA-seq for escapee definition.** Allele-specific expression per gene was calculated from genotype-assigned read counts. First, lowly expressed genes with an average allelic read count lower than ten were excluded (summing both haplotypes). Escape of an X-linked gene was generally quantified as the fraction of read counts on the inactivated X against the total read count across both active and inactive X ($Xi / (Xi + Xa)$, allelic ratio, also known as $d$-score). Genes were considered as escaping when the allelic ratio exceeded 0.1 (10% of expression from the inactive X). As the causal gene of XCI, *Xist* was not considered an escapee. A small number of other genes showed allelic ratios >0.8, which probably represents either strain-specific expression or technical artefacts due to erroneous mapping or single nucleotide polymorphisms (SNPs) with non-reference genotypes, rather than genuine expression largely restricted to the Xi. These genes were excluded for subsequent analysis. For all analyses of escape in the E6 cell line, we used the set of 133 genes

(134 including *Xist*) which showed an allelic ratio >0.1 in all untreated replicates. The use of N-masking of the genome should reduce mapping bias at SNP positions. Furthermore, we focus on relative changes (reductions) in allelic ratios, which should be unaffected by technical artefacts.

**Escapee meta-analysis.** To obtain a consensus of escaping genes from literature, we performed a meta-analysis of papers that used genome-wide expression profiling to derive escapees. For each study (Supplementary Table 1), we considered the genes defined as escapees or silenced by the experimental and analysis methodology used in that paper. In each study, a gene is therefore considered 'escaping', 'silenced' or 'not detected'. We then classified genes as 'constitutive escapees' if they were detected in at least three and escaped in more than 50% of the studies. We defined genes as 'facultative escapees' if they were escaping in less than 50% of studies or were only detected in fewer than three (but escaped in at least one study). Finally, we defined genes as 'silenced' if they were not escaping in any assayed study.

**Differential allelic imbalance analysis.** To test for differential allelic imbalance (differences in escape between conditions), we used binomial generalized linear models (R package stats, v4.2.0). We modelled the number of reads mapping to the inactive X as binomially distributed with the formula $(k, n) = 1 + $ treatment and tested for the significance of a treatment coefficient using a Wald test. $P$ values were adjusted using the Holm–Bonferroni method.

**Modelling of escape trajectories.** We used nonlinear parametric regression to fit an exponential decay curve to the allelic ratios after *Xist* overexpression. Specifically, we used the nls function (R, stats) to fit the allelic ratios $r = Xi / (Xi + Xa)$ with the exponential decay function $r = a * \exp(-k * t) + b$, where $t$ represents the number of days of Dox treatment, $a$ is a scaling parameter of the decay, $b$ allows for an asymptotic offset and $k$ specifies the decay constant. In particular, $k$ relates to the speed (half-life) of silencing loss by $t_{1/2} = -\ln(0.5) / k$ and $b$ quantifies whether there is residual escape after prolonged *Xist* overexpression. We also fit the same model without $b$, to test which genes showed evidence for residual escape as opposed to full silencing. After excluding all genes with a regression $R^2 < 0.3$ for both fits, we used the Bayesian information criterion to classify whether genes showed residual escape.

**Definition of escapee gene groups.** To define locally close groups of escapees on the X chromosome, we iterated through all escapees on the X, grouping genes into one 'gene group' if the gene start position was within 100 kb of each other and splitting whenever this was not the case. This approach partitioned the escapee set into 11 groups of three or more genes (with 3–14 escapees) and 57 singles or pairs.

**Differential expression analysis.** We used DESeq2 (R, v1.36.0) to test both haplotype-specific and total expression counts for differential expression using standard settings, using the formula ~Treatment and adjusting $P$ values using the Benjamini–Hochberg correction.

**Allele-specific analysis of DNA methylation data.** Processing of DNA methylation was performed using the methylseq nextflow pipeline with the emseq option (v2.3.0, v3.3.0, NextFlow v22.10.6, v24.10.02). Reads were trimmed using TrimGalore (v0.6.7) and aligned to the same N-masked genome (GRCm38) using bismarck (v0.24.0), and we used SNP_split (v0.5.0) to assign mapped reads to parental genomes, similar to the RNA-seq data. We finally used bismark_methylation_extractor from the Bismarck software to generate base-level methylation counts for both haplotypes. We retrieved genomic coordinates of CpG islands (http://genome.ucsc.edu/cgi-bin/hgTrackUi?g=cpgIslandExt), gene bodies and promoters (2 kb upstream to 200 bp downstream

of transcription start sites) (EnsDb.Mmusculus.v79, v2.99.0). To quantify DNA methylation in promoters, we used regions of accessible chromatin defined by ATAC-seq data in matched cell lines (see below) if they overlapped with annotated promoter regions. We quantified average methylation values across per-base methylation values $b$ = methylated / (methylated + unmethylated) in the regions of interest.

**Allele-specific Capture Hi-C analysis.** Allele-specific Capture Hi-C analysis was performed using the Hi-C Pro pipeline (v2.11.4)[92] as previously described[89]. Paired-end reads were first independently aligned using bowtie2 to the same N-masked genome reference. Read pairs were then assigned to either C57BL/6 J or CAST-EiJ genomes, and allele-specific BAM files for downstream analysis were generated. Low-quality reads, multiple hits, singletons and read pairs that did not map to the same genotype were then removed. Further filtering for valid interactions, which excludes reads outside of the captured region, was performed, and only valid pairs were used to build the normalized contact maps at a 10-kb resolution with the cooler package (v0.8.9). Insulation scores were calculated using the cooltools package (v0.3.2). Comparative Hi-C contact maps were generated using the HiCExplorer suite. The maps were first normalized to observed / expected values using HiCTransform, followed by comparative analysis with HiCcompare. All data were visualized using pyGenomeTracks[93]. Capture Hi-C statistics including sequencing depth and mapped reads are presented in Supplementary Table 8.

**Allele-specific ATAC-seq analysis.** All steps for the pre-processing of ATAC-seq data can be reproduced using a nextflow pipeline available at https://github.com/yuviaapr/allele-specific_ATAC-seq. In brief, initial quality checks of the raw FASTQ files were performed using FastQC (v0.11.9). Reads were trimmed using TrimGalore (v0.6.3). A reference genome for allele-specific mapping was constructed as described for the allele-specific RNA-seq analysis and used to generate a bowtie index. Paired-end reads were aligned using bowtie2 (v2.3.4.1) to the N-masked genome index with the following parameters: –very-sensitive -X 2000. Duplicates were removed using Picard Tools (v2.20.8) before downstream analyses. Peaks were called using MACS2 (v2.2.7.1) using the following parameters: -g mm –buffer-size 100 -q 0.01 –keep-dup all –min-length 100 –format BAMPE –nomodel. Consensus peaks were generated using the 'dba.peakset' function from the DiffBind Bioconductor R package (v3.8.4), with the requirement that peaks should be present in at least two BAM files. Blacklist regions (from ENCODE for BSgenome.Mmusculus.UCSC.mm10) were removed using the dba.blacklist function, and the raw counts for the resulting consensus peak set were generated using the dba.count function using summits = FALSE. Peak counts were normalized using the dba.normalize function, with default settings. Differential accessibility analysis between the split BAM files coming from the Xi and Xa was performed using the dba.analyze function with methods = DBA_DESEQ2 (calling the DESeq2 Bioconductor R package (v1.38.3) in the background). Peaks were annotated to their nearest gene using the ChIPseeker Bioconductor R package (v1.34.1) using the mm10 genome from the TxDb.Mmusculus.UCSC.mm10. knownGene Bioconductor R package (v3.10.0).

**Allele-specific CUT&RUN analysis.** All steps for the pre-processing of CUT&RUN data can be reproduced using a nextflow pipeline available at https://github.com/yuviaapr/allele-specific_CUTandRUN. In brief, paired-end reads were mapped with Bowtie2 (v2.3.4.1) (–very-sensitive -X 1000 –no-mixed –no-discordant) against an N-masked genome reference (generated as described for allele-specific RNA-seq analysis). Low-quality and mitochondrial reads were filtered using Samtools (v1.9). Genotype assignment was performed with SNPsplit (v0.3.4), followed by duplicate removal with Picard Tools (v2.20.8). Three BAM files were generated: C57BL/6, CAST-EiJ and total reads (allelic and

unassigned). BigWig files were generated from each BAM file using bamCoverage, with scale factors calculated using edgeR on 10 kb binned coverage data.

**Processing of embryo RNA-seq data.** The embryo RNA-seq data were pre-processed as described above (see 'Allele-specific pre-processing of RNA-seq data' section), using the SNPs between C57BL/6 and JF1 mice (https://ftp.ebi.ac.uk/pub/databases/mousegenomes/REL-2112-v8-SNPs_Indels/mgp_REL2021_snps.vcf.gz). To confirm the annotated genotypes of the embryos, reads were mapped to the rtTA transgene sequence using Bowtie2 (v2.5.4), and any embryo with >150 mapped transgene reads was considered rtTA positive. For developmental staging, principal component analysis (PCA; prcomp, stats v4.2.0) was performed on total read counts after size-factor normalization (estimateSizeFactors, DESeq2, v1.36.0) and subsetting on highly variable genes (getTopHVGs, scran, v1.24.1). The first principal component separated embryo stages as determined by morphological staging and was used as a developmental pseudotime. For the allelic analysis, genes were considered escapees if they showed an allelic ratio >0.1 in at least 50% of embryos. Genes were annotated into escape categories as in the meta-analysis and classified as 'E4.5 and E8.5 specific' if they did not escape in E6 and/or were not annotated as escapees in the meta-analysis.

**Processing of human RNA-seq data.** The RNA-seq count tables from normal brain tissues were collected through the GTEX data portal (v8, https://www.gtexportal.org/home/downloads/adult-gtex/bulk_tissue_expression). Only female samples were used ($n$ = 728). The raw RNA-seq counts were normalized using the trimmed mean of $M$ values (TMM) method[94]. Human escapees were obtained from the work of Tukianinen et al.[7]. Only genes classified as 'escape' and 'variable' were considered. In total, 128 genes were detected in the GTEX data. The X-linked gene expression levels were obtained by calculating the mean expression of all genes per sample, and the normalized ($z$-score) mean expression of escapees versus the normalized expression of Xist was plotted.

**Statistics and reproducibility.** No statistical methods were used to predetermine sample size. The experiments were not randomized and the investigators were not blinded to allocation during experiments and outcome assessment.

### Reporting summary

Further information on research design is available in the Nature Portfolio Reporting Summary linked to this article.

## Data availability

RNA-seq, Capture Hi-C, ATAC-seq and Methyl-seq data generated in this study have been deposited in the Gene Expression Omnibus under accession number GSE259400. Previously published GTEX data that were re-analysed here are available in the data portal (v8, https://www.gtexportal.org/home/downloads/adult-gtex/bulk_tissue_expression). All sequencing datasets were aligned using NCBI RefSeq GRCm38 or GRCm39 genome assembly and using the corresponding RefSeq transcript annotations Mus_musculus.GRCm38.100.gtf or Mus_musculus.GRCm39.113.gtf. All other data supporting the findings of this study are available from the corresponding author on reasonable request. Source data are provided with this paper.

## Code availability

All code to reproduce the analysis presented in the paper is available via GitHub at https://github.com/odomlab2/xist_project. The preprocessing workflows for RNA-seq and CUT&RUN data are available via GitHub at https://github.com/yuviaapr/allele-specific_RNA-seq and https://github.com/yuviaapr/allele-specific_CUTandRUN.

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

## Acknowledgements

This work was funded by an ERC Advanced Investigator award to E.H. (award no. XPRESS - AdG671027); the European ITN Innovative and Interdisciplinary Network 'ChromDesign', under the Marie Skłodowska-Curie grant agreement no. 813327 to E.H.; a Marie Skłodowska-Curie Actions Individual Fellowship to A.L. (grant no. IF-838408); core funding from the German Cancer Research Center (to O.S. and D.T.O.) and the European Molecular Biology Laboratory (to O.S.); the European Research Council (grant agreement nos. 810296 / DECODE to O.S. and 788937 / CTCFStableGenome and 615584 / EvoGeneticsTFBinding to D.T.O.); Wellcome Investigator Award (award no. 202878_Z_16_Z to D.T.O.); as well as the Bundesministerium für Bildung und Forschung Germany, project MERGE, Förderkennzeichen grant no. 031L0174C (to O.S.). H.Y.C. is an Investigator of the Howard Hughes Medical Institute and he is supported by the National Institutes of Health (NIH) grant no. RM1-HG007735. We thank T. Pollex for critical and thoughtful reading of the manuscript, P. Ginno for discussion, P. Gestraud for help with data visualization and members of the Heard, Stegle and Odom laboratories for discussion. We thank V. Benes, the Genomics Core Facility and the laboratory animal resources at EMBL (Heidelberg) for support and assistance. We also thank L. Pace for provisioning JF1 female mice.

## Author contributions

A.L. and E.H. conceived the study. A.H. performed the NPC experiments (that is RNA-seq, Capture Hi-C and CUT&RUN) trained by A.L. and with technical support from I.R. unless stated otherwise. J.P. and E.K. processed and analysed all the presented data unless stated otherwise. C.P. and A.L. generated the NPC line and performed RNA-FISH and some of the RNA-seq experiments. L.V. prepared RNA-seq and Capture Hi-C libraries, and F.J. prepared enzymatic methylation sequencing libraries. Y.A.P.-R. and L.C. performed the ATAC-seq experiments and compiled the reference list of escapees. Y.A.P.-R. processed the ATAC-seq data and N.Servaas performed the analysis. N.Servant processed the Capture Hi-C with A.H. and analysed the human data. A.L. performed the astrocyte and ES cell experiments. A.L. and E.K. performed the in vivo experiments. C.L. prepared RNA libraries for the 129/Sv×Cast-EiJ NPCs. H.Y.C. supervised C.L. J.B.Z. supervised N.Servaas. O.S. and D.O. supervised J.P. and provided feedback throughout the project. A.L. and E.H. supervised the work and wrote the manuscript with input from E.K., J.P., A.H., D.T.O., O.S. and all other authors.

## Funding

## Competing interests

H.Y.C. is a cofounder of Accent Therapeutics, Boundless Bio, Cartography Biosciences and Orbital Therapeutics; he was an advisor of 10x Genomics, Arsenal Biosciences, Chroma Medicine, Exai Bio and Spring Discovery until 15 December 2024. H.Y.C. is an employee and stockholder of Amgen as of 16 December 2024. The other authors declare no competing interests.

## Additional information

**Extended data** is available for this paper at https://doi.org/10.1038/s41556-025-01823-6.

**Correspondence and requests for materials** should be addressed to Edith Heard or Agnese Loda.

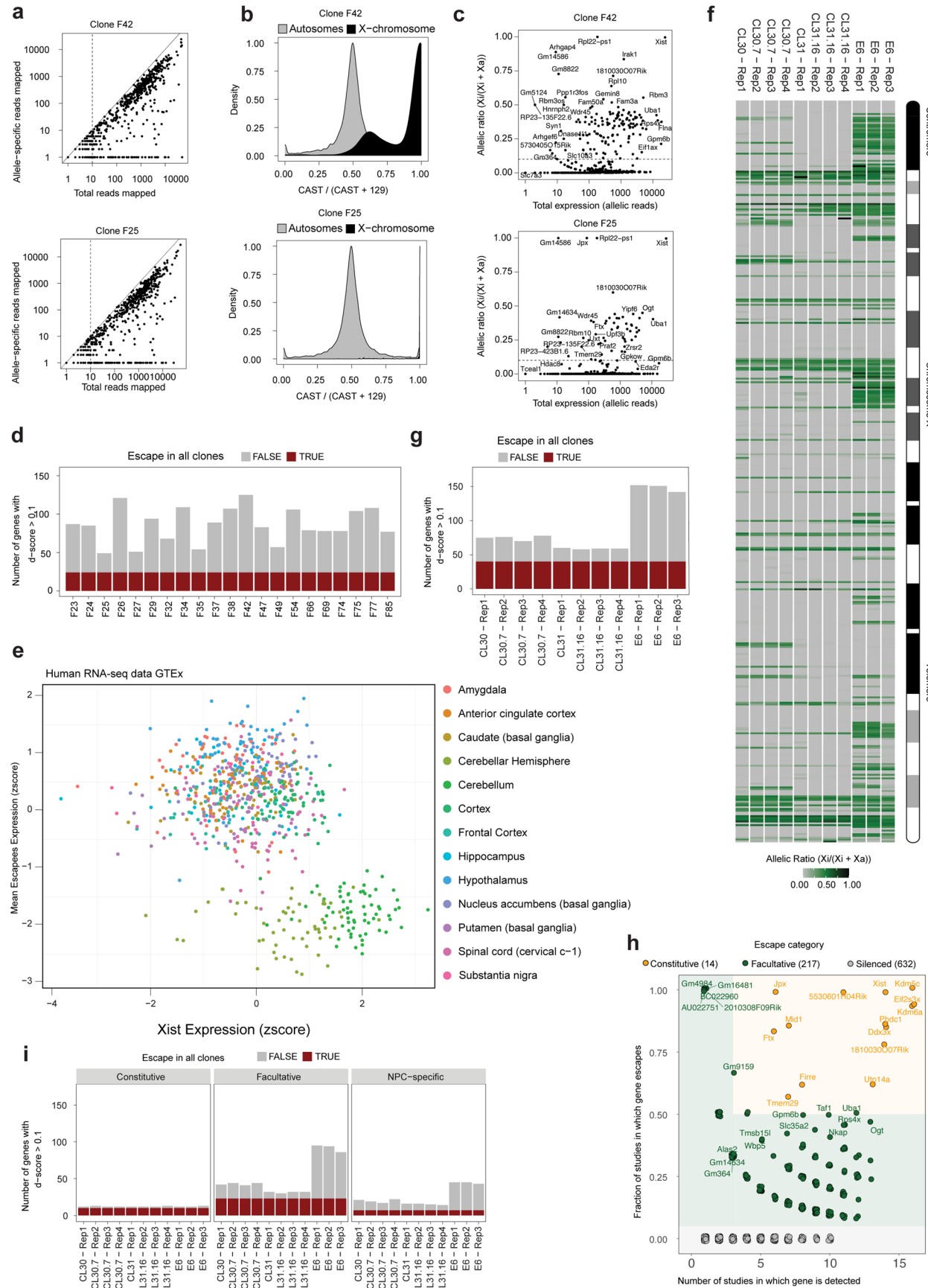

**Extended Data Fig. 1 | See next page for caption.**

**Extended Data Fig. 1 | Quantification of escape across neural progenitor cell lines and definition of escape categories across different NPC clones.**
**a**, Scatterplots showing the total number of reads (x-axis) and the number of allele-specific reads (y-axis) mapped to each gene in two out of 21 clonal F1 hybrid (129/Sv x Cast/EiJ) neural progenitor cell lines F42 (top, 1401252 reads) and F25 (bottom, 1297574 reads). The diagonal represents equal values. **b**, Density plot showing the distribution of allelic ratios (Cast/EiJ / (Cast/EiJ + 129/Sv); Xa / (Xi + Xa)) for the X-chromosome (black) and all autosomes (grey) in cell lines F42 (top, 16105 autosomal genes, 559 X-chromosomal genes) and F25 (bottom, 16662 autosomal genes, 572 X-chromosomal genes). Those cell lines were chosen to represent high (F42) and low (F25) general escape. **c**, Scatterplot showing the total number of reads per gene against their allelic ratio (Xi / (Xi + Xa)) in cell lines F42 (top, 377393 reads) and F25 (bottom, 359765 reads). Most genes show an allelic ratio of close to 0 (no expression from the Xi), while a subset shows higher allelic ratios (escape from XCI). **d**, Barplot quantifying the number of escapees (allelic ratio > 0.1) across the 21 clonal F1 hybrid (129/Sv x Cast/EiJ) neural progenitor cell lines. The red bar indicates the number of genes escaping

in all 21 clones (n = 24 genes) **e**, Scatterplot showing the relationship between mean escapee expression (zscore) and *Xist* expression level (zscore) in brain samples from the GTEx database. Each dot represents an individual sample, with colors indicating different brain tissue types (n = 728 datasets). **f**, Schematic of the mouse X chromosome and heatmap showing the allelic ratios of expressed X-linked genes in three F1 hybrid NPC clones (C57BL/6 x Cast/EiJ) NPC clones; E6, CL30 and CL31. Three replicates are shown for clone E6. CL30.7 and CL31.6 are sub-clones of CL30 and CL31 (Dossin et al. 2020) and 4 replicates are shown for each clone and sub-clone. **g**, Quantification of the number of escapees in the three NPC clones. The red bar indicates the number of genes escaping in all shown clones (n = 45 genes) **h**, Classification of X-linked genes in three categories based on their XCI status as described in 19 studies spanning multiple cell types and developmental contexts: (i) constitutive escapee (escapes XCI in > 50% of the studies); (ii) facultative escapee (variably escaping XCI in different studies and in at least one of the considered studies); and (iii) silent genes (escapes in no study). **i**, As in (**g**) but for the different escape categories defined in (**h**). The red bar indicates escape in all cell lines and replicates.

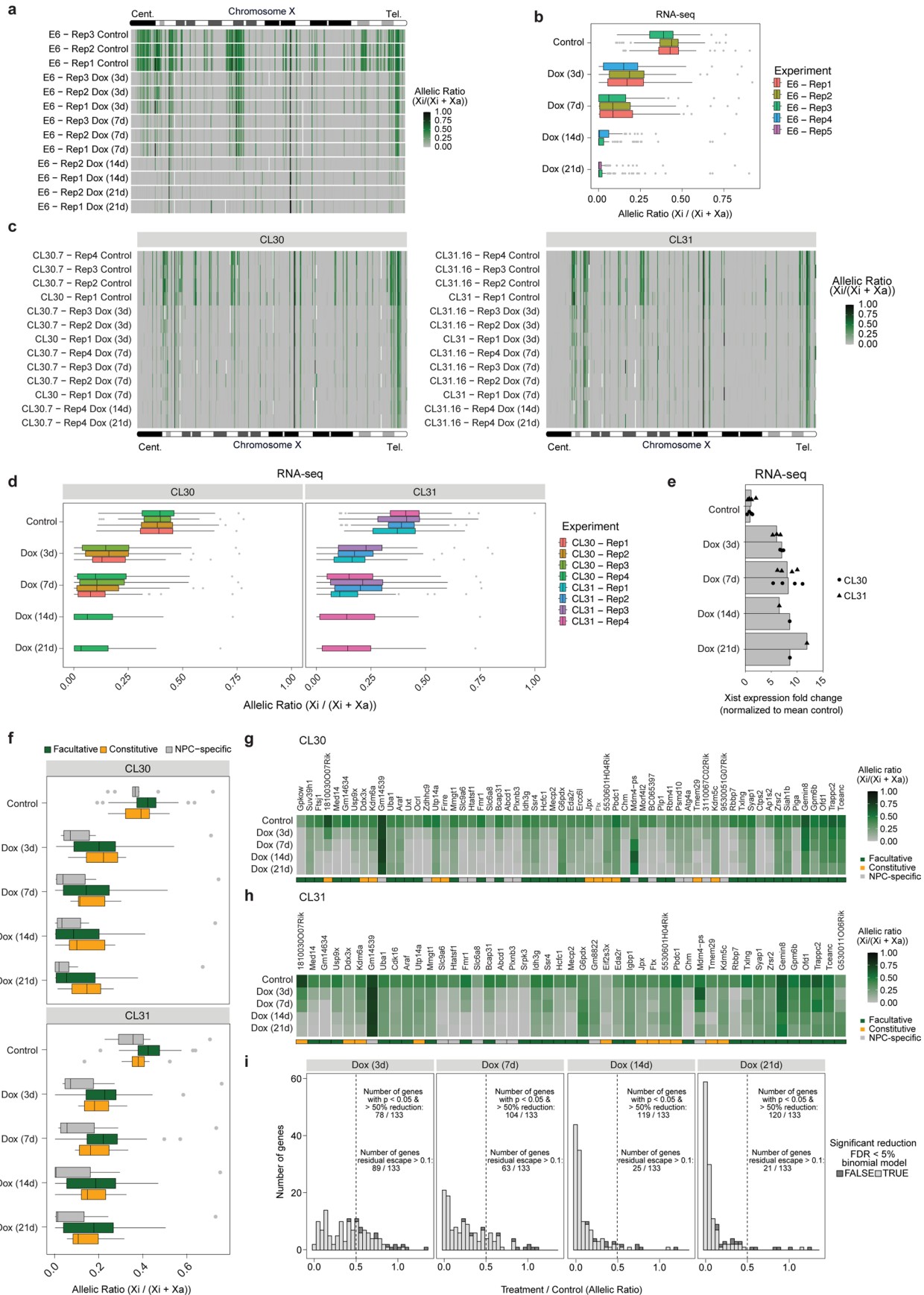

**Extended Data Fig. 2 | See next page for caption.**

**Extended Data Fig. 2 | Reduction of XCI escape upon *Xist* upregulation across clones and replicates. a**, Schematic of the mouse X chromosome and heatmap showing X-linked transcript allelic ratios for expressed genes in the untreated condition in clone E6 and after 3, 7, 14 and 21 days of Dox treatment. Data relative to 2 to 3 replicates per time point is shown. **b**, Box plots showing the changes in allelic ratios of escapees across the time course of Dox treatment for clone E6. Individual biological replicates are shown for each timepoint (Control, Dox (3 d) and Dox (7 d): n = 3, Dox (14 d) and Dox (21 d): n = 2 **c**, Same as (**a**), for clones CL30, CL31 and subclones. **d**, Same as (**b**), for clones CL30, CL31 and subclones. Individual biological replicates are shown for each timepoint (both CL30 and CL31: Control: n = 4, Dox (3 d): n = 3, Dox (7 d): n = 4, Dox (14 d): n = 1, Dox (21 d): n = 1) **e**, RNA-seq data showing the fold change of *Xist* expression compared to untreated cells across the time course of Dox treatment for clones CL30 and CL31. **f**, Box plots showing the changes in allelic ratios for different categories of escapees across the time course of Dox treatment for CL30 (Constitutive n = 11; Facultative n = 37, NPC-specific n = 13 genes) and CL31 (Constitutive n = 11; Facultative n = 28, NPC-specific n = 9 genes) Shown is the average for the primary and subcloned cell lines. **g**, Heatmap showing the allelic ratios of escapees identified in clone CL30. Shown is the average for the primary and subcloned cell lines. Escapees are assigned to three different categories as described in Extended Data Fig. 1h and described in the Methods. The escape category for each gene is indicated below the heatmap. **h**, same as (**g**) for clone CL31. **i**, Bar plots quantifying the decrease in escape upon Dox treatment in clone E6. Shown is the allelic ratio after treatment normalised on the untreated sample for each gene (Treatment / Control (Allelic Ratio)) (n = 133 escapees). Additionally, genes are stratified based on whether a significant decrease in allelic ratio was observed (generalized linear model, adjusted using the Benjamini-Hochberg procedure) and whether they retained an allelic ratio > 0.1. (Control, Dox (3 d) and Dox (7 d): n = 3, Dox (14 d) and Dox (21 d): n = 2 biological replicates). All box plots show median, 25- and 75-quantiles and 1.5x inter-quartile range. Source numerical data are available in source data.

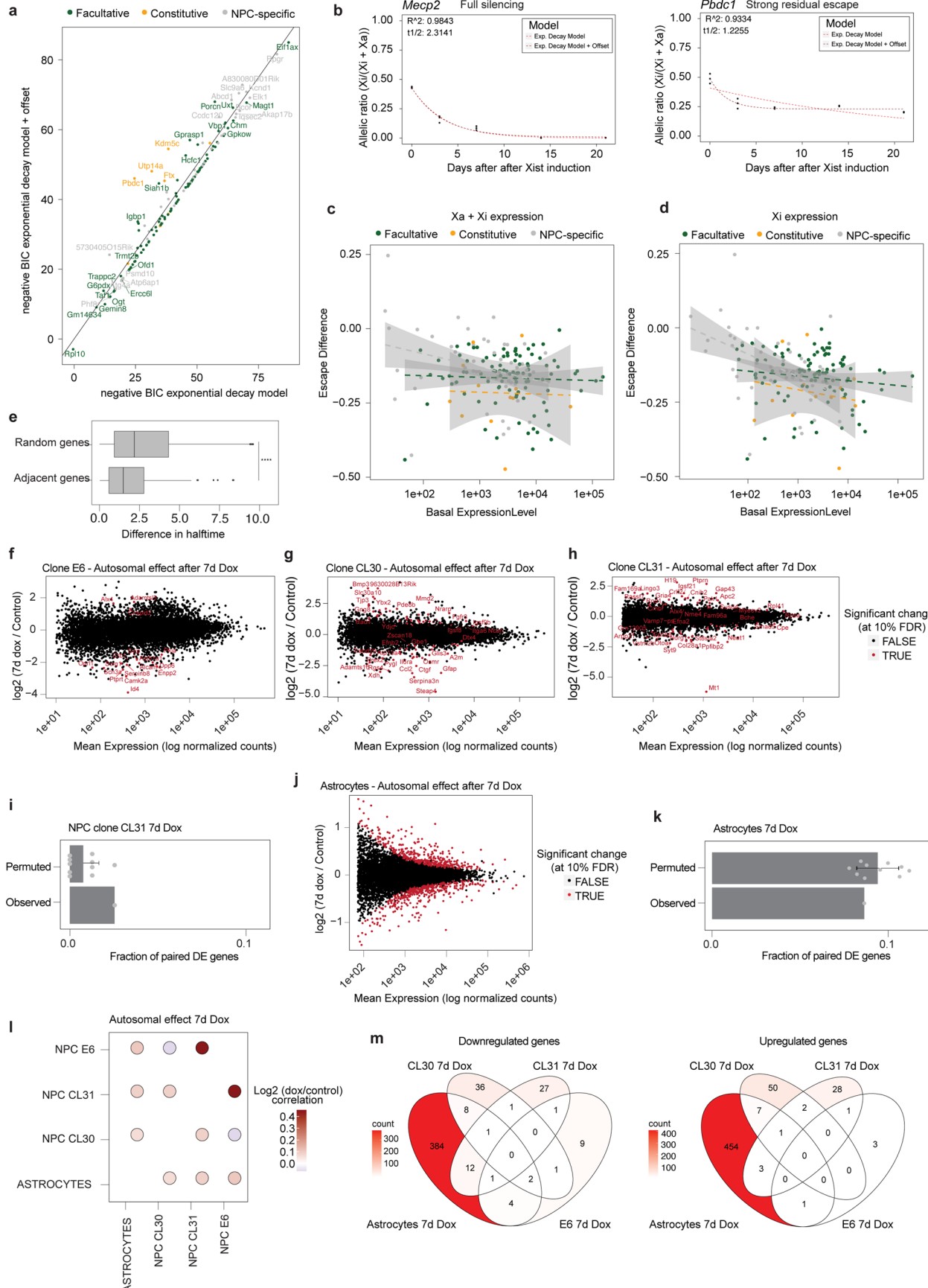

**Extended Data Fig. 3 | See next page for caption.**

**Extended Data Fig. 3 | Modelling of decay curves and autosomal differential expression analysis. a**, Scatterplot showing the negative Bayesian Information Criterion to compare model fits between the exponential decay model and the exponential decay model with offset, fit on silencing data from the E6 clone. Shown are 122 genes with averaged data from individual biological replicates per time point Control: n = 3, Dox (3 d): n = 2, Dox (7 d): n = 3, Dox (14 d): n = 2, Dox (21 d): n = 2. The diagonal indicates equal values and for genes above the diagonal the data is better explained by a non-zero offset parameter. Based on the BIC-based model comparison, 55.5% of constitutive escapees (n = 5), 18.3% (n = 15) of facultative escapees and 35.5% of NPC-specific genes (n = 11 genes). show significant, that is a lower BIC for the model including an offset term, residual escape. **b**, Example model fits for different genes, including the R2 goodness-of-fit statistic for the full silencing model, and the silencing half-life. Genes are shown to represent the spectrum of response curves, including complete and fast silencing (*Mecp2*), and fast silencing but strong residual escape (*Pbdc1*). **c**, Scatterplot showing that the magnitude of change in escape is not a function of gene expression level across Xi and Xa. The escape difference represents the difference in allelic ratio between 7 days of Dox treatment and untreated cells. **d**, as (**c**), but comparing escape differences against expression level on the Xi only. (**c**) and (**d**) show data across 133 genes (Constitutive n = 12; Facultative n = 84, NPC-specific n = 37). **e**, Box plots showing differences between half times for randomly chosen and adjacent gene pairs. P-value < 10^-15 using a two-sided Wilcoxon rank sum test. The box plots show median, 25- and 75-quantiles and 1.5x inter-quartile range. Numbers of individual biological replicates underlying the half times calculation as indicated in (**a**). **f**, MA-plot showing differential expression analysis of autosomal genes after 7 days of Dox treatment for clone E6. Significant genes are indicated in red. (n = 3 replicates per condition) **g**, as (**f**) for CL30 (n = 4 replicates per condition). **h**, as (**f**) for CL31 (n = 4 replicates per condition). **i**, Barplots assessing genomic colocalization of autosomal differentially expressed genes for CL31. The fraction of differentially expressed genes for which a nearby gene (within 100 kb) is also differentially expressed is shown. The 'Shuffled' bar plot represents the observed fraction after randomly permuting significance p-values across all detected genes (n = 10 iterations). Error bars indicate the standard deviation across 10 permutations. Bars depict mean, errorbars standard errors. Three individual biological replicates were included for Control and 7 d Dox treatment. **j**, as (**f**) for astrocytes (n = 11687 expressed genes, 2 replicates per condition). **k**, as (**i**) for astrocytes (n = 10 iterations). Two biological replicates were included for Control and 7 d Dox treatment. **l**, Dotplot comparing autosomal effects of *Xist* overexpression across experiments. For each cell line, log2 fold changes between Dox treated and untreated cells are calculated and points show pearson correlation coefficients between these log2 fold changes for each two cell lines. **m**, Venn diagrams showing the overlap of genes between downregulated and upregulated genes in the different cell lines. Source numerical data are available in source data.

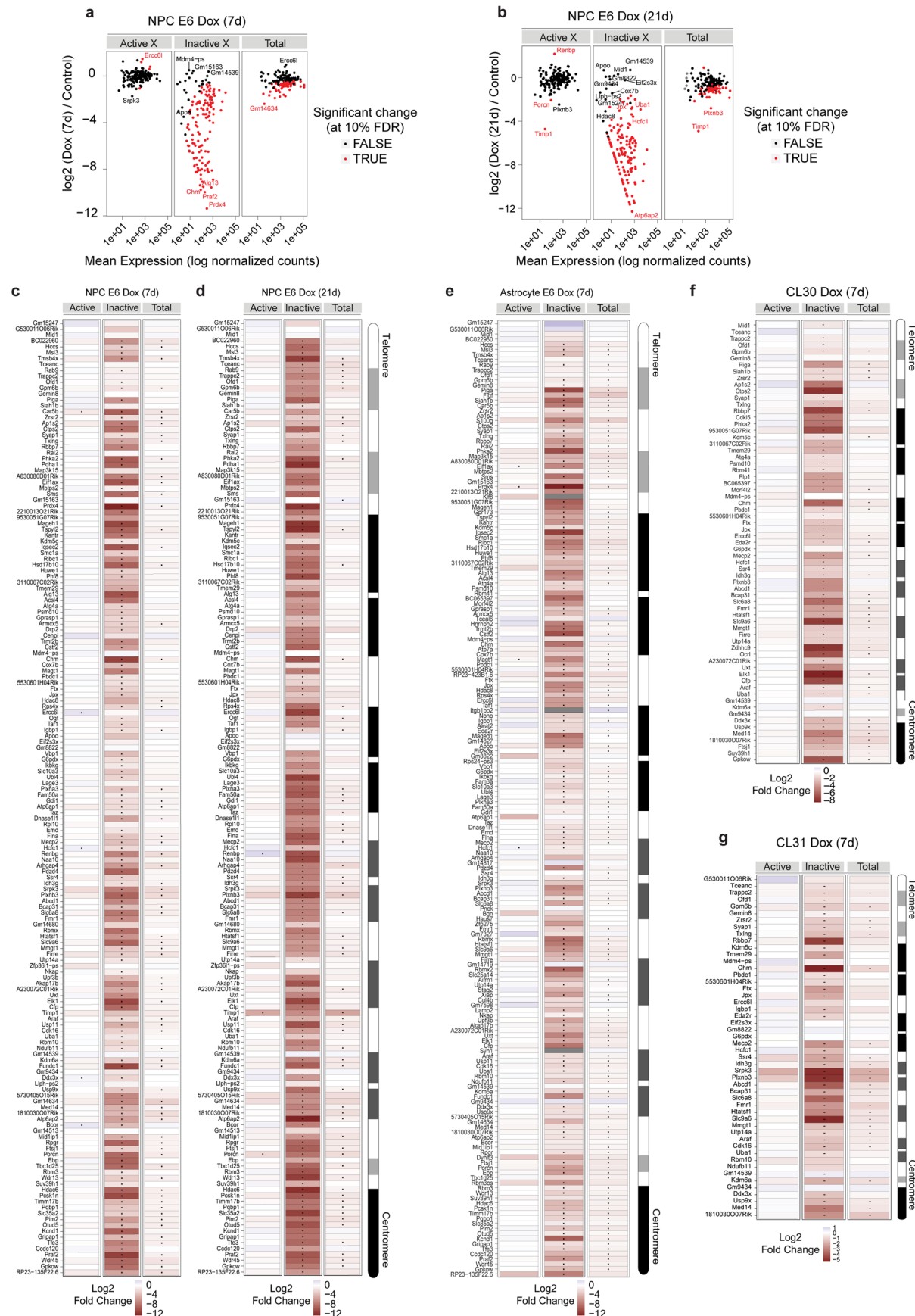

**Extended Data Fig. 4 | See next page for caption.**

**Extended Data Fig. 4 | Allele-specific differential expression analysis of X-linked genes. a**, MA-plots showing the results of a differential expression analysis in NPC clone E6 after 7 days of Dox treatment for X-linked genes considering reads from the Xa only, reads from the Xi only and total reads (including those not assignable to an allele) (n = 3 replicates per condition). P-values are computed using DESeq2 and FDR-adjusted. **b**, as (**a**), after 21 days of Dox treatment (3 replicates for control samples, 2 for Dox (21 d)). **c**, Heatmap showing the fold changes of X-linked gene expression after 7 days of Dox treatment for clone E6 as summarised in (**a**) for each gene individually. **d**, as (**c**), after 21 days of Dox treatment. **e**, as (**c**) for differentiated astrocytes. **f-g**, as in (**c**) for CL30 and CL31 respectively after 7 days of Dox treatment.

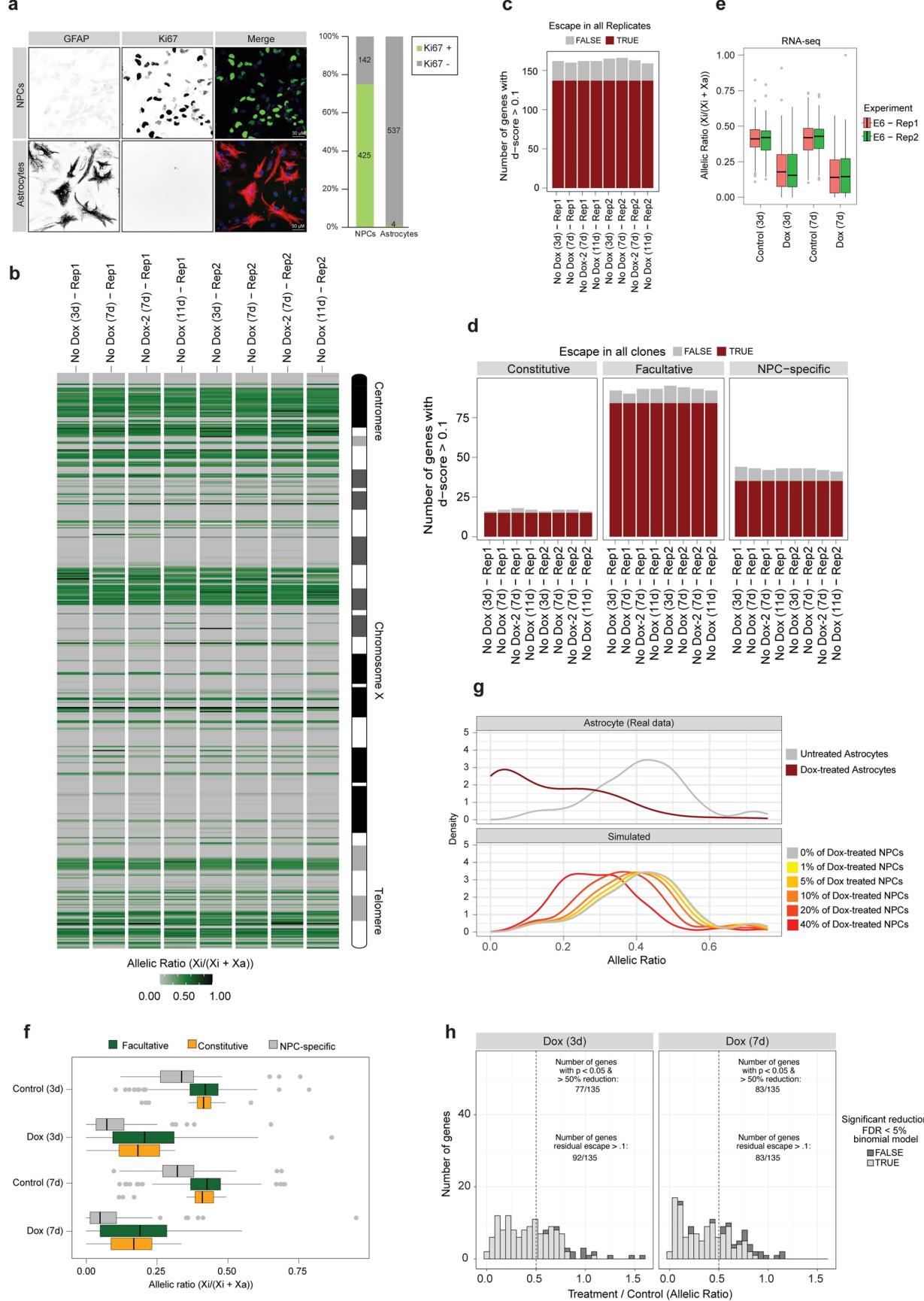

**Extended Data Fig. 5 | See next page for caption.**

**Extended Data Fig. 5 | Reduction of XCI escape upon *Xist* upregulation in astrocytes. a**, Immunofluorescence staining of E6 NPCs and differentiated astrocytes showing the astrocyte marker GFAP (red) and the proliferation marker Ki67 (green). DNA was stained with DAPI (blue). Number of cells: NPCs, n = 567; astrocytes, = 541. The percentage of cycling cells (Ki67 + ) is quantified and shown in the barplot. **b**, Schematic of the mouse X chromosome and heatmap showing the allelic ratios of expressed X-linked genes in astrocytes. The different control replicates are shown. **c**, Quantification of the number of escapees in F1 hybrid astrocytes by allele-specific RNA-seq. The red bar indicates the number of genes escaping in all shown control samples (n = 136 genes). **d**, Quantification of escapees identified in astrocyte control samples across the different escape categories (constitutive, facultative, and NPC-specific). The red bar represents escape events observed in all replicates (n = 10 constitutive, 89 facultative, 38 NPC-specific genes). **e**, Box plots showing the changes in allelic ratios of escapee distributions across the time course of Dox treatment for E6 astrocytes. n = 2 biological replicates for each timepoint. **f**, Box plots showing the changes in allelic ratios for the different categories of escapees across the time course of Dox treatment for E6 astrocytes (n = 10 constitutive, 89 facultative, 38 NPC-specific genes). **g**, Top: density plots of allelic ratios of X-linked escapees in untreated (grey) and Doxycycline-treated astrocytes (dark red), demonstrating effective silencing upon Dox treatment. Bottom: density plots of simulated allelic ratios of X-linked escapees in untreated astrocytes with defined fractions of Dox-treated NPCs (0-40%). Low values indicate absence of escape from XCI (allelic ratio <0.1) **h**, Barplots quantifying the decrease in escape upon Dox treatment. Shown is the average allelic ratio after treatment normalised on the untreated sample for each gene (Treatment / Control (Allelic Ratio)) (n = 136 genes). Additionally, genes are stratified based on whether a significant decrease in allelic ratio was observed (binomial linear model, adjusted p-value < 0.05, see Methods) and whether they retained an allelic ratio > 0.1. **c-h**, show data across two replicates. All box plots show median, 25- and 75-quantiles and 1.5x inter-quartile range. Source numerical data are available in source data.

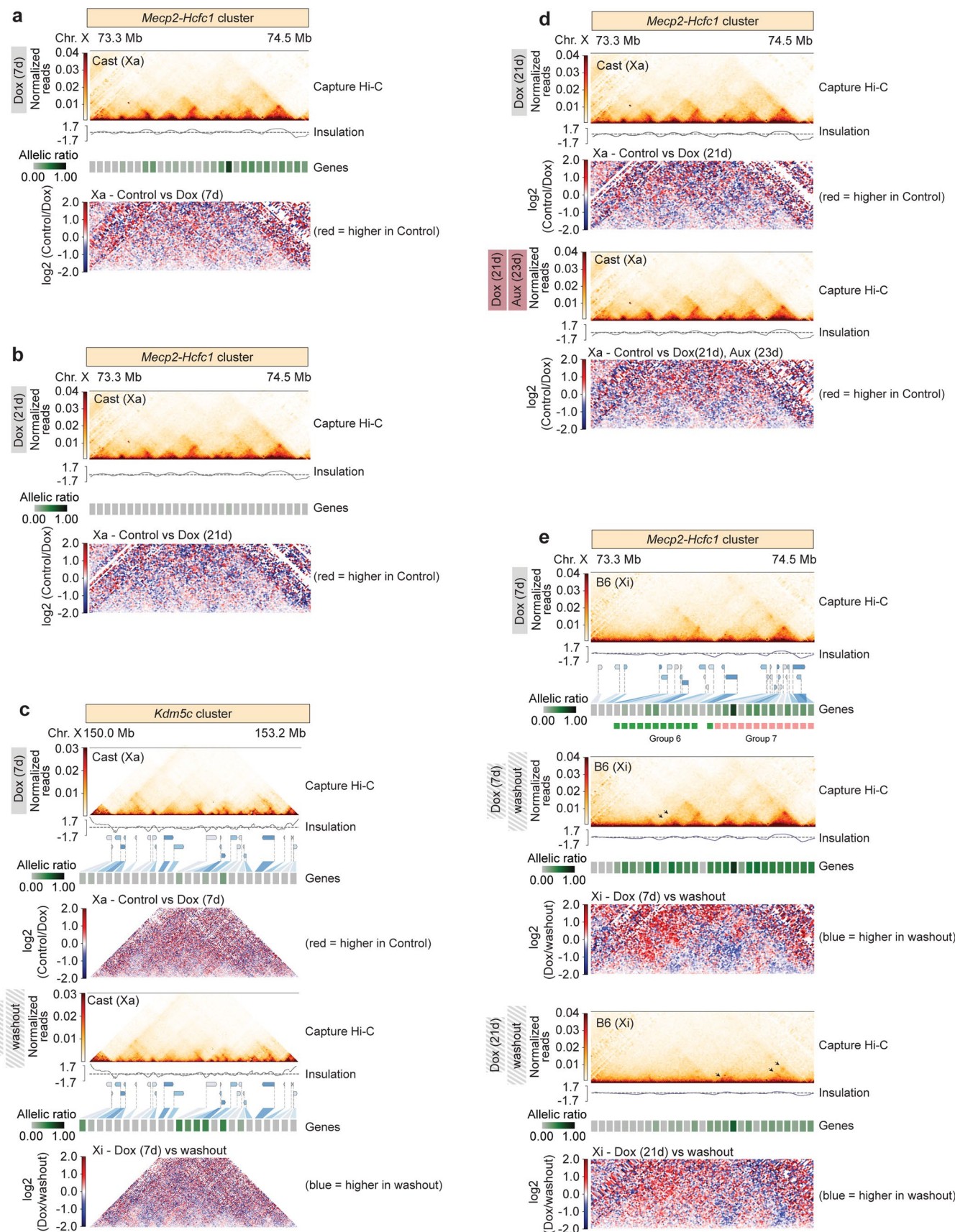

**Extended Data Fig. 6 | See next page for caption.**

**Extended Data Fig. 6 | Changes in 3D topology at the *Mecp2-Hcfc1* and *Kdm5c* clusters across different Dox treatment conditions. a,** Capture Hi-C interactions and insulation score for the active (Xa) X chromosome at the *Mecp2-Hcfc1* cluster in clone E6 upon 7 days of Dox treatment. Capture Hi-C data is shown at 10 kb resolution. Heatmaps showing the allelic ratios for 29 X-linked genes included in the captured regions are shown. Differential map (bottom) shows changes in genome topology between 7 day Dox treated samples and control samples. **b**, as in (**a**) but for 21 days of Dox treatment. **c**, Capture Hi-C interactions and insulation score for the active (Xa) X chromosome at the *Kdm5c* cluster in clone E6 upon 7 days of Dox treatment (top) and washout (bottom). Capture Hi-C data is shown at 10 kb resolution. Heatmaps showing the allelic ratios for 25 X-linked genes included in the captured regions are shown. Differential maps show changes in genome topology between the indicated samples. **d**, Capture Hi-C interactions and insulation score at the *Mecp2-Hcfc1*

cluster for Xa in clone CL30.7. Top to bottom: 21 days of Dox treatment; 23 days of Auxin treatment in combination with 21 days of Dox treatment. Capture Hi-C data is shown at 10 kb resolution. Differential maps show changes in genome topology between 21 days Dox treated samples and control samples (top) and 23 days of Auxin and 21 days of Dox treated samples compared to control (bottom). **e**, Capture Hi-C interactions and insulation score at the *Mecp2-Hcfc1* in clone E6 upon Dox treatment and washout. Capture Hi-C interactions are shown for the inactive (Xi) X chromosome upon Dox treatment for 7 days (top) and upon Dox washout following 7 (middle) and 21 days of Dox treatment (bottom). Capture Hi-C data is shown at 10 kb resolution. Heatmaps showing the allelic ratios for 29 X-linked genes included in the captured regions are shown. Differential maps show changes in genome topology between the indicated treated samples and control samples.

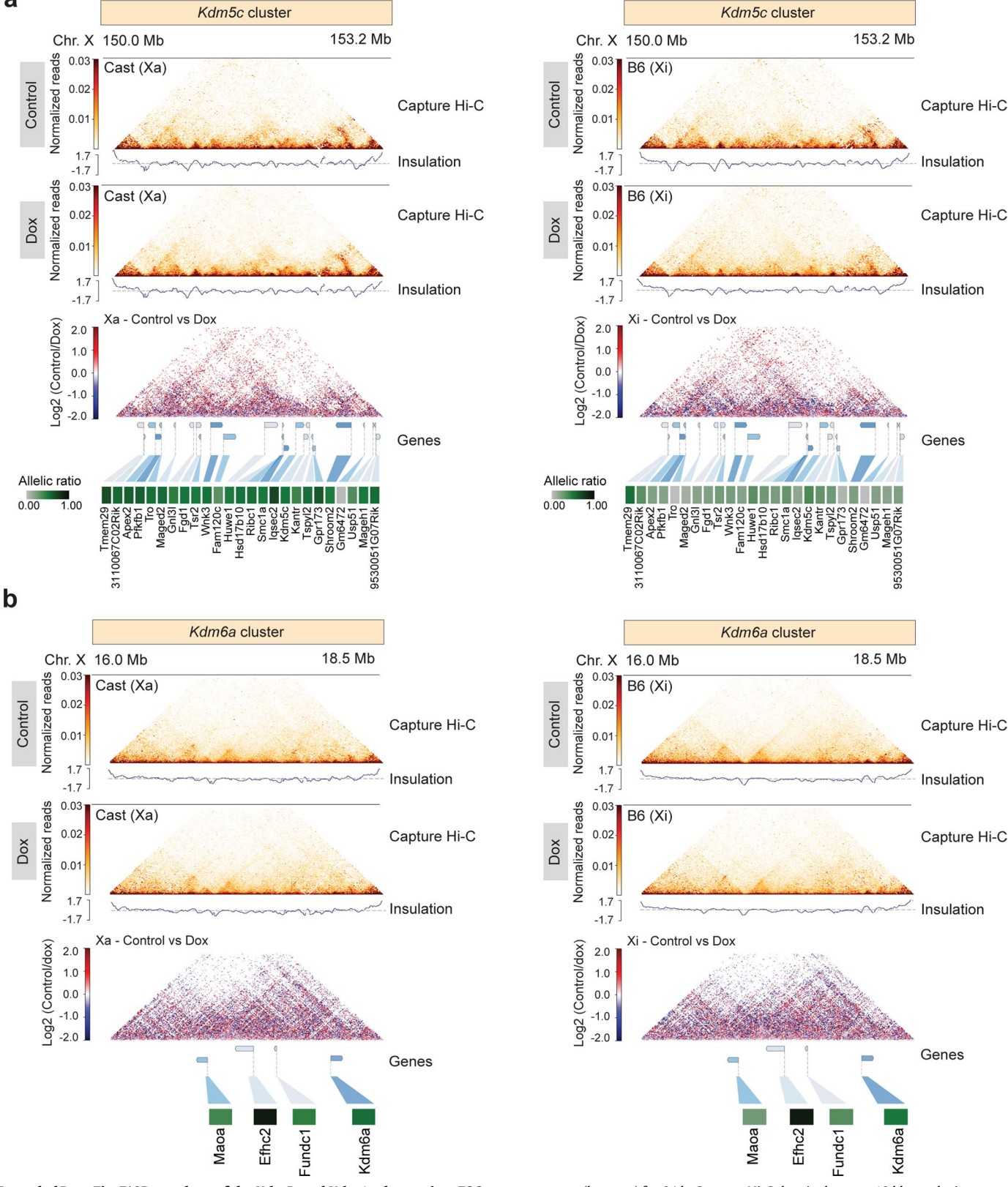

**Extended Data Fig. 7 | 3D topology of the *Kdm5c* and *Kdm6a* clusters in mESCs upon Dox treatment. a-b**, Capture Hi-C interactions and insulation score at the (**a**) *Kdm5c* and (**b**) *Kdm6a* cluster in mESCs prior to and upon Dox treatment. Capture Hi-C interactions are shown for the active (Xa, left) and the inactive (Xi, right) X chromosomes in untreated condition (Control) (top) and upon Dox treatment (bottom) for 24 h. Capture Hi-C data is shown at 10 kb resolution. Heatmaps showing the allelic ratios for 25 (**a**) and 4 (**b**) X-linked genes included in the captured regions are shown. Differential maps show changes in genome topology between the indicated treated and control samples.

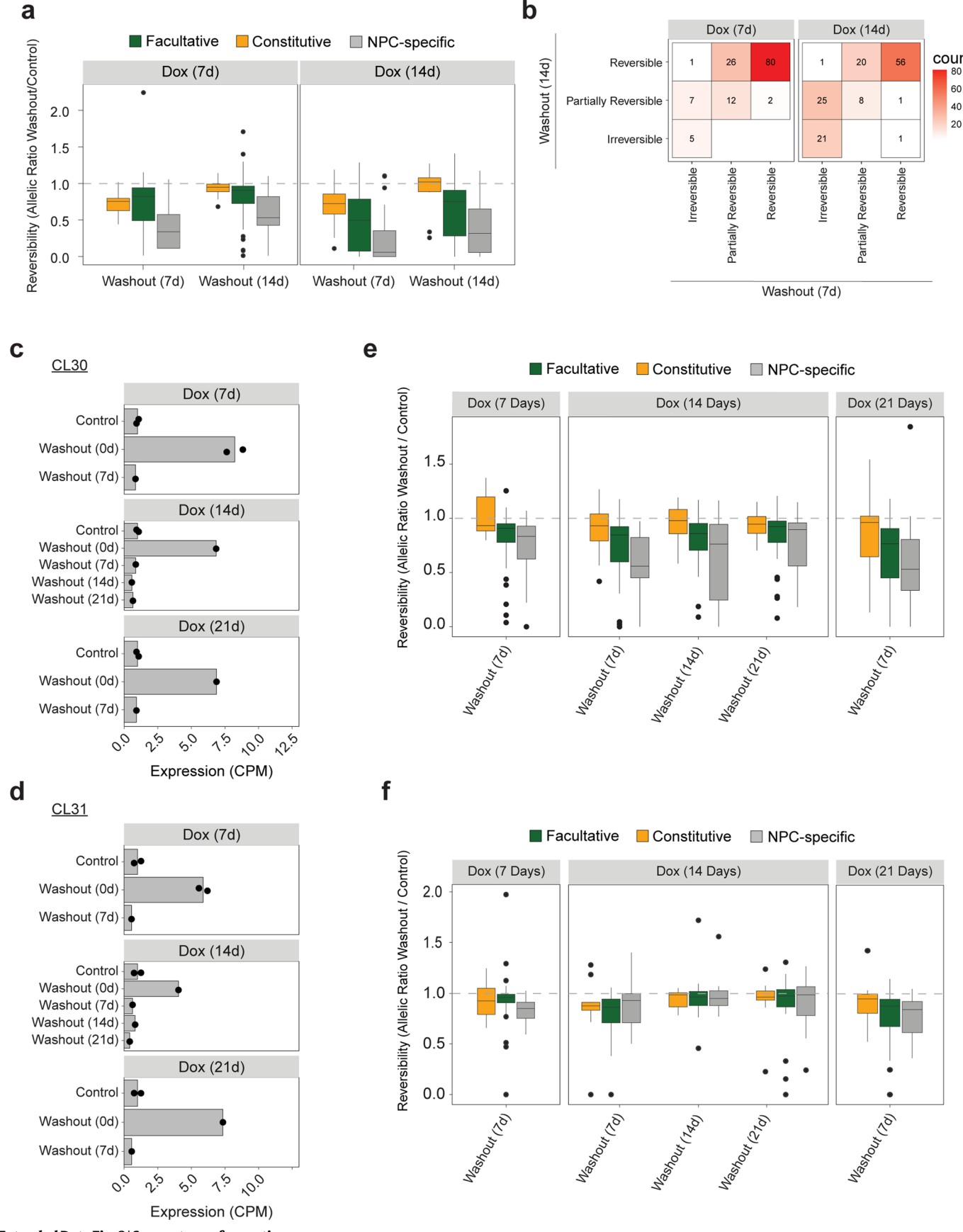

**Extended Data Fig. 8 | See next page for caption.**

**Extended Data Fig. 8 | Prolonged washout of Dox results in limited changes in silencing reversibility. a**, Box plot quantifying the reversibility of escape per gene for each time point by computing the ratio of allelic ratio after washout divided by the allelic-ratio in untreated samples. Data related to clone E6 is shown as a single biological replicate, with constitutive: n = 12, facultative n = 84, NPC-specific n = 37 genes. **b**, Heatmap comparing number of escapees per category of reversibility after 7 and 14 days of Dox washout. Data related to clone E6 is

shown. **c**, Xist RNA levels (normalised CPM) in clones CL30 across the Dox time course experiment (Control, Dox (7 d): n = 2, all other conditions: n = 1). **d**, as in **(c)** for clone CL31 (Control, Dox (7 d): n = 2, all other conditions: n = 1). **e**, as in **(a)** for clone CL30 (Constitutive: n = 11 genes, facultative n = 37, NPC-specific n = 13). **f**, as in **(a)** for clone CL31 (Constitutive: n = 11 genes, facultative n = 28, NPC-specific n = 8). All box plots show median, 25- and 75-quantiles and 1.5x inter-quartile range. Source numerical data are available in source data.

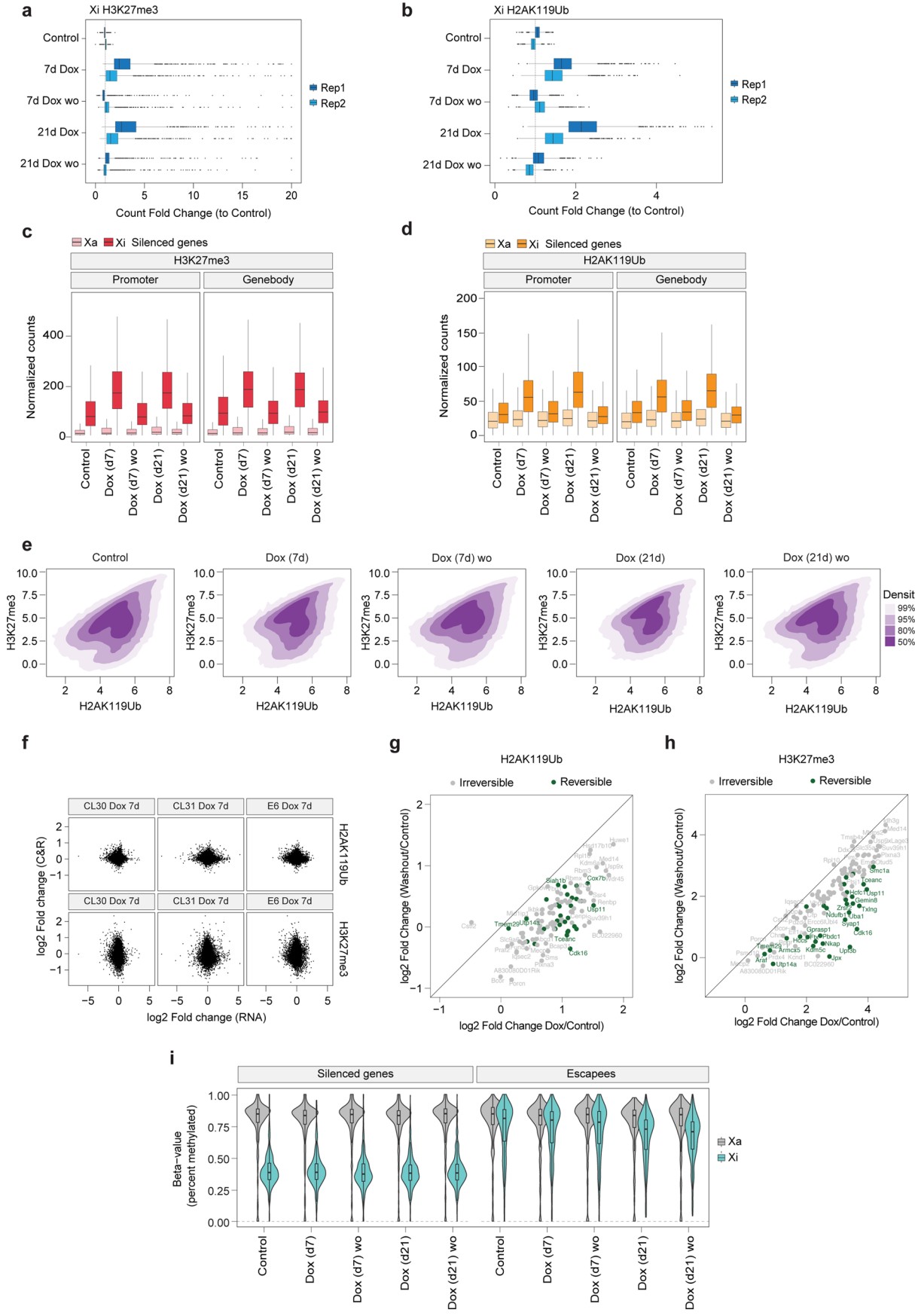

**Extended Data Fig. 9 | See next page for caption.**

**Extended Data Fig. 9 | Histone CUT&RUN and DNA methylation analysis upon Dox treatment and washout. a**, Box plot showing the enrichment and loss of H3K27me3 on the Xi across 10 kb windows with > 10 average read counts spanning the inactive X chromosome (Xi) during Dox treatment (Dox (7 d), Dox (21 d)) and washout (for 7 days after each treatment (Dox (7 d) wo, Dox (21 d) wo)). Shown is the fold change of normalized counts relative to the untreated condition (Control) in two replicates (n = 1,499 10-kb bins). **b**, same as **(a)** for H2AK119Ub (n = 1,499 10-kb bins). **c**, Box plot comparing H3K27me3 coverage (normalized counts) across promoters and gene bodies of silenced X-linked genes after Dox treatment and washout. Levels are shown separately for the Xa (light red) and Xi (dark red) (n = 6514 intergenic, 1540 promoter, 1262 gene body genomic intervals). **d**, as in **(c)** for H2AK119Ub. H2AK119Ub levels are shown separately for the Xa (yellow) and Xi (orange) (n = 6514 intergenic, 1540 promoter, 1262 gene body genomic intervals). **e**, Density plots showing the correlation between H3K27me3 and H2AK119Ub enrichments across 10 kb windows with > 10 average read counts spanning all autosomes during the Dox treatment time course and washout. The axes represent normalized log counts. **f**, Scatterplots showing the changes in H3K27me3 (bottom) and H2AK119Ub (top) levels (log2 fold change relative to Control) compared to the changes in gene expression (log2 fold change relative to Control) for autosomal genes after 7 days of Dox treatment in NPC clones CL30, CL31 and E6. **g**, Scatterplot showing the $\log_2$ Fold Change in H2AK119Ub levels in 21 days Dox treated versus washout conditions, compared to the untreated condition (Control) for reversibly (green, n = 30 genes) and irreversibly (grey, n = 86 genes) silenced escapees. The diagonal line represents equal changes between treatment and washout. Genes below the diagonal lose the mark upon washout. **h**, same as **(g)** but for H3K27me3. reversible genes (green, n = 30 genes) and irreversible genes (grey, n = 86 genes) **i**, Violin plots showing allele-specific DNA methylation levels across gene bodies for silenced and escaping genes, as derived from the gene expression data in Fig. 1 (escapees = 145 genes; silenced = 226 genes). Shown are the Xa (grey) and Xi (light blue) chromosomes separately. Differences observed for escapees are significant for the Xi for all comparisons to the untreated condition (Control): 8.83e-8 (Dox (7 d), 2.37e-7 (Dox (7 d) wo), 4.22e-28 (Dox (21 d)), 4.17e−27 (Dox (21 d) wo). P-values are calculated using a paired Wilcoxon's rank sum test. All box plots show median, 25- and 75-quantiles and 1.5x inter-quartile range. **c-i**, average data across two biological replicates Source numerical data are available in source data.

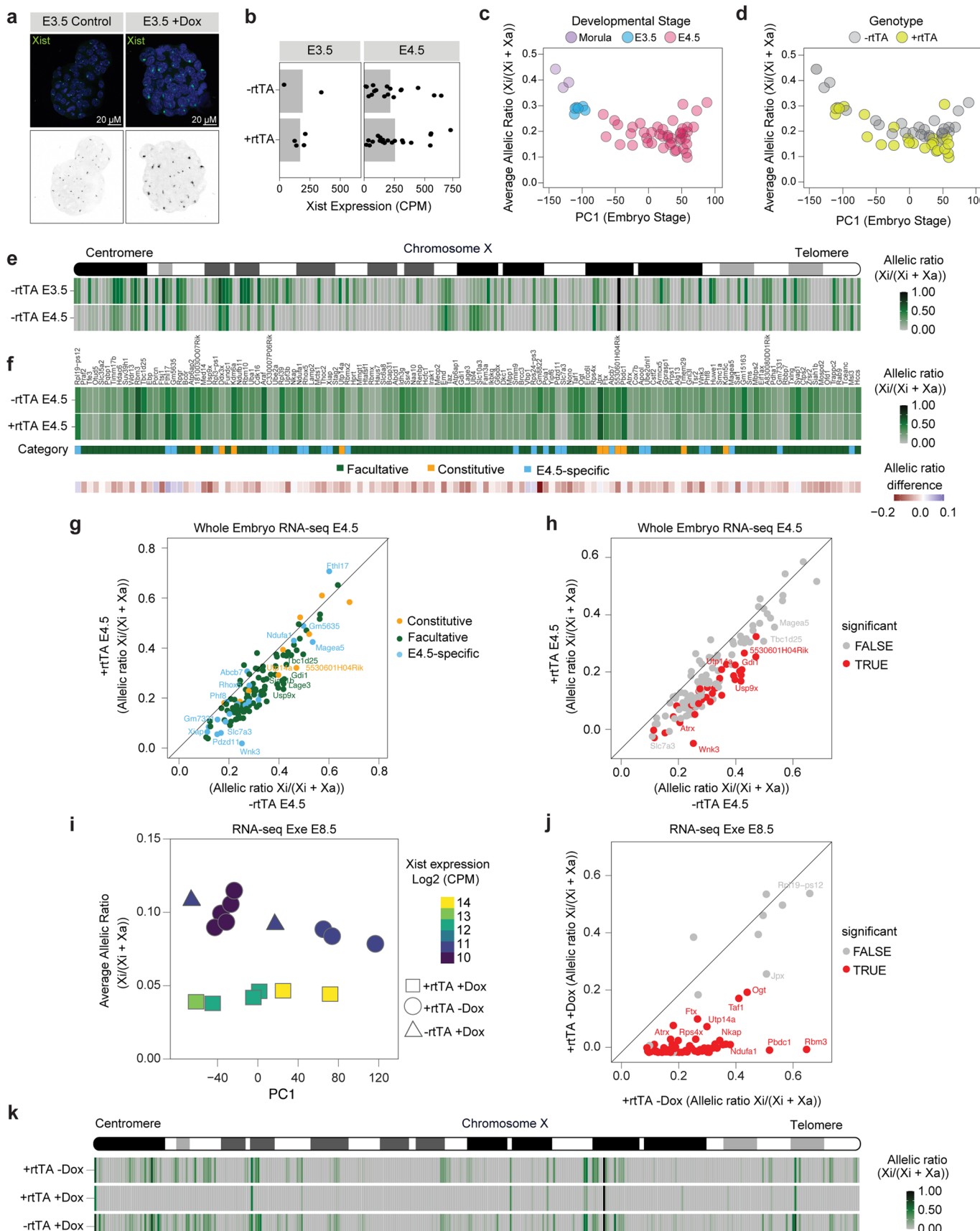

**Extended Data Fig. 10 | See next page for caption.**

**Extended Data Fig. 10 | Reduction of XCI escape following *Xist* upregulation in preimplantation embryos and extraembryonic tissues. a**, RNA-FISH for Xist RNA (green) in E3.5 XX embryos obtained by crossing male TX B6 mice (X$^{ptet}$Y; R26$^{rtTA/rtTA}$) with WT JF1 females. DNA is stained with DAPI (blue). Data are representative of n = 3 experiments. **b**, Whole embryo RNA-seq at E3.5 and E4.5 showing Xist RNA levels (normalised CPM) in females carrying the rtTA transactivator compared to WT females. Each dot represents one embryo. Biological replicates: For E3.5, -rtTA n = 2 embryos, +rtTA n = 4 embryos; For E4.5, -rtTA n = 17 embryos, +rtTA n = 20 embryos. **c**, Scatterplot showing average escape across the embryo stage as defined by principal component analysis on the whole transcriptome excluding X chromosomes. Average escape is defined as the mean allelic ratio of all expressed X-linked genes. Developmental stage annotations are defined based on embryo morphology. Biological replicates: For E3.5, -rtTA n = 2 embryos, +rtTA n = 4 embryos; For E4.5, -rtTA n = 17 embryos, +rtTA n = 20 embryos. **d**, as (**c**), but showing rtTA genotypes based on RNA-seq. Embryos are classified as +rtTA by PCR if at least 150 reads mapped to the rtTA transgene (see Methods). Differences between -rtTA and +rtTA samples are significant. Same samples were used in c and d. (p = 0.0006, two-sided Wilcoxon rank-sum test). **e**, Schematic of the mouse X chromosome and heatmap showing the allelic ratios of X-linked genes expressed in female embryos at E3.5 and E4.5. For each gene and developmental stage, the median allelic ratio is shown. Genes with an allelic ratio > 0.8 have been excluded except for *Xist*. For E3.5, -rtTA n = 2 embryos, For E4.5, -rtTA n = 17. **f**, Heatmap showing the mean allelic ratios of 131 escapees in E4.5 embryos carrying rtTA (+rtTA) and in matching controls (-rtTA). Escapees are categorised as in Extended Data Fig. 1h, with genes called

'E4.5-specific' if they show an allelic ratio > 0.1 in > 50% of -rtTA embryos and do not escape in NPCs. Below, the mean difference in allelic ratios between -rtTA and +rtTA embryos are shown. -rtTA n = 17 embryos, +rtTA n = 20 embryos. **g**, Scatterplot showing mean allelic ratios for individual genes for -rtTA and +rtTA embryos for the different escapee categories. rtTA n = 17 embryos, +rtTA n = 20 embryos. **h**, Scatterplot showing mean allelic ratios for individual genes for -rtTA and +rtTA embryos. Significant genes are indicated in red (padj < 0.1, two-sided t-test, adjusted using the Benjamini-Hochberg procedure). rtTA n = 17 embryos, +rtTA n = 20 embryos. **i**, Plot showing average escape for each extraembryonic sample, categorized by rtTA genotypes and Dox treatment. Samples are colored according to *Xist* expression levels (CPM) Principal component analysis (PCA) was performed on the whole transcriptome excluding X chromosomes. Biological replicates: -rtTA, +Dox n = 2 embryos; +rtTA, +Dox n = 6 embryos; +rtTA, -Dox n = 8 embryos. **j**, Scatterplot showing mean allelic ratios for individual genes in +rtTA embryos, either untreated or Dox-treated. Significant genes are indicated in red (padj < 0.1, two-sided t-test, adjusted using the Benjamini-Hochberg procedure). Biological replicates: -rtTA, +Dox n = 2 embryos; +rtTA, +Dox n = 6 embryos; +rtTA, -Dox n = 8 embryos. **k**, schematic of the mouse X chromosome and heatmap showing the allelic ratios of X-linked genes expressed in female extraembryonic tissues at E8.5 for the indicated conditions. For each gene, the median allelic ratio is shown. Genes with an allelic ratio > 0.8 have been excluded except for *Xist*. Source numerical data are available in source data. Biological replicates: -rtTA, +Dox n = 2 embryos; +rtTA, +Dox n = 6 embryos; +rtTA, -Dox n = 8 embryos.

Agnese Loda

# Reporting Summary

## Statistics

For all statistical analyses, confirm that the following items are present in the figure legend, table legend, main text, or Methods section.

| n/a | Confirmed | |
|---|---|---|
| ☐ | ☒ | The exact sample size (*n*) for each experimental group/condition, given as a discrete number and unit of measurement |
| ☒ | ☐ | A statement on whether measurements were taken from distinct samples or whether the same sample was measured repeatedly |
| ☐ | ☒ | The statistical test(s) used AND whether they are one- or two-sided *Only common tests should be described solely by name; describe more complex techniques in the Methods section.* |
| ☐ | ☒ | A description of all covariates tested |
| ☐ | ☒ | A description of any assumptions or corrections, such as tests of normality and adjustment for multiple comparisons |
| ☐ | ☒ | A full description of the statistical parameters including central tendency (e.g. means) or other basic estimates (e.g. regression coefficient) AND variation (e.g. standard deviation) or associated estimates of uncertainty (e.g. confidence intervals) |
| ☐ | ☒ | For null hypothesis testing, the test statistic (e.g. *F*, *t*, *r*) with confidence intervals, effect sizes, degrees of freedom and *P* value noted *Give P values as exact values whenever suitable.* |
| ☒ | ☐ | For Bayesian analysis, information on the choice of priors and Markov chain Monte Carlo settings |
| ☒ | ☐ | For hierarchical and complex designs, identification of the appropriate level for tests and full reporting of outcomes |
| ☐ | ☒ | Estimates of effect sizes (e.g. Cohen's *d*, Pearson's *r*), indicating how they were calculated |

*Our web collection on statistics for biologists contains articles on many of the points above.*

## Software and code

Policy information about availability of computer code

| Data collection | No software was used for data collection. |
|---|---|
| Data analysis | All code to reproduce the analysis presented in the paper is available on github in the repository https://github.com/odomlab2/xist_project. The preprocessing workflows for RNA-Seq and CUT&RUN data are available at the following links: https://github.com/yuviaapr/allele-specific_RNA-seq and https://github.com/yuviaapr/allele-specific_CUTandRUN. |
| | Published software used: trim_galore (v0.6.6, v0.6.7, v0.6.3), Picard Tools (v2.20.8), MACS2 (v2.2.7.1), bowtie2 (v2.3.4.1), SNPsplit (SNPsplit_genome_preparation script, v0.3.4, v0.5.0), STAR (v2.5.3a, v2.7.2b), featureCounts (v2.0.1), stats, v4.2.0, DESeq2 (R, v1.36.0, v1.38.3, v.1.44.0), EnsDb.Mmusculus.v79, (v2.99.0), cooler (v0.8.9), cooltools (v0.3.2), pyGenomeTracks, DiffBind (v3.8.4),Nextflow (20.04.1), BSgenome.Mmusculus.UCSC.mm10, ChipSeeker (v1.34.1), TxDb.Mmusculus.UCSC.mm10.knownGene (v3.10.0), scran (v1.24.1), methylseq nextflow pipeline (v2.3.0, v3.0.0, NextFlow v22.10.6,v24.10.4), Hi-C Pro pipeline (v2.11.4), FastQC (v0.11.8, v0.11.9), samtools (v1.9), Bismark (v0.23.1), deeptools (v3.5.1, v3.5.2), HiCExplorer (v3.7.2), Fiji/ImageJ software. |

For manuscripts utilizing custom algorithms or software that are central to the research but not yet described in published literature, software must be made available to editors and reviewers. We strongly encourage code deposition in a community repository (e.g. GitHub). See the Nature Portfolio guidelines for submitting code & software for further information.

## Data

Policy information about availability of data

All manuscripts must include a data availability statement. This statement should provide the following information, where applicable:
- Accession codes, unique identifiers, or web links for publicly available datasets
- A description of any restrictions on data availability
- For clinical datasets or third party data, please ensure that the statement adheres to our policy

All newly generated data was deposited in the Gene Expression Omnibus (GEO) database, under the accession number GSE259400.

## Research involving human participants, their data, or biological material

Policy information about studies with human participants or human data. See also policy information about sex, gender (identity/presentation), and sexual orientation and race, ethnicity and racism.

| | |
|---|---|
| Reporting on sex and gender | This research does not involve human participants, their data, or biological material. |
| Reporting on race, ethnicity, or other socially relevant groupings | This research does not involve human participants, their data, or biological material. |
| Population characteristics | This research does not involve human participants, their data, or biological material. |
| Recruitment | This research does not involve human participants, their data, or biological material. |
| Ethics oversight | This research does not involve human participants, their data, or biological material. |

Note that full information on the approval of the study protocol must also be provided in the manuscript.

# Field-specific reporting

Please select the one below that is the best fit for your research. If you are not sure, read the appropriate sections before making your selection.

☒ Life sciences　　　☐ Behavioural & social sciences　　　☐ Ecological, evolutionary & environmental sciences

For a reference copy of the document with all sections, see nature.com/documents/nr-reporting-summary-flat.pdf

# Life sciences study design

All studies must disclose on these points even when the disclosure is negative.

| | |
|---|---|
| Sample size | No statistical analyses were used to predetermine sample sizes. Adequate statistics has been applied throughout the manuscript in order to make sure that the observed effects are significant given the reported sample size |
| Data exclusions | No data was excluded |
| Replication | Experiments were independently repeated and the number of replicates is indicated in the manuscript |
| Randomization | Samples were not randomized, given that samples were grouped according to their respective genotypes. |
| Blinding | No blinding was performed during collection or data analysis. |

# Reporting for specific materials, systems and methods

We require information from authors about some types of materials, experimental systems and methods used in many studies. Here, indicate whether each material, system or method listed is relevant to your study. If you are not sure if a list item applies to your research, read the appropriate section before selecting a response.

## Materials & experimental systems

| n/a | Involved in the study |
|---|---|
| ☐ | ☒ Antibodies |
| ☐ | ☒ Eukaryotic cell lines |
| ☒ | ☐ Palaeontology and archaeology |
| ☐ | ☒ Animals and other organisms |
| ☒ | ☐ Clinical data |
| ☒ | ☐ Dual use research of concern |
| ☒ | ☐ Plants |

## Methods

| n/a | Involved in the study |
|---|---|
| ☒ | ☐ ChIP-seq |
| ☒ | ☐ Flow cytometry |
| ☒ | ☐ MRI-based neuroimaging |

## Antibodies

| | |
|---|---|
| Antibodies used | anti-GFAP (#173002, Synaptic System, dilution: 1:400) , anti-Ki67 (#556003, BD biosciences, dilution: 1:200), anti-H3K27me3 (#9733, Cell Signaling, dilution: 1:100), anti-H2AK119ubi (#D27C4, Cell Signaling, dilution: 1:100) |
| Validation | All commercially available antibodies were validated by the manufacturers. |

## Eukaryotic cell lines

Policy information about cell lines and Sex and Gender in Research

| | |
|---|---|
| Cell line source(s) | TX1072 female mouse embryonic stem cells (mESCs) used in this study were derived from a cross of a TX/TX R26rtTA/rtTA female (Savarese 2006) with a Mus musculus castaneus male according to the animal care guidelines of Institut Curie (Paris) as described in Schulz 2014. NPC lines were generated upon in vitro differentiation of this ESC line. Astrocytes were generated upon in vitro differentiation of the NPC line E6, derived from TX1072 ESCs. |
| Authentication | TX1072 ESCs were not authenticated. Next Generation Sequencing confirmed their identity as mouse cells |
| Mycoplasma contamination | All cell lines used in this study were tested negative for mycoplasma contamination. |
| Commonly misidentified lines (See ICLAC register) | N/A |

## Animals and other research organisms

Policy information about studies involving animals; ARRIVE guidelines recommended for reporting animal research, and Sex and Gender in Research

| | |
|---|---|
| Laboratory animals | Female pre-implantation embryos and E8.5 extraembryonic tissues analyzed in this study were derived from natural mating between 8 to 40 week-old C57BL/6 TX males Xptet/Y; R26rtTA/WT  or Xptet/Y; R26rtTA/rtTA (Savarese 2006) with wild type Mus musculus molossinus (JF1/Ms) females. 5 to 7 week-old and 8 to 10 week-old females were used for superovulation and natural mating, respectively. Mice were housed under a 12h light/dark cycle (light from 07:00 to 19:00), with ad libitum access to food and beverages. Temperatures of 18C to 23C,  40-60% humidity. |
| Wild animals | The study did not involved wild animals. |
| Reporting on sex | We crossed female and male mice to obtain pre-implantation embryos or E8.5 embryos. As our work focuses on X-chromosome inactivation, which is a female-specific process, we only analyzed female embryos. We determined the sex of the embryos by PCR. |
| Field-collected samples | This study did not involve samples collected from the field. |
| Ethics oversight | All experimental designs and procedures were performed in agreement with the rules and regulations of the Institutional Animal Care and Use Committee (IACUC) of the European Molecular Biology Laboratory (EMBL) under protocols number 019-03-21EH and 24-007_HD_EH. |

Note that full information on the approval of the study protocol must also be provided in the manuscript.

# Plants

Seed stocks

*Report on the source of all seed stocks or other plant material used. If applicable, state the seed stock centre and catalogue number. If plant specimens were collected from the field, describe the collection location, date and sampling procedures.*

Novel plant genotypes

*Describe the methods by which all novel plant genotypes were produced. This includes those generated by transgenic approaches, gene editing, chemical/radiation-based mutagenesis and hybridization. For transgenic lines, describe the transformation method, the number of independent lines analyzed and the generation upon which experiments were performed. For gene-edited lines, describe the editor used, the endogenous sequence targeted for editing, the targeting guide RNA sequence (if applicable) and how the editor was applied.*

Authentication

*Describe any authentication procedures for each seed stock used or novel genotype generated. Describe any experiments used to assess the effect of a mutation and, where applicable, how potential secondary effects (e.g. second site T-DNA insertions, mosiacism, off-target gene editing) were examined.*

