## [Peer Review File · Nature Cell Biology]

Escape from X inactivation is directly modulated by Xist non-coding RNA

Corresponding Author: Dr Agnese Loda

Version 0:

Decision Letter:

Dear Dr Loda,

Your manuscript "Escape from X inactivation is directly modulated by levels of Xist non-coding RNA", has now been seen by 2 referees, who are experts in X-chromosome inactivation (referee 2); epigenetics (referee 3), and whose comments are pasted below. Reviewer#1 was unable to review and submit a report on the revised manuscript and Reviewer#3 has kindly provided comments on whether Reviewer#1's original concerns have been adequately addressed. In light of the reviewers' advice, we regret that we cannot offer to publish the study in Nature Cell Biology.

As you will see, while the referees acknowledge the additional experiments performed, they continue to raise concerns with regard to the generalisability of the findings, including whether fully differentiated cells exhibit time-dependent silencing of X-inactivation escapees and the strength of the main claims. Reviewers continue to note that only a single sustained increase in Xist RNA over time was examined, not multiple levels of Xist RNA as suggested by the title. Reviewers request clarification for the role of SPEN in Xist-mediated silencing which we note, is central to the main claims of the manuscript. We feel that taken together, these reservations are sufficiently important to preclude publication of this study in Nature Cell Biology.

Although we cannot offer to publish your manuscript, I suggest that you consider Nature Communications or the EMBO Journal as a suitable venue for this work. To transfer your manuscript, please use our manuscript transfer portal. You will not have to re-supply manuscript metadata and files, unless you wish to make modifications. For more information, please see our [manuscript transfer FAQ](http://www.nature.com/authors/author_resources/transfer_manuscripts.html?WT.mc_id=EMI_NPG_1511_AUTHORTRANSF&WT.ec_id=AUTHOR) page.

We would be happy to consult with our colleagues at Nature Communications and the EMBO Journal to see whether they would be interested in taking this manuscript further with the existing peer review history. Please do let me know if you would like us to pursue this option.

We are very sorry that we could not be more positive on this occasion, but we thank you for the opportunity to consider this work.

With kind regards,
Sabrya Carim

Sabrya Carim, PhD
(she/her/hers)
Senior Editor, Nature Cell Biology
Nature Portfolio

Springer Nature
The Campus, 4 Crinan Street, London N1 9XW, UK
sabrya.carim@springernature.com
<https://orcid.org/0000-0001-9485-1938>

Reviewers' comments:

Reviewer #2 (Remarks to the Author):

This reviewer appreciates the authors' efforts in addressing the previous questions and acknowledges that the paper presents useful and timely data. However, it remains challenging to ascertain the precise conclusions and the degree to which the available data support them. In particular, it would be helpful to clarify which observations can or cannot be generalized and to ensure that no alternative explanations are unintentionally overlooked. The following points detail specific major and minor concerns:

Major Points

In Vitro Differentiated Astrocytes

While the newly added data on in vitro differentiated astrocytes is appreciated, further evidence seems necessary to support the claim that "Xist RNA can efficiently initiate gene silencing in terminally differentiated post-mitotic cells."

Specifically, what is the percentage of any remaining dividing cells in the culture? How can the authors rule out that the observed silencing effect does not simply occur within a small fraction of these dividing cells?

Line 148–152: Inverse Correction

The "inverse correction" that was found through the authors' particular approach—without explicitly accounting for cell-type heterogeneity—may not be directly relevant to the findings in mouse NPCs.

Lines 309–314: Limited Overlap and Different Cell Types

The statement that a limited overlap between NPCs and astrocytes indicates "differing autosomal effects across different cell types" may require additional context or data. A small overlap alone might not conclusively demonstrate lack of shared target genes or mechanism.

Essential Role of SPEN

The claim that "SPEN is essential for Xist-mediated silencing of nearly all escapees in differentiated cells" appears to rely heavily on Extended Data Figure 6, where the statistical significance is lacking and directionality of each gene's change is not entirely clear.

Additionally, the response to the previous question 4c does not seem to have been incorporated into the revised manuscript, leaving an alternative explanation of the data unaddressed.

Question of Generalizability (Previous Question 5)

While this reviewer apologizes that the originally suggested analysis might not have been fully applicable, the core issue remains: how do the authors ensure that the findings from just two analyzed clusters can be generalized to all XCI escapees?

Minor Points

Placement of Related Work

The expanded discussion on the differences between this and related studies is helpful. It may further benefit readers if these related works are briefly introduced earlier in the manuscript to set expectations for the Results section (though this is not strictly required).

References to Lifespan

The toned-down discussion of lifespan claims is appreciated. However, it may strengthen clarity to remove indirect references to lifespan entirely if there is no direct evidence within the manuscript to support them.

Reviewer #3 (Remarks to the Author):

Hauth, Panten, Heard and colleagues present a revised manuscript testing whether the Xist lncRNA can silence escape genes outside of the putative developmental window when Xist RNA establishes silencing of genes on the inactive X chromosome. Based on initial observations by Wutz & Jaenisch (2000, *Molecular Cell*, PMID: 10882105), Xist RNA-mediated gene silencing is reversible in early stages of embryonic stem cell (ESC) differentiation but becomes irreversible upon further differentiation. Here, the authors revisit this observation and test if overexpression of Xist RNA in differentiating and terminally differentiated cells can establish de novo gene silencing irreversibly.

The authors use XX female mouse neural progenitor cells (NPCs), which have undergone X-inactivation, and ectopically over-express Xist RNA from the inactive-X chromosome. They test for de novo silencing of X-inactivation escape genes, which normally "escape" silencing by Xist RNA and are expressed from both the active- and inactive-X chromosomes. Several hypotheses have been put forth to explain why certain genes escape X-inactivation, and it is thought that constitutive escape genes might harbor dosage-sensitive housekeeping functions (Bellott et al., 2014, *Nature*, PMID: 24759411). Given this, escape genes may be under distinct regulation and selective pressures compared to genes typically subject to X-inactivation.

A better experiment to test their question would be to induce Xist RNA from the X chromosome in post-mitotic cells that do not normally express Xist RNA. It is unfortunate that the authors did not perform such an experiment (e.g. XY male NPCs or post-mitotic cells harboring the dox-inducible Xist promoter).

The authors should discuss the possibility that their observations for reversible and irreversible silencing of X-inactivation escape genes might not be universally applicable for all X-linked genes.

In response to reviewer comments, the authors have included several new pieces of data, which significantly improve the manuscript. In a new main figure, they now over-express Xist in astrocytes to test if their observations from NPCs hold true in a terminally differentiated, post-mitotic cell type. In addition to DNA methylation profiling, the authors now perform chromatin profiling for the PRC2-catalyzed mark histone H3K27me3 and PRC1-catalyzed histone H2AK119ub1 mark to test if these chromatin modifications are recruited to escape genes silenced reversibly or irreversibly by Xist RNA over-expression. Moreover, they now test the impact of Xist RNA over-expression in the extra-embryonic tissues of E8.5 embryos. The E8.5 data are considerably stronger than the E3.5 and E4.5 data that were originally presented.

A few points should be clarified further.

Major Points

1. The title of the manuscript suggests that multiple levels of Xist RNA will be examined. The manuscript, however, tests a single sustained increase in Xist RNA level over time. The wording of the title should be changed to reflect this. The authors show that at a sustained exposure to the same amount of elevated Xist RNA, more and more escape genes become progressively silenced over time in one NPC clone, E6. However, this progressive increase in escape gene silencing upon prolonged Xist over-expression is not observed in the other two NPC clones, CL30 and CL31. Moreover, the astrocytes differentiated from NPC clone E6 also do not exhibit a time-dependent increase in escape gene silencing. Therefore, the pattern of progressive escape gene silencing seen in clone E6 does not seem to represent the majority of the data. Clone E6 also has significantly more escape genes prior to the increase in Xist RNA expression relative to CL30 and CL31. The authors should present a comparison of Xist RNA expression levels between all three of these NPC lines. If the authors' primary argument is correct, then CL30 and CL31 clones should express higher levels of Xist RNA relative to E6, since fewer genes escape silencing in CL30 and CL31 clones.

2. In addition to showing escape gene allelic ratios, the authors should show what happens to X-linked genes subject to silencing due to X-inactivation upon the loss of SPEN in Figure 3. The authors are arguing that de novo silencing of X-linked genes can be established during a later phase of differentiation than previously thought. It thus stands to reason that at this stage genes originally subject to X-inactivation are also expected to become de-repressed in the absence of silencing factors like SPEN. A de-repression of genes originally subject to X-inactivation would be consistent with the authors' contention of a larger window of X-inactivation establishment. If genes originally subject to X-inactivation are not de-repressed in the absence of SPEN, the authors should consider if the regulation of escape genes may be inherently different than that of genes silenced originally due to X-inactivation, as mentioned above. Perhaps genes escape X-inactivation due to the cells requiring those genes to escape X-inactivation, promoting the 'reversibility' of their silencing that is observed upon washout of dox (see below).

3. In Figure 5, the authors show that ectopic silencing of escape genes by Xist RNA over-expression can be reversed upon washout of doxycycline. The authors induce Xist RNA over-expression for a range of days (3-21), but the washout period is the same (7 days) for all dox induction lengths. Would a longer or equivalent washout length (e.g. 14 days of induction followed by 14 days washout) result in a greater number of escape genes being considered 'reversibly' silenced? The new chromatin profiling data in Figure 6 suggest that with increased Xist RNA over-expression, the H3K27me3 chromatin mark increases in a time-dependent manner. It would thus be expected that a greater accumulation of silencing chromatin modifications with prolonged Xist over-expression might require a longer washout period to sufficiently deplete these modifications.

Minor Points

1. In reference to new Figure 2C, the authors state in the text that Xist RNA is upregulated 19.7-37.6-fold. The y-axis in Figure 2C goes up to 20, so it is unclear how this range was determined.

2. Figure 6G is difficult to interpret with the inclusion of all genes on the Xi (both subject to X-inactivation and silenced de novo). The authors should just present the CpG methylation of the de novo silenced genes.

Addressing of comments by Reviewer 1:

Several points raised by Referee #1 were adequately addressed in the revised manuscript. Notably, the authors now examine the impact of Xist overexpression in terminally differentiated astrocytes, investigate its effects on autosomal gene expression, and extend their in vivo analyses to a later embryonic stage (E8.5). However, some of Referee #1's concerns remain incompletely addressed.

1. Referee #1 suggested examining Xist overexpression in fully differentiated cells rather than partially differentiated NPCs. The authors addressed this by using astrocytes, which is a sufficient response. However, the data indicate that astrocytes do not exhibit a time-dependent progressive silencing of X-inactivation escapees, unlike NPCs.

2. Referee #1 noted that the authors' in vivo analyses at E3.5 and E4.5 involved imprinted XCI, whereas their in vitro work in NPCs involved random XCI, making direct comparisons problematic. In the revised manuscript, the authors still do not analyze random XCI in vivo. They justify this by arguing that imprinted and random XCI share similar kinetics of gene silencing. However, alternative approaches—such as single-cell RNA-seq or leveraging a Xist mutation on the Castaneus X

chromosome to bias XCI—could have allowed for a more direct in vivo comparison. Instead, the authors added an additional time point (E8.5) for imprinted XCI, which strengthens their in vivo data but does not address the absence of random XCI in vivo analysis.

3. Referee #1 requested that the authors compare their identified escape genes in NPCs and early embryos to previously established categories of early-, intermediate-, or late-silenced escape genes. The authors find that their escapees generally fall into the “not silenced/escape” or “late/slow silencing” groups. How many of their reversibly- vs. irreversibly-silenced escape genes identified in this study belong to those two categories above? If the irreversibly-silenced escape genes belong to the late/slow silencing group this would be consistent with the authors’ hypothesis that the window of silencing is much larger than previously thought.

4. Given previous reports that Xist RNA can spread to autosomes when overexpressed, Referee #1 asked the authors to examine autosomal gene expression changes. The authors now provide such an analysis and argue that the absence of clustered differentially expressed autosomal genes suggests a lack of direct Xist-mediated effects. However, they should be cautious in concluding that Xist has no autosomal impact, as 3D chromatin organization could cause localized effects that do not manifest as widespread clustering. Additionally, the lack of overlap in differentially expressed autosomal genes between cell types is not necessarily unexpected, given potential differences in chromatin architecture and Xist proximity.

5. Finally, several textual clarifications were made at Referee #1’s request.

**For Nature Portfolio general information and news for authors, see <http://npg.nature.com/authors>.

Version 1:

Decision Letter:

Our ref: NCB-A57253A-Z

2nd September 2025

Dear Dr. Loda,

Thank you for submitting your revised manuscript "Escape from X inactivation is directly modulated by Xist non-coding RNA" (NCB-A57253A-Z) and for your patience. It has now been seen by the original referees and their comments are below. The reviewers find that the paper has improved in revision, and therefore we'll be happy in principle to publish it in Nature Cell Biology, pending minor revisions to satisfy the referees' final requests and to comply with our editorial and formatting guidelines.

Please ensure to textually address the final points from Reviewer#3 - discuss limitations and appropriately tone down claims (Referee#2 pt 2) in a revised version.

If the current version of your manuscript is in a PDF format, please email us a copy of the file in an editable format (Microsoft Word or LaTeX)—we can not proceed with PDFs at this stage.

*Please note our articles must have 5 to 8 main figures and they can have up to 10 Extended Data figures. they become supplementary figures. Supplementary materials are less accessed than our main and ED figures so we try to limit the use of supplementary figures as much as we can. Therefore, we advise that you consolidate your figures to 8 main figures and up to 10 extended data figures.

Please ensure that all figures fit into a single standard page and adhere to a maximum page size of roughly 180mm wide x 200mm high, but also please use the full page space to fill the figure. At present all figures are too tiny to be legible once re-sized during the production process. To ensure legibility once figures are re-sized, please use a font size of no smaller than 6pt Arial or Helvetica throughout the figures.

We are now performing detailed checks on your paper and will send you a checklist detailing our editorial and formatting requirements in about 1-2 weeks. **Please do not upload the final materials and make any revisions until you receive this additional information from us**. You will be able to make all the requested textual revisions and formatting modifications and then submit the files.

Thank you again for your interest in Nature Cell Biology Please do not hesitate to contact me if you have any questions.

Best wishes,
Sabrya

Sabrya Carim, PhD
(she/her/hers)
Senior Editor, Nature Cell Biology
Nature Portfolio

Springer Nature
The Campus, 4 Crinan Street, London N1 9XW, UK
sabrya.carim@springernature.com
<https://orcid.org/0000-0001-9485-1938>

Reviewer #2 (Remarks to the Author):

The authors have addressed my questions, and I greatly appreciate their efforts in doing so.

Reviewer #3 (Remarks to the Author):

Here, the authors show that increased expression of Xist can induce de novo silencing of genes that normally escape X-inactivation. Prior work by Wutz & Jaenisch suggested that Xist RNA-mediated silencing is restricted to an early, narrow window during the differentiation of female embryonic stem cells (ESCs). However, the authors provide evidence that this restriction can be overcome by elevating Xist RNA levels, resulting in the de novo silencing of genes that escape X-inactivation outside of that window.

The authors have added additional analyses and made textual changes to improve the manuscript. In particular, the authors now include analyses with longer periods of Xist RNA upregulation in CL30 and CL31 ESC lines and evaluate the impact of longer doxycycline washout periods. A few key issues remain, though.

1. In their rebuttal the authors clarify that unlike neural progenitor cell (NPC) clone E6, NPC clones CL30 and CL31 were derived under doxycycline-induced Xist RNA expression conditions. This distinction should be made clear in the main text. The authors should clearly explain that CL30 and CL31 were exposed to higher Xist RNA levels during their differentiation, which may have irreversibly silenced genes that otherwise would have escaped X-inactivation (e.g. facultative escape genes). This past exposure may account for the reduced number of X-inactivation escapees observed in these clones relative to E6, independent of their uninduced Xist RNA expression levels. Thus, a direct correlation of uninduced Xist RNA levels vs. number of escape genes in clones E6 and CL30/31 may not be straightforwardly interpretable. However, readers may still expect such a comparison given the authors' emphasis on the anti-correlation between Xist RNA levels and number of escapees. Therefore, the "Xist histories" of these clones should be clarified in the main text to preempt confusion and to reconcile this apparent inconsistency.

2. Foundational work by Wutz & Jaenisch (2000) defined a relatively narrow window during which Xist RNA can initiate gene silencing during ESC differentiation. The data in the present manuscript suggest this window may be broader than previously appreciated. In the prior round of revisions, this reviewer asked whether genes normally subject to X-inactivation might become de-repressed in SPEN-deficient cells, which would support a broader window of X-inactivation. The authors responded by citing their 2020 study showing that SPEN is not required to maintain X-linked gene silencing when depleted for up to 48 hours. However, in the current manuscript SPEN is depleted for 3 or 7 days. Given that gene reactivation may occur gradually, especially for previously silenced loci, it is reasonable to test whether longer-term SPEN loss leads to reactivation of Xist RNA-silenced genes. The authors already have RNA-seq data in hand to perform this analysis. Even if the authors remain confident that SPEN is dispensable to maintain silencing of inactivated genes, testing X-linked gene reactivation at later time points would substantiate that claim and more thoroughly address the possibility of an extended window of differentiation during which silencing can occur.

3. In response to Reviewer #2, the authors now quantify the percentage of post-mitotic astrocytes by staining for proliferation marker Ki67. The authors find that the majority of astrocytes are post-mitotic, with 0.73% of cells staining positively for Ki67. The authors should include the simulated data from Figure R1 in the manuscript to preempt the concern that the ~1% of actively dividing astrocytes explain the observed silencing of escapees upon doxycycline treatment.

Version 2:

Decision Letter:

Dear Dr Loda,

I am pleased to inform you that your manuscript, "Escape from X inactivation is directly modulated by Xist non-coding RNA",

has now been accepted for publication in Nature Cell Biology.

Please note that *Nature Cell Biology* is a Transformative Journal (TJ). Authors may publish their research with us through the traditional subscription access route or make their paper immediately open access through payment of an article-processing charge (APC). Authors will not be required to make a final decision about access to their article until it has been accepted. [Find out more about Transformative Journals](https://www.springernature.com/gp/open-research/transformative-journals)

Authors may need to take specific actions to achieve compliance with funder and institutional open access mandates. If your research is supported by a funder that requires immediate open access (e.g. according to [Plan S principles](https://www.springernature.com/gp/open-science/plan-s-compliance) or the [NIH public access policy](https://www.springernature.com/gp/open-science/us-federal-agency-compliance)) then you should select the gold OA route, and we will direct you to the compliant route where possible. Because authors warrant under our subscription licensing terms that they haven't committed to licensing any version of their article under a licence inconsistent with the terms of our agreement – including the applicable embargo period – publication under the subscription model isn't suitable for authors whose funders require no embargo.

If you have not already done so, we strongly recommend that you upload the step-by-step protocols used in this manuscript to protocols.io (<https://protocols.io>), an open online resource that allows researchers to share their detailed experimental know-how. All uploaded protocols are made freely available and are assigned DOIs for ease of citation. Protocols and Nature Portfolio journal papers in which they are used can be linked to one another, and this link is clearly and prominently visible in the online versions of both. Authors who performed the specific experiments can act as primary authors for the Protocol as they will be best placed to share the methodology details, but the Corresponding Author of the present research

paper should be included as one of the authors. By uploading your Protocols onto protocols.io, you are enabling researchers to more readily reproduce or adapt the methodology you use, as well as increasing the visibility of your protocols and papers. You can also establish a dedicated workspace to collect your lab Protocols. Further information can be found at <https://www.protocols.io/help/publish-articles>.

Nature Cell Biology encourages authors presenting evidence for cell, biological, molecular, and genetic interactions to consider communicating these findings using Biofactoid (<https://biofactoid.org/>). This tool helps users share a searchable representation of interactions (e.g. binding, gene expression, post-translational modification) between genes, gene products, or chemicals. Information added to Biofactoid, with author attribution, is shared on social media and public databases, such as Pathway Commons, where it can be discovered and analyzed in the context of a large and growing corpus of knowledge.

With kind regards,

Daryl

Daryl Jason Verzosa David, PhD

Senior Editor, Nature Cell Biology
Advisory Editor, npj Biological Physics and Mechanics
Nature Portfolio

Heidelberger Platz 3, 14197 Berlin, Germany
Email: daryl.david@nature.com
ORCID: <https://orcid.org/0000-0002-9253-4805>

** Visit the Springer Nature Editorial and Publishing website at http://editorial-jobs.springernature.com?utm_source=ejp_NCB_email&utm_medium=ejp_NCB_email&utm_campaign=ejp_NCB for more information about our career opportunities. If you have any questions please click [here](mailto:editorial.publishing.jobs@springernature.com).

In light of the reviewers' comments, we are glad to provide a revised version of our manuscript in which we included substantial new data to address the remaining concerns. We believe that the additional data has further strengthened our conclusions, and clarified the outstanding issues raised by reviewers 2 and 3.

NEW ANALYSES AND REPLICATES.

Below is a summary of the new data and analyses in the manuscript:

- (i) To address the reviewer's concern that a subset of dividing cells account for the silencing effect observed upon Xist upregulation, we newly profile the cycling state of the differentiated astrocytes. Our results that 99% of astrocytes are non-dividing eliminates this possible alternative explanation (Extended Data Fig. 5a).
- (ii) To provide even more robust statistical significance supporting the essential role of SPEN in Xist-mediated silencing, we provide additional replicates for the entire SPEN time-course experiment for two independent clones (Fig. 3 and Extended Data Fig. 6).
- (iii) To demonstrate that Xist upregulation leads to progressive silencing of escapees over time in all tested NPC clones and not only in clone E6, we provide new data obtained upon Xist upregulation up to 21 days in clones CL30 and CL31 (Extended Data Fig. 2).
- (iv) To address the reviewer's query about whether longer doxycycline wash-out lengths following 14 days of Xist upregulation lead to higher levels of escapee gene reactivation, we provide new data obtained in three independent NPC clones showing that longer doxycycline wash-out results in only a limited increase in the number of reversibly silenced genes (Extended Data Fig. 10).
- (v) To more robustly characterize the autosomal effect in NPC clones we added more replicates to our analysis of the autosomal effects upon Xist upregulation (Extended Data Fig. 3).

MANUSCRIPT REVISED FOR CLARITY.

We appreciate from the reviewers' comments that certain aspects of our revised manuscript may not have been sufficiently clear. We have addressed these points by heavily revising the relevant sections.

- (i) To address the reviewer's concern about a potential effect of Xist-mediated silencing of autosomal genes in trans, we clarified that even though our data do not provide striking evidence for a trans silencing effect, we cannot formally exclude this scenario, nor a potential common autosomal effect across different cell types.
- (ii) To address the reviewer's concern about lack of time-dependent gene silencing in astrocytes, we clarified that we did not aim to compare time-responsiveness between NPCs and astrocytes as it would be difficult to compare the kinetics of silencing between dividing and non-dividing cells. In fact, the unanticipated finding of our experiments in astrocytes is that Xist RNA has the capacity to initiate XCI, even in non-dividing cells, a context that to our knowledge was never tested before.

(iii) To acknowledge the reviewer's point about the generalisability of the 3D topology experiments, we toned down our general conclusions as our study focused on two clusters of escapees on the Xi.

(iv) We now explicitly discuss the fact that our *in vivo* analyses in post-implantation embryos focus on the impact of Xist up-regulation on escapees of the paternal Xi in extraembryonic tissues rather than on random XCI in the embryo-proper.

Point-by-point responses to the specific points made by the reviewers follow. All changes that were made to the main text are highlighted in green in the revised version of the manuscript. The changes that were made during the previous round of revision remain in blue in the main text.

Reviewers' comments:

Reviewer #2 (Remarks to the Author):

This reviewer appreciates the authors' efforts in addressing the previous questions and acknowledges that the paper presents useful and timely data.

However, it remains challenging to ascertain the precise conclusions and the degree to which the available data support them. In particular, it would be helpful to clarify which observations can or cannot be generalized and to ensure that no alternative explanations are unintentionally overlooked.

We thank the reviewer for their helpful suggestions. We have now revised the results and discussion to more clearly convey the precise conclusions of our study, as listed here. We demonstrated that:

1) Increased Xist RNA levels can effectively silence escapees (of all types) in both dividing and non-dividing differentiated cells, and in pre- and post-implantation mouse embryos (Fig. 1, 2, 7 and Extended Data Fig. 2, 5, 12)

2) SPEN is the key mediator of Xist's impact on escapee silencing in differentiated cells (Fig. 3 and Extended Data Fig. 6)

3) Higher Xist levels *in vivo* can also lead to effective gene silencing of escapees, during a developmental time window (E3.5-E8.5) that was not previously thought to be permissive for Xist's silencing action (Fig. 7 and Extended Data Fig. 12)

4) Upon prolonged exposure to higher Xist levels, silencing becomes irreversible for most facultative escapees and this is likely to be mainly dependent on promoter DNA methylation (Fig. 5, 6 and Extended Data Fig. 11).

5) The 3D TAD-like architecture of two large escapee regions on the inactive X are dependent on the activity of the genes within them, rather than on Xist RNA alone. Without SPEN-mediated gene silencing, increased Xist RNA levels alone are not sufficient for loss of TAD-like structures at the *Mecp2-Hcfc1* cluster on the Xi (Fig. 4 and Extended Data Fig. 7, 8, 9).

Our analyses involve not only *in vitro* differentiated cells (dividing NPCs and post-mitotic astrocytes) but also *in vivo* pre- and post-implantation mouse embryos, and we even examined human data sets (Extended Data Fig. 1e). Therefore even if our study is not exhaustive for all cell

types and stages of life, our conclusion, that Xist RNA levels can have a major impact on escapee silencing is true in multiple differentiated cell types and cell states.

Most of our conclusions are based on X-chromosome-wide transcriptional and epigenomic data (RNA-seq, CUT&RUN, EM-seq), covering various cell types and developmental stages, and can therefore be broadly generalized—except for point 5.

For point 5, we focused on two major escapee clusters on the Xi (plus one more in ESCs; see Extended Data Fig. 9). These regions were key to discovering TAD-like structures on the Xi (Giorgetti et al., Nature 2016). While we believe this focus supports our conclusions, we acknowledged the reviewer's concern about generalisability and revised the results and discussion sections accordingly.

The following points detail specific major and minor concerns:

Major Points

In Vitro Differentiated Astrocytes

While the newly added data on in vitro differentiated astrocytes is appreciated, further evidence seems necessary to support the claim that “Xist RNA can efficiently initiate gene silencing in terminally differentiated post-mitotic cells.” Specifically, what is the percentage of any remaining dividing cells in the culture? How can the authors rule out that the observed silencing effect does not simply occur within a small fraction of these dividing cells?

We thank the reviewer for asking for clarification on this important point. To confirm that the observed silencing effect is seen in non-dividing cells, we performed immunofluorescence staining using the nuclear proliferation marker Ki67, which is a classic marker that is strictly associated with cell proliferation and is absent in non-dividing cells [PMID: 10653597]. This result can be found in Extended Data Figure 5a. No sign of Ki67 can be seen and the E6 differentiated astrocytes are indeed non-dividing. Upon quantification we found that less than 1% of these differentiated astrocytes were Ki67-positive (0.73%, n=541; Extended Data Fig. 5a). For comparison, 75% of E6 NPCs (the precursors of E6 astrocytes) were Ki67-positive, as expected for dividing cells. This confirms that our astrocyte cultures are a uniform, non-dividing population, allowing us to confidently assess the impact of Xist over-expression on escapee expression and hopefully addresses the reviewer's concerns about a dividing subpopulation of cells in the astrocytes.

This new data is shown in Extended Data Figure 5a, and described in lines 295-297:

“The quantification of Ki-67-positive cells in differentiated astrocytes confirmed that 99,3% of the cell population is non-dividing, whereas in E6 NPCs 75% of cells are dividing prior to differentiation [Extended Data Fig. 5a].”

We also ran computer simulations to explore whether the observed escapee silencing in astrocytes might be explained by a small number of dividing cells. Comparing allelic ratios in Dox treated and untreated astrocytes showed strong silencing (Figure R1, Top). Simulated mixes with 1–10% treated NPCs had minimal effect, while 20–40% caused broader shifts (Figure R1, Bottom). Since these patterns do not match treated astrocytes and less than 1% of astrocytes are dividing, we feel that we can be confident that the silencing observed cannot be due to NPC contamination.

Figure R1: Simulation of mixtures of silencing and non-silencing cells

(a) Top: Density plots of allelic ratios of X-linked escapees in untreated (grey) and Doxycycline-treated astrocytes (dark red), demonstrating effective silencing upon Dox treatment. Bottom: density plots of simulated allelic ratios of X-linked escapees in untreated astrocytes with defined fractions of Dox-treated NPCs (0-40%) . Low values indicate absence of escape from XCI (allelic ratio <0.1).

Line 148–152: Inverse Correction

The “inverse correction” that was found through the authors’ particular approach—without explicitly accounting for cell-type heterogeneity—may not be directly relevant to the findings in mouse NPCs.

We believe the reviewer is referring to an inverse correlation. It is not totally clear to us why this would be considered irrelevant to our mouse NPC findings. The human in vivo data supports the broader relevance of our results. We used the same approach as Richart et al. (Cell, 2022) [PMID: 35597241], which identified an anticorrelation between XIST levels and escapees in breast cancer. We therefore considered this a valid strategy.

In addition, our own findings in NPCs and mice, are consistent with both Richart et al. in normal mammary and cancer cells, and Wen et al. (Nature, 2024) [PMID: 38538789], which focused on the human brain. As noted in the discussion, our analysis compares different brain regions with varying XIST levels—not within individual cell types. We have tried to be clear about the limitations of the human data, including the absence of allelic resolution, in lines 146–148:

“Even if random XCI in these samples prevents us from assessing allele-specific Xi transcriptional activity of escapees, we found that in brain areas in which we detected higher levels of XIST, escapees are expressed at lower levels, supporting our observation in mouse NPCs [Extended Data Fig. 1e].”

Thanks to the reviewer’s comments, we have now further clarified the approach we took in the discussion:

Lines 884-888:

“Our analysis of human brain transcriptomic data from the GTEx project similarly found that higher levels of XIST RNA correlate with lower expression levels of escapees. In particular, the anti-correlation was found across different cell types of the brain characterized by higher or lower XIST expression levels. A similar approach showed that XIST RNA levels vary across breast cancer subtypes, with those expressing the highest XIST levels being characterized by lower levels of breast cancer specific X-linked escape⁴⁷”

Lines 309–314: Limited Overlap and Different Cell Types

The statement that a limited overlap between NPCs and astrocytes indicates “differing autosomal effects across different cell types” may require additional context or data. A small overlap alone might not conclusively demonstrate lack of shared target genes or mechanism.

We thank the reviewer for their request to clarify about the differences we see between NPCs and astrocytes. In the previously revised manuscript we described our observations in two different contexts of Xist up-regulation: dividing NPCs and derived, non-dividing astrocytes. Our data in these two contexts suggest that autosomal effects seen following Xist up-regulation may differ in different cell types. We have now changed the relevant text to avoid it being misleading and to take into account the reviewer’s comment that our findings cannot completely rule out shared target genes or mechanism:

Lines 316-318:

“When we compared the subsets of genes that are affected in NPCs and astrocytes, we found limited overlap, suggesting that the autosomal targets of Xist upon Xist upregulation are potentially different across different cell types [Extended data Fig. 3l,m]”

Essential Role of SPEN

The claim that “SPEN is essential for Xist-mediated silencing of nearly all escapees in differentiated cells” appears to rely heavily on Extended Data Figure 6, where the statistical significance is lacking and directionality of each gene's change is not entirely clear.

We thank the reviewer for their comment and would like to clarify as there appears to have been a misunderstanding here: In Extended Data Fig. 6 our claims are supported by strong statistical significance. Here is the relevant text from the original figure legend:

“We observe a significant decrease in allelic ratios upon dox treatment after 3d (CL30: $p_{\text{adj}} = 2.1\text{e-}12$, CL31: $p_{\text{adj}} = 1.4\text{e-}8$) and 7d (CL30: $p_{\text{adj}} = 1.0\text{e-}12$, CL31: $p_{\text{adj}} = 4.8\text{e-}14$) but no significant differences between Control and Dox treated samples in the presence of auxin after 3d (CL30: $p_{\text{adj}} = 1$, CL31: $p_{\text{adj}} = 1$) and 7d (CL30: $p_{\text{adj}} = 1$, CL31: $p_{\text{adj}} = 1$)”

What this means is that we found significant silencing of escapees when looking at untreated control (i.e. no Dox, no Aux) vs 3 days and 7 days of Dox treatment (i.e. Xist up-regulation in the presence of SPEN), but not when we compare untreated control vs Dox+Aux treatment (i.e. Xist up-regulation in the absence of SPEN) after 3 and 7 days of Xist induction.

Our data thus demonstrates that there is no escapee silencing following induction of Xist RNA in the absence of SPEN - therefore there is no statistical difference in allelic ratios +/- Xist induction (i.e. no difference in SPEN depleted cells treated with Dox when compared with the control in which Xist upregulation is not induced).

To avoid any confusion, we have now corrected the previously misleading lines as follows:

Lines 362-366:

“Whilst the induction of increased Xist RNA levels for 3 and 7 days in the presence of SPEN leads to statistically significant silencing of escapees in both NPC clones (i.e. -Dox vs +Dox comparison), when this is done in the absence of SPEN (i.e. +Dox,+Aux), no significant changes in the allelic-ratios of escapees were found (i.e. -Dox vs +Dox,+Aux comparison) [Fig 3b-d, Extended Data Fig. 6a, Supplementary Table 6]. This data clearly demonstrates that silencing of escapees by increased Xist RNA levels cannot occur in the absence of SPEN [Fig. 3b-d, Extended Data Fig. 6a, Supplementary Table 6].”

Furthermore, thanks to the reviewer's comment we now added new replicates for all tested time points and for both CL30 and CL31 clones. This new data supports our findings with even more robust statistical significance as shown in new Fig. 3 and described in the legend of Extended Data Fig. 6:

“We observe a significant decrease in allelic ratios upon Dox treatment after 3d (CL30: $p_{\text{adj}} = 5.7\text{e-}28$, CL31: $p_{\text{adj}} = 4.6\text{e-}17$) and 7d (CL30: $p_{\text{adj}} = 3.3\text{e-}37$, CL31: $p_{\text{adj}} = 1.7\text{e-}21$) compared to Control, but no significant differences between Control and Dox treated samples in the absence of SPEN (i.e. +dox,+aux) after 3d (CL30: $p_{\text{adj}} = 1$, CL31: $p_{\text{adj}} = 1$) and 7d (CL30: $p_{\text{adj}} = 1$, CL31: $p_{\text{adj}} = 1$)”

We also now show the pairwise comparison between cells treated with Dox (Xist upregulation) and cells treated with Dox+Auxin (Xist upregulation and absence of SPEN) for both NPC clones. We found statistically significant differences between these two conditions (+Dox vs +Dox+Aux), demonstrating that in absence of SPEN Xist upregulation does not result in significant silencing of escapees. This pairwise comparison is also described in the legend of Extended Data Fig. 6 as follows:

“We observe significant changes in allelic ratios upon Dox treatment in the presence of SPEN compared to its absence (i.e. +dox, -/+aux) after 3d (CL30: $p_{\text{adj}} = 2.3e-21$, CL31: $p_{\text{adj}} = 5e-13$) and 7d (CL30: $p_{\text{adj}} = 5.2e-25$, CL31: $p_{\text{adj}} = 6e-15$)”

Finally, we found statistically significant changes in escapees allelic ratios upon SPEN loss (i.e. +Aux) in both NPC clones CL30 and CL31 (p value < 0.05). This data is shown in Extended Data Fig. 6a and discussed as follows:

Lines 359-361:

“Depletion of SPEN in absence of Xist upregulation led to significant upregulation of escapees, in line with previous data showing that SPEN dampens gene expression on the Xi to a certain extent [Extended Data Fig. 6a].”

As for the directionality of each gene, the overwhelming majority of escapees show the expected direction of change. Six escapees previously shown dampening of gene expression in the absence of SPEN (i.e. *Bcor1*, *Ctps2* and *Ap1s2* in CL30; *Srpk3*, *Pdzd4* and *Tmem29* in CL31). By including new replicates to our analysis we found that *Tmem29* is not consistently dampened in the absence of SPEN, whereas *Bcor1* and *Pdzd4* dropped out of the informative genes, as these are lowly expressed genes and did not pass the threshold to be assigned as escapees across all replicates. In the case of *Ctps2*, *Ap1s2* and *Srpk3* these three genes are still dampened in the absence of SPEN as in the case of *Zdhhc9* and *G530011O06Rik* after 7 days of Dox induction. We discussed this in lines 367-372, as follows:

“Indeed the overwhelming majority of escapees across the three categories were unaffected by Xist overexpression in the context of SPEN depletion [Fig. 3b-d]. A few exceptions included the escapees *Ctps2*, *Ap1s2*, *Srpk3*, *Zdhhc9* and *G530011O06Rik* which are dampened upon *Xist* upregulation even in the absence of SPEN in at least one clone or time point [Fig. 3b]. However this is in line with previous observations showing that SPEN is dispensable for silencing of a small subset of X linked genes during XCI.”

Additionally, the response to the previous question 4c does not seem to have been incorporated into the revised manuscript, leaving an alternative explanation of the data unaddressed.

Following the reviewer's advice we now included these considerations in the revised manuscript. The analysis performed with new replicates confirms our observations with additional insights. In the case of *Gm7331*, this gene dropped out of the informative genes as it did not pass the escape threshold in all replicates. For *G6pdx*, the data does not show a consistently higher allelic ratio across clones and time points. This locus in the CL30 and CL31 clones is tagged with a GFP/Tomato Hygro/Blasticidin resistance cassette which may influence the transcriptional status of *G6pdx* on both alleles, and therefore the allelic changes observed upon Dox/Aux treatment are difficult to interpret. However, in E6 NPCs and astrocytes, the *G6pdx* locus is untagged and it becomes silenced upon Xist upregulation [Fig. 1f; Fig. 2d]. The pseudogenes *Gm14539* and *Gm8822* and *Mdm4-ps* show higher allelic ratio to some extent (either in one of the clones, or in either time point) because the annotation of SNPs at this loci is subjected to misannotations. To

avoid leaving an alternative explanation of the data unaddressed, we now incorporated these observations to the legend of Figure 3 as follows:

“The four genes *Gm14539*, *Gm8822*, *G6pdx* and *Mdm4-ps* show either unchanged or increased (rather than decreased) allelic ratio upon Dox treatment. *Gm14539*, *Gm8822* and *Mdm4-ps* are pseudogenes with homologues on other chromosomes, and the annotation of SNPs at this loci is likely to be subjected to misannotations. The *G6pdx* locus in the CL30 and CL31 clones is tagged with a GFP/Tomato Hygro/Blasticidin resistance cassette - which may influence the transcriptional status of this particular gene on both alleles. The allelic changes observed for *G6pdx* are therefore difficult to interpret in these clones.”

Question of Generalizability (Previous Question 5)

While this reviewer apologizes that the originally suggested analysis might not have been fully applicable, the core issue remains: how do the authors ensure that the findings from just two analyzed clusters can be generalized to all XCI escapees?

We thank the reviewer for asking us to better justify whether we can extrapolate what we see in these two regions, to other XCI escapee regions. The *Mecp2-Hcfc1* and *Kdm5c* clusters are two of the largest regions of escape that led to the discovery of TAD-like structures on the Xi [Giorgetti et al., Nature 2016]. Our Capture Hi-C data here is consistent with this previous study and also provides a much higher allelic resolution view of these Xi domains. We now explain in the text that we studied these two large regions of escape but we did not formally test changes in 3D topology at all escapees clusters on the Xi:

Lines 926-928:

“We show that in NPCs this reversibility in 3D chromatin organisation at the “*Mecp2-Hcfc1*” cluster is not simply a consequence of higher levels of Xist RNA coating the Xi, but strictly depends on SPEN-mediated silencing.”

Lines 931-936:

“While we expect these effects on 3D topology upon Xist-mediated silencing of escapees to occur at all clusters of escapees on the Xi, in this study we focus on two large domains, namely *Mecp2-Hcfc1* and *Kdm5c* clusters for the following reasons: (i) the “*Mecp2-Hcfc1*” cluster is the most gene dense region on the Xi and encompassed up to 30 facultative escapees (1 Mb in the Capture Hi-C) showing the most variable patterns of silencing and reactivation kinetics; (ii) the “*Kdm5c*” cluster, includes a single constitutive escapee *Kdm5c*, surrounded by a cluster of facultative escapees.”

Minor Points

Placement of Related Work

The expanded discussion on the differences between this and related studies is helpful. It may further benefit readers if these related works are briefly introduced earlier in the manuscript to set expectations for the Results section (though this is not strictly required).

We are glad the clarified discussion is helpful

References to Lifespan

The toned-down discussion of lifespan claims is appreciated. However, it may strengthen clarity to remove indirect references to lifespan entirely if there is no direct evidence within the manuscript to support them.

We removed the toned-down references to the lifespan.

Lines 876-877:

“Our study provides direct evidence that Xist RNA participates in the regulation of X-linked escapees in a post-developmental context.”

Lines 879-881:

“Given our findings, Xist/XIST RNA levels are therefore likely to be critical for sex-specific regulatory programs, which, when misregulated, may lead to phenotypic consequences including sex-biased diseases.”

Reviewer #3 (Remarks to the Author):

Hauth, Panten, Heard and colleagues present a revised manuscript testing whether the Xist lncRNA can silence escape genes outside of the putative developmental window when Xist RNA establishes silencing of genes on the inactive X chromosome. Based on initial observations by Wutz & Jaenisch (2000, *Molecular Cell*, PMID: 10882105), Xist RNA-mediated gene silencing is reversible in early stages of embryonic stem cell (ESC) differentiation but becomes irreversible upon further differentiation. Here, the authors revisit this observation and test if overexpression of Xist RNA in differentiating and terminally differentiated cells can establish de novo gene silencing irreversibly.

The authors use XX female mouse neural progenitor cells (NPCs), which have undergone X-inactivation, and ectopically over-express Xist RNA from the inactive-X chromosome. They test for de novo silencing of X-inactivation escape genes, which normally “escape” silencing by Xist RNA and are expressed from both the active- and inactive-X chromosomes. Several hypotheses have been put forth to explain why certain genes escape X-inactivation, and it is thought that constitutive escape genes might harbor dosage-sensitive housekeeping functions (Bellott et al., 2014, *Nature*, PMID: 24759411). Given this, escape genes may be under distinct regulation and selective pressures compared to genes typically subject to X-inactivation.

A better experiment to test their question would be to induce Xist RNA from the X chromosome in post-mitotic cells that do not normally express Xist RNA. It is unfortunate that the authors did not perform such an experiment (e.g. XY male NPCs or post-mitotic cells harboring the dox-inducible Xist promoter).

We thank the reviewer for this new suggestion. However, we believe inducing Xist from the single, active X chromosome in male cells would not be informative in elucidating how the escapees on the inactive X chromosome are controlled by Xist RNA, which is the main point of our study. The

situation in male cells would be totally different as Xist RNA is not normally expressed in males and the X chromosome is fully active and in a very different overall chromatin state. In the present study, we set out to test whether Xist up-regulation can silence escapees, and for this it is the inactive X chromosome and its escapees that must be tested. Furthermore in male cells, upregulating Xist could silence essential X-linked genes where there is only one copy of these genes and would lead to rapid cell death, as shown previously in ES cells by Wutz and Jaenisch in 2000 [PMID: 10882105] and 2002 [PMID: 11780141]. Thus inducing Xist in male cells is a very different question to the one we have set out to explore in our present study, which is to test the role of *increased* Xist RNA levels on the few genes that are expressed (escape XCI) from an already inactivated X chromosome and that are also expressed from the active X.

The authors should discuss the possibility that their observations for reversible and irreversible silencing of X-inactivation escape genes might not be universally applicable for all X-linked genes.

We were not totally clear what the reviewer is referring to, as we do not generalize our findings to all X-linked genes—only to escapees. We show that nearly all escapees are sensitive to increased Xist RNA levels in a SPEN-dependent manner. However, constitutive escapees can regain expression when Xist levels drop, unlike most variable escapees, which remain silenced. This highlights that irreversible silencing does not apply to all escapees and suggests that specific mechanisms allow constitutive escapees to resist permanent silencing. We describe this in the text as follows:

Lines 942-951

“Constitutive escapees are thought to be highly dosage-sensitive⁷⁹. Thus, they presumably need to be protected from complete Xist-mediated silencing not only during development, when XCI is first initiated, but also in adult tissues, as any increase in Xist RNA levels could potentially lead to their inactivation with deleterious effects on tissue homeostasis. Indeed, the dosage regulation of constitutive escapees that have retained a Y-chromosome homolog during the evolution of sex chromosomes is thought to be absolutely critical because these genes are involved in many fundamental functions such as chromatin regulation, protein translation and ubiquitination^{79,80}. The capacity of constitutive escapees to override even prolonged *Xist* upregulation, suggest that these genes are likely to have evolved gene-specific strategies that allow them to avoid the repressive epigenetic machinery brought to the Xi by Xist and SPEN.”

In response to reviewer comments, the authors have included several new pieces of data, which significantly improve the manuscript. In a new main figure, they now over-express Xist in astrocytes to test if their observations from NPCs hold true in a terminally differentiated, post-mitotic cell type. In addition to DNA methylation profiling, the authors now perform chromatin profiling for the PRC2-catalyzed mark histone H3K27me3 and PRC1-catalyzed histone H2AK119ub1 mark to test if these chromatin modifications are recruited to escape genes silenced reversibly or irreversibly by Xist RNA over-expression. Moreover, they now test the impact of Xist RNA over-expression in the extra-embryonic tissues of E8.5 embryos. The E8.5 data are considerably stronger than the E3.5 and E4.5 data that were originally presented.

We appreciate the feedback that all of the new data provided significantly improved the manuscript.

A few points should be clarified further.

Major Points

1. The title of the manuscript suggests that multiple levels of Xist RNA will be examined. The manuscript, however, tests a single sustained increase in Xist RNA level over time. The wording of the title should be changed to reflect this.

We thank the reviewer for pointing this out and we have now changed the wording of the title as follows: “Escape from X inactivation is directly modulated by Xist non-coding RNA”

The authors show that at a sustained exposure to the same amount of elevated Xist RNA, more and more escape genes become progressively silenced over time in one NPC clone, E6. However, this progressive increase in escape gene silencing upon prolonged Xist overexpression is not observed in the other two NPC clones, CL30 and CL31.

To address this point raised by the reviewer, we tested longer times of Xist upregulation in both clones CL30 and CL31, for up to 21 days (and not only 3 and 7 days as in the previous version of our manuscript). We found that the statistically significant differences in escapees allelic ratios between control and Dox-treated samples after 3, 7, 14 and 21 days of Dox induction increases across the time course, with adjusted p values ranging from $2.15e-13$ at day 3 to $4.8e-17$ at day 21 of Dox treatment for CL30 and from $4.1e-10$ at day 3 to $8.2e-12$ at day 21 of Dox treatment for CL31. This data demonstrates time-dependent silencing of escapees upon Xist overexpression for both clones, and is shown in Extended Data Figure 2d. We described this data in lines 191-193, as follows:

“Similar results were obtained upon Xist upregulation up to 21 days in CL30 and CL31 clones, with silencing of escapees showing increasing statistical significance as the time of Dox treatment increases [Extended Data Fig. 2d].

While this analysis provides statistical significance for time-dependent silencing efficiency in CL30 and CL31, we would like to point out that the much lower number of escapees in CL30 (61) and CL31 (48) compared to E6 (133) renders it difficult to make a direct comparison. In fact, the majority of the escapees that become silenced upon Xist induction in E6, are already silenced in CL30 and CL30 to start with. To underline these differences we now included the number of escapees in each of the clones in the Figure legend of Fig. 1 and Extended Data Fig. 2 and we discuss this point as follows:

Lines 193-196:

“This data demonstrate progressive Xist-mediated silencing over time in all tested NPC clones, even if a direct comparison of silencing dynamics between clone E6 and clones CL30 and CL31 is limited by the

lower number of escapees in CL30 (61) and CL31 (48) compared to clone E6 (133) [Fig. 1g; Extended Data Fig. 2f].”

Moreover, the astrocytes differentiated from NPC clone E6 also do not exhibit a time-dependent increase in escape gene silencing. Therefore, the pattern of progressive escape gene silencing seen in clone E6 does not seem to represent the majority of the data.

In this study we did not set out to test time-responsiveness in astrocytes, because we believe that dividing and non-dividing cells are not really comparable timing-wise. Our aim here was to test whether increased levels of Xist RNA can initiate XCI in post-mitotic cells. The data shown in Figure 2 and Extended Data Fig. 5 demonstrates that is the case as we found that the majority of escapees are silenced by day 3 of Dox in Astrocytes as we observed in NPCs. This data also show that in astrocytes silencing of escapees does not augment by day 7 of Dox treatment. Thanks to the reviewer’s comment we have now better explained these findings:

Lines 303-312:

“Allelic-ratios of the majority of escapees changed significantly following 3 and 7 days of Xist upregulation compared to untreated astrocytes, demonstrating that Xist RNA can efficiently initiate gene silencing in terminally differentiated, post-mitotic cells [Fig. 2d, Extended data Fig. 5e-g]. However, while 3 days of Dox led to similar escapee silencing effects in NPCs and astrocytes, with 58% and 57% of genes showing significant reduction in allelic ratio of at least 50%, respectively, 7 days of increased Xist levels do not lead to further silencing in astrocytes, unlike in NPCs [Fig. 2e, Extended data Fig. 5e-g, Extended data Fig. 2i]. This may reflect different kinetics of gene silencing between dividing and non-dividing cells. Non-dividing cells may need longer exposure to Xist RNA levels to enable efficient escapee silencing. Alternatively, cell division may be a requirement to establish efficient gene silencing over time.”

Clone E6 also has significantly more escape genes prior to the increase in Xist RNA expression relative to CL30 and CL31. The authors should present a comparison of Xist RNA expression levels between all three of these NPC lines. If the authors’ primary argument is correct, then CL30 and CL31 clones should express higher levels of Xist RNA relative to E6, since fewer genes escape silencing in CL30 and CL31 clones.

We thank the reviewer for raising this important point. In Fig. 1a of our manuscript, we compare 21 NPC clones derived without Dox-induced Xist modulation, from wildtype XX F121 ESCs and found a significant anticorrelation between Xist RNA levels and escape. Clone E6 was generated similarly from TX1072 ESCs. In contrast, CL30 and CL31 were derived from TX1072 ESCs treated with Dox throughout differentiation, leading to elevated Xist levels from the start (as in Dossin et al., Nature 2020). This was done to skew XCI towards the BL6 allele. The reduced number of escapees in these lines probably reflects their prolonged Xist exposure during differentiation, consistent with our conclusion that timing and level of Xist expression affect escape. Therefore, while all three clones are useful for studying the impact of dialing up Xist levels from an inactive X chromosome that shows different degrees of escape, they cannot be used to compare the basal levels of Xist and escape with clone E6, as their Xist histories differ. Thus a direct comparison of basal Xist and escape levels is not appropriate.

2. In addition to showing escape gene allelic ratios, the authors should show what happens to X-linked genes subject to silencing due to X-inactivation upon the loss of SPEN in Figure 3. The authors are arguing that de novo silencing of X-linked genes can be established during a later phase of differentiation than previously thought. It thus stands to reason that at this stage genes originally subject to X-inactivation are also expected to become de-repressed in the absence of silencing factors like SPEN. A de-repression of genes originally subject to X-inactivation would be consistent with the authors' contention of a larger window of X-inactivation establishment. If genes originally subject to X-inactivation are not de-repressed in the absence of SPEN, the authors should consider if the regulation of escape genes may be inherently different than that of genes silenced originally due to X-inactivation, as mentioned above. Perhaps genes escape X-inactivation due to the cells requiring those genes to escape X-inactivation, promoting the 'reversibility' of their silencing that is observed upon washout of dox (see below).

We have previously shown (Dossin et al. Nature 2020), that SPEN is not required to maintain the stable repression of previously silenced X-linked genes in NPCs. Maintenance of silencing is probably ensured by epigenetic modifications such as DNA methylation as outlined in the Dossin et al. paper.

The reviewer asks whether we consider that genes escaping XCI may be inherently different to those that are originally silenced due to XCI. Indeed this is a fundamental question that we have explored in our study:

- (i) all escapees are sensitive to Xist RNA, if the levels of Xist are increased. In this sense these genes are not inherently different in their sensitivity to Xist RNA and SPEN. However they are clearly different in their degree of responsiveness and the stability of the changes seen.
- (ii) we show that some escapees become irreversibly silenced after prolonged higher Xist exposure - and thus are probably not inherently different from X-inactivated genes.
- (iii) finally we show that constitutive escapees are able to re-express and not become irreversibly silenced. This suggests that they may have an inherent difference in resisting the downstream epigenetic silencing compared to other genes on the X chromosome.

3. In Figure 5, the authors show that ectopic silencing of escape genes by Xist RNA over-expression can be reversed upon washout of doxycycline. The authors induce Xist RNA over-expression for a range of days (3-21), but the washout period is the same (7 days) for all dox induction lengths. Would a longer or equivalent washout length (e.g. 14 days of induction followed by 14 days washout) result in a greater number of escape genes being considered 'reversibly' silenced?

This is indeed an interesting question. As requested by the reviewer we now include RNA-seq data showing the effects of 14 days of Dox washout following 7 and 14 days of Dox treatment in clone E6. We also tested 14 and 21 days of washout after 14 days of Dox in clones CL30 and CL31. This data is shown in Extended Data Fig. 10 and demonstrates that extending the time of Dox washout from 7 to 14 days leads to only a limited increase in the number of reversible genes compared to shorter washout. We described this new data and analyses in lines 528-532, as follows:

“We tested whether a longer washout of Dox after 14 days of Xist induction would increase the number of genes that become reactivated and found only a limited increase in the number of reversible genes compared to shorter washout times (7 days) [Extended Data Fig. 10 a,b]. We confirmed these results in clones CL30 and CL31 in which we tested silencing reversibility up to 21 days of Dox washout following 14 days of Xist upregulation [Extended Data Fig. 10 c-f].”

The new chromatin profiling data in Figure 6 suggest that with increased Xist RNA over-expression, the H3K27me3 chromatin mark increases in a time-dependent manner. It would thus be expected that a greater accumulation of silencing chromatin modifications with prolonged Xist over-expression might require a longer washout period to sufficiently deplete these modifications.

We agree with the reviewer that a longer washout of Dox may be required to see a stronger decrease in H3K27me3 enrichment and we now explicitly discuss this possibility, in lines 921-922:

“In particular we cannot exclude that longer washout of Dox would result in further decrease of H3K27me3 enrichment on the Xi”

Minor Points

1. In reference to new Figure 2C, the authors state in the text that Xist RNA is upregulated 19.7-37.6-fold. The y-axis in Figure 2C goes up to 20, so it is unclear how this range was determined.

We thank the reviewer for pointing this out, this was a mistake and we correct it.

Lines 301-303:

“Xist upregulation in astrocytes following 3 and 7 days of Dox was manifest both by RNA-FISH [Fig. 2b] and RNA-seq [Fig. 2c], with a 17.52 - 21.14-fold enrichment compared to untreated controls [Fig. 2c].”

2. Figure 6G is difficult to interpret with the inclusion of all genes on the Xi (both subject to X-inactivation and silenced de novo). The authors should just present the CpG methylation of the de novo silenced genes.

While the changes in promoter methylation of de novo silenced genes are shown in Fig. 6h, in Fig 6g we show changes at CpG islands chromosome-wide (without assigning CpG islands to specific genes). We think the two plots give complementary information in terms of DNA methylation along the entire Xa and Xi chromosomes - and at escapees promoters.

Addressing of comments by Reviewer 1:

Several points raised by Referee #1 were adequately addressed in the revised manuscript. Notably, the authors now examine the impact of Xist overexpression in terminally differentiated astrocytes, investigate its effects on autosomal gene expression, and extend their in vivo analyses to a later embryonic stage (E8.5).

However, some of Referee #1's concerns remain incompletely addressed.

We appreciate the acknowledgement by the reviewer that we adequately addressed several of the points raised by Reviewer 1. We clarify the remaining points below.

1. Referee #1 suggested examining Xist overexpression in fully differentiated cells rather than partially differentiated NPCs. The authors addressed this by using astrocytes, which is a sufficient response. However, the data indicate that astrocytes do not exhibit a time-dependent progressive silencing of X-inactivation escapees, unlike NPCs.

Please also refer to our comment above. In this study we did not perform a long time-course experiment in astrocytes because it was not our aim to test time-responsiveness in non-dividing cells. In fact, we induced Xist upregulation for 3 and 7 days in astrocytes knowing that in NPCs the difference in gene silencing efficiency becomes more clear at later time points (after 14 and 21 days). Our experiments demonstrate that Xist RNA can initiate XCI, even in post-mitotic cells, a context that was never addressed before. We believe that this is a novel insight to fully understand Xist silencing function. To acknowledge the reviewer's concern and avoid misunderstanding, we now further discuss the differences observed between NPCs and astrocyte in lines 303-312, as follows:

“Allelic-ratios of the majority of escapees changed significantly following 3 and 7 days of Xist upregulation compared to untreated astrocytes, demonstrating that Xist RNA can efficiently initiate gene silencing in terminally differentiated, post-mitotic cells [Fig. 2d, Extended data Fig. 5e-g]. However, while 3 days of Dox led to similar escapee silencing effects in NPCs and astrocytes, with 58% and 57% of genes showing significant reduction in allelic ratio of at least 50%, respectively, 7 days of increased Xist levels do not lead to further silencing in astrocytes, unlike in NPCs [Fig. 2e, Extended data Fig. 5e-g, Extended data Fig. 2i]. This may reflect different kinetics of gene silencing between dividing and non-dividing cells. Non-dividing cells may need longer exposure to Xist RNA levels to enable efficient escapee silencing. Alternatively, cell division may be a requirement to establish efficient gene silencing over time.”

2. Referee #1 noted that the authors' in vivo analyses at E3.5 and E4.5 involved imprinted XCI, whereas their in vitro work in NPCs involved random XCI, making direct comparisons problematic. In the revised manuscript, the authors still do not analyze random XCI in vivo. They justify this by arguing that imprinted and random XCI share similar kinetics of gene silencing. However, alternative approaches—such as single-cell RNA-seq or leveraging a Xist mutation on the Castaneus X chromosome to bias XCI—could have allowed for a more direct in vivo comparison. Instead, the authors added an additional time point (E8.5) for imprinted XCI, which strengthens their in vivo data but does not address the absence of random XCI in vivo analysis.

The reviewer raises some interesting suggestions. We did not perform single-cell RNA-seq analysis as the sparsity of this kind of data affects the reliable quantification of allele-specific transcriptional changes in lowly and moderately expressed genes such as escapees. However, to address the reviewer's point, we did analyse random XCI in vivo by looking at E8.5 embryonic tissues matching the extraembryonic samples shown in Figure 7 (Figure R2 a,b). This data is now

included below in Rebuttal Figure 2 and clearly shows the expected effects of Xist upregulation on escapee expression levels in tissues that underwent random XCI, including different responses across different categories of escapees (Figure R2c, d). We did not include this data in the m/s to avoid confusion as the control “no Dox/untreated” embryos underwent random XCI and their bulk RNA-seq sequencing would not be directly comparable with the treated embryos in which XCI is skewed because of Dox induction of the BL6 allele.

Figure R2: Xist RNA modulates the expression levels of escapee in embryonic tissues

a, Experimental outline: TX/Y males carrying the rtTA transactivator (Xptet/Y; R26rtTA/rtTA or Xptet/Y;R26rtTA/WT) were crossed with wild type JF1 females. Xist RNA overexpression was induced by adding Dox to the drinking water of pregnant females for 5 days, from E3.5 to E8.5. RNA-seq was performed using RNA extracted from embryonic tissues. **b**, RNA-seq data showing the fold change in *Xist* expression (normalised CPM) in treated embryos compared to untreated embryos upon Dox treatment. **c**, Heatmap showing X-linked transcript allelic ratios in untreated and Dox treated embryos. **d**, Box plots showing mean allelic ratios for treated and untreated embryos for the different escapee categories.

In the present study, we have thus looked at the effects of Xist upregulation in multiple different contexts, including NPCs, astrocytes, preimplantation embryos and post-implantation extraembryonic tissues, and we believe that our data strongly supports the general role of Xist RNA as a regulator of escapees across different cellular and developmental contexts as well as in random and in imprinted XCI. Following the Reviewer's comment we now explicitly state that our in vivo strategy focused on iXCI:

Line 791-793:

“In summary, our in vivo experiments, which focused on cell and tissue contexts where imprinted (not random) XCI has taken place, clearly demonstrate that Xist RNA is a direct regulator of XCI escape in multiple contexts (such as NPCs and astrocytes) and regardless of the initial type (imprinted or random) of XCI.”

3. Referee #1 requested that the authors compare their identified escape genes in NPCs and early embryos to previously established categories of early-, intermediate-, or late-silenced escape genes. The authors find that their escapees generally fall into the “not silenced/escape”

or “late/slow silencing” groups. How many of their reversibly- vs. irreversibly-silenced escape genes identified in this study belong to those two categories above? If the irreversibly-silenced escape genes belong to the late/slow silencing group this would be consistent with the authors’ hypothesis that the window of silencing is much larger than previously thought.

We thank the reviewer for this suggestion. We looked into this comparison and found that the 87% and 67% of reversible and irreversible genes, respectively, belong to the “late/slow silencing” category. Even if only 30 genes are still reversible, the fact that a higher percentage of reversible genes (compared to irreversible) belong to this category likely reflects their higher resistance to Xist-mediated silencing during XCI, as expected from our findings. We included this data in Supplementary Table 3.

4. Given previous reports that Xist RNA can spread to autosomes when overexpressed, Referee #1 asked the authors to examine autosomal gene expression changes. The authors now provide such an analysis and argue that the absence of clustered differentially expressed autosomal genes suggests a lack of direct Xist-mediated effects. However, they should be cautious in concluding that Xist has no autosomal impact, as 3D chromatin organization could cause localized effects that do not manifest as widespread clustering.

This is an important point that we have now clarified thanks to the reviewer's concern. While we believe that our data point to a rather indirect effect of Xist RNA on autosomal genes, we now explained that we cannot formally exclude a direct effect *in trans*, as follows:

Lines 216-221:

“The fact that autosomal genes are not only downregulated but also upregulated, and that the genes affected are not clustered at specific autosomal loci [Extended Data Fig. 3i], suggests that this is probably a secondary, indirect effect of Xist upregulation, presumably due to the change in dosage of various X-linked escapee gene products, many of which can affect chromatin and transcription. However we cannot exclude that the direct action of Xist RNA *in trans* may also account for some of the autosomal effect.”

Additionally, the lack of overlap in differentially expressed autosomal genes between cell types is not necessarily unexpected, given potential differences in chromatin architecture and Xist proximity.

We thank the reviewer for pointing this out and we further toned down our consideration as follows:

Lines 316-318:

“When we compared the subsets of genes that are affected in NPCs and astrocytes, we found limited overlap, suggesting that the autosomal effects observed upon Xist upregulation are potentially different across different cell types [Extended data Fig. 3l,m].

5. Finally, several textual clarifications were made at Referee #1’s request.

We appreciate the reviewers' acknowledgement that our revised manuscript has satisfactorily addressed the concerns raised in the previous round of review. Below, we clarify the remaining points and indicate the corresponding changes made in the manuscript, including the addition of the requested data.

Reviewer #2:

Remarks to the Author:

The authors have addressed my questions, and I greatly appreciate their efforts in doing so.

We are glad to note that all remaining questions have been addressed.

Reviewer #3:

Remarks to the Author:

Here, the authors show that increased expression of Xist can induce de novo silencing of genes that normally escape X-inactivation. Prior work by Wutz & Jaenisch suggested that Xist RNA-mediated silencing is restricted to an early, narrow window during the differentiation of female embryonic stem cells (ESCs). However, the authors provide evidence that this restriction can be overcome by elevating Xist RNA levels, resulting in the de novo silencing of genes that escape X-inactivation outside of that window.

The authors have added additional analyses and made textual changes to improve the manuscript. In particular, the authors now include analyses with longer periods of Xist RNA upregulation in CL30 and CL31 ESC lines and evaluate the impact of longer doxycycline washout periods. A few key issues remain, though.

We appreciate the feedback that the new data have improved the manuscript.

1. In their rebuttal the authors clarify that unlike neural progenitor cell (NPC) clone E6, NPC clones CL30 and CL31 were derived under doxycycline-induced Xist RNA expression conditions. This distinction should be made clear in the main text. The authors should clearly explain that CL30 and CL31 were exposed to higher Xist RNA levels during their differentiation, which may have irreversibly silenced genes that otherwise would have escaped X-inactivation (e.g. facultative escape genes). This past exposure may account for the reduced number of X-inactivation escapees observed in these clones relative to E6, independent of their uninduced Xist RNA expression levels. Thus, a direct correlation of uninduced Xist RNA levels vs. number of escape genes in clones E6 and CL30/31 may not be straightforwardly interpretable. However, readers may still expect such a comparison given the authors' emphasis on the anti-correlation between Xist RNA levels and number of escapees. Therefore, the "Xist histories" of these clones should be clarified in the main text to preempt confusion and to reconcile this apparent inconsistency.

We thank the reviewer for this suggestion and we included a clear description of the different strategies that were used to differentiate NPC clones CL30, CL31 and E6 as follows:

Line 143-146

“The reduced number of escapees in clones CL30 and CL31 is probably because they were originally differentiated in the presence of Dox from TX1072 ESCs (in order to skew XCI), unlike clone E6. Thus clones CL30 and 31 had a prolonged exposure to elevated Xist levels during their derivation, consistent with our observation that timing and level of Xist expression affect escape.”

2. Foundational work by Wutz & Jaenisch (2000) defined a relatively narrow window during which Xist RNA can initiate gene silencing during ESC differentiation. The data in the present manuscript suggest this window may be broader than previously appreciated. In the prior round of revisions, this reviewer asked whether genes normally subject to X-inactivation might become de-repressed in SPEN-deficient cells, which would support a broader window of X-inactivation. The authors responded by citing their 2020 study showing that SPEN is not required to maintain X-linked gene silencing when depleted for up to 48 hours. However, in the current manuscript SPEN is depleted for 3 or 7 days. Given that gene reactivation may occur gradually, especially for previously silenced loci, it is reasonable to test whether longer-term SPEN loss leads to reactivation of Xist RNA-silenced genes. The authors already have RNA-seq data in hand to perform this analysis. Even if the authors remain confident that SPEN is dispensable to maintain silencing of inactivated genes, testing X-linked gene reactivation at later time points would substantiate that claim and more thoroughly address the possibility of an extended window of differentiation during which silencing can occur.

As suggested by the reviewer, we tested whether genes subjected to XCI become de-repressed upon SPEN depletion in NPCs. This data is now shown in Supplementary Figure 1 and demonstrates no reactivation of Xi silenced genes in absence of SPEN, confirming that SPEN is dispensable to maintain silencing of inactivated genes as previously published in Dossin et al., 2020. We discuss this data in lines 211-212:

“We also confirmed that SPEN is dispensable for XCI maintenance as we observed no reactivation of Xi silenced genes in its absence [Supplementary Fig. 1].”

3. In response to Reviewer #2, the authors now quantify the percentage of post-mitotic astrocytes by staining for proliferation marker Ki67. The authors find that the majority of astrocytes are post-mitotic, with 0.73% of cells staining positively for Ki67. The authors should include the simulated data from Figure R1 in the manuscript to preempt the concern that the ~1% of actively dividing astrocytes explain the observed silencing of escapees upon doxycycline treatment.

As suggested by the reviewer, we added the simulated data from Figure R1 to Extended Data Fig. 5g and discuss them in line 189-191:

“To exclude that escapee silencing might be explained by a small number of dividing cells, we simulate mixtures of untreated astrocytes with defined fractions of Dox-treated NPCs [Extended Data Fig. 5g]”